# The kinetochore protein KNL-1 regulates the actin cytoskeleton to control dendrite branching

Henrique Alves Domingos[1]*, Mattie Green[1]*, Vasileios R. Ouzounidis[1], Cameron Finlayson[1], Bram Prevo[1], and Dhanya K. Cheerambathur[1]

**The function of the nervous system is intimately tied to its complex and highly interconnected architecture. Precise control of dendritic branching in individual neurons is central to building the complex structure of the nervous system. Here, we show that the kinetochore protein KNL-1 and its associated KMN (Knl1/Mis12/Ndc80 complex) network partners, typically known for their role in chromosome-microtubule coupling during mitosis, control dendrite branching in the *Caenorhabditis elegans* mechanosensory PVD neuron. KNL-1 restrains excess dendritic branching and promotes contact-dependent repulsion events, ensuring robust sensory behavior and preventing premature neurodegeneration. Unexpectedly, KNL-1 loss resulted in significant alterations of the actin cytoskeleton alongside changes in microtubule dynamics within dendrites. We show that KNL-1 modulates F-actin dynamics to generate proper dendrite architecture and that its N-terminus can initiate F-actin assembly. These findings reveal that the postmitotic neuronal KMN network acts to shape the developing nervous system by regulating the actin cytoskeleton and provide new insight into the mechanisms controlling dendrite architecture.**

## Introduction

Neurons have specialized compartments for transmitting information, including dendrites, which receive and integrate signals, and axons, which relay them. The shape, size, and trajectory of dendritic and axonal processes determines how a neuron communicates and functions. Dendrites display diverse shapes and complex branching patterns that are unique to neuronal types (Jan and Jan, 2001; Lefebvre et al., 2015). The dendrite arborization pattern defines how input signals are distributed and integrated, ultimately shaping the functional properties of neurons and their connectivity within neural circuits. Proper development and organization of dendritic arbors is therefore crucial for establishing the neural circuits that regulate animal behavior. Notably, alterations in dendrite morphology and organization are associated with neurodevelopmental and neurodegenerative disorders (Jan and Jan, 2010).

Dendrite morphogenesis initiates when extrinsic guidance cues interact with specific cell surface receptors on the neuron and activate intracellular signaling pathways that trigger cytoskeletal rearrangements to enable dendrite growth and branching (Jan and Jan, 2010; Lefebvre, 2021). The Rho family of GTPases (Cdc-42, Rac1, and RhoA) are key downstream effectors of the guidance cues during dendrite development (Li et al., 2000; Luo et al., 1996; Threadgill et al., 1997; Nakayama et al., 2000). During dendrite outgrowth, dynamic actin filaments form a network of lamellipodial and filopodial protrusions that guide and propel the growth cone. Rho GTPases influence dendrite arborization by regulating actin filament assembly and the formation of actin structures such as filopodia and lamellipodia. Dendritic arborization occurs when filopodia emerge from pre-existing branches and undergo cycles of extension and retraction until they become stabilized (Scott and Luo, 2001). Consequently, dendrite arbor growth and stabilization entail extensive actin remodeling and careful coordination between actin growth states and structures. Various actin regulators, including F-actin nucleators and modulators of F-actin dynamics, have been shown to affect dendrite morphogenesis (Sundararajan et al., 2019b; Ouzounidis et al., 2023). However, it is unclear how the activity of different actin regulators is coordinated and controlled spatially and temporally to generate specific dendrite patterns.

The microtubule cytoskeleton is also critical for dendrite morphogenesis. Microtubules provide structural support for dendrites and serve as tracks to transport key building materials and organelles. Mutations in molecular motors that affect transport severely impair dendrite morphology, and loss of microtubule-associated proteins results in dendrite development defects (Taylor et al., 2015; Maniar et al., 2011; Jaworski et al., 2009; Cao et al., 2020; Kahn et al., 2018). In addition, microtubule nucleators are found at branch points of developing

---

[1]Institute of Cell Biology, School of Biological Sciences, The University of Edinburgh, Edinburgh, UK.

*H. Alves Domingos and M. Green contributed equally to this paper.   Correspondence to Dhanya K. Cheerambathur: Dhanya.Cheerambathur@ed.ac.uk.

dendrites, where they organize microtubule arrays to regulate branching (Nguyen et al., 2014). Studies of the role of microtubules in dendritic spines, small actin-rich protrusions on dendrite branches that are sites of high synaptic activity, have shown that dynamic microtubules enter the actin-rich spines in a synaptic activity–dependent manner and that this is facilitated by actin cytoskeleton remodeling (Jaworski et al., 2009; Merriam et al., 2011; Hu et al., 2008). Thus, a crosstalk between the two cytoskeletal networks appears to exist within dendrites, but its molecular basis remains elusive.

Recent studies have revealed that kinetochore proteins, which connect chromosomes to dynamic spindle microtubules in dividing cells, play an unexpected postmitotic role during neurodevelopment. The conserved 10-subunit Knl1 complex/Mis12 complex/Ndc80 complex (KMN) network, which serves as the primary chromosome-microtubule coupler in mitosis (Musacchio and Desai, 2017), is enriched along developing dendrites and axons of Caenorhabditis elegans sensory neurons and in the synapses and axons of Drosophila melanogaster neurons (Cheerambathur et al., 2019; Zhao et al., 2019; Ouzounidis et al., 2024). Depletion of KMN components in C. elegans sensory neurons results in dendrite growth defects, indicating that KMN function is critical for dendrite development (Cheerambathur et al., 2019). Knockdown of Mis12 causes dendrite branching defects in D. melanogaster embryonic sensory neurons and increases filopodial protrusions in dendrites of rat hippocampal neurons, underscoring the importance of the Mis12 complex for proper dendrite formation (Zhao et al., 2019). Moreover, kinetochore proteins promote dendrite regeneration of D. melanogaster sensory neurons, highlighting their involvement in the maintenance of dendrite morphology (Hertzler et al., 2020).

The mechanisms by which kinetochore proteins contribute to dendrite development and maintenance are not well understood, but some observations imply that microtubule binding activity is important. The neuronal function of the KMN network in C. elegans sensory neurons depends on the microtubule coupling interface within the Ndc80 complex (Cheerambathur et al., 2019; Ouzounidis et al., 2024). Additionally, the knockdown of kinetochore proteins in D. melanogaster sensory neurons increases the number of microtubule plus ends in dendrites, suggesting a role in the regulation of microtubule dynamics (Hertzler et al., 2020).

Our previous characterization of the KMN network's postmitotic role in neurons focused on dendrite extensions of the C. elegans amphid sensory neuron bundles, which consist of dendrites from 12 neurons (Cheerambathur et al., 2019). This precluded a detailed analysis of cellular structures within dendrites at the single-neuron level. To overcome this limitation, here we examined the role of the KMN network component KNL-1 during dendrite branching of the mechanosensory neuron PVD. Dendrite development in PVD is stereotypical and invariant, and its cytoskeletal organization is well characterized, making it an ideal system for investigating kinetochore protein function. We demonstrate that KNL-1 is concentrated in the extending dendrite processes and that PVD-specific degradation of KNL-1 perturbs higher-order branch formation, behavioral defects, and premature neurodegeneration. Furthermore, we show

that KNL-1 degradation does not affect microtubule organization within the PVD but alters F-actin and microtubule dynamics. Branching abnormalities associated with KNL-1 loss were ameliorated by altering actin filament dynamics, and ectopic targeting of KNL-1 to the neuronal plasma membrane–induced F-actin assembly. We conclude that the kinetochore protein KNL-1 contributes to the dendrite guidance pathway through the regulation of the neuronal actin cytoskeleton.

## Results

### KNL-1 localizes to developing dendrites in the PVD neuron

The PVD neuron in C. elegans is a type of mechanosensory neuron located between the body wall muscle and the epidermis of the animal. A unique feature of the PVD is the highly arborized dendrites that extend throughout the animal's body (Fig. 1 A). PVD dendrites form a distinctive branching pattern composed of non-overlapping repeating units called "menorahs," which are important for integrating and processing sensory inputs (Fig. 1 A) (Smith et al., 2010; Oren-Suissa et al., 2010; Halevi et al., 2002; Tsalik et al., 2003). PVD dendrite branches are established during the larval stages and the branching follows a stereotypical developmental program that begins at the late L2 larval stage and concludes by the late L4 larval stage (Fig. 1 B). To visualize the localization of the kinetochore protein KNL-1 in the PVD neuron, we used a split-GFP system (Kamiyama et al., 2016). The N-terminus of KNL-1 was fused to seven copies of the β11 strand of GFP at the endogenous locus, and the complementing GFP β1–10 was expressed under the unc-86 promoter, which is active in the early stages of PVD development. The cell body of PVD is located near the animal tail, and two primary (1°) dendrites extend from the cell body toward the anterior and posterior of the animal. KNL-1 was enriched in the cell body and along the extending 1° dendrites at the L2 larval stage (Fig. 1 C). During the L3 and L4 stages, higher-order branches develop from 1° dendrites to generate the distinctive menorah shape. KNL-1 signal persisted along the 1° dendrites in the L3 and L4 stages but became more punctate (Fig. S1, A–C). These discrete KNL-1 puncta were found along the developing 1° and 3° dendrites as well as at a subset of 1° and 3° branch points (Fig. 1 D and Fig. S1, B–J). These results indicated that KNL-1 is present in developing PVD neurons and localized in a manner consistent with playing a role in PVD dendrite formation.

### KNL-1 and other KMN network components act to restrict dendritic branching

To investigate the function of KNL-1 in the PVD neuron, we utilized an auxin-inducible degron (AID) to selectively degrade KNL-1 (KNL-1 DEG). We fused an AID peptide sequence along with GFP to KNL-1 at the endogenous locus and expressed the plant-specific F-box protein TIR1 under PVD-specific promoters using the Mos1-mediated single copy insertion system (Fig. 2 A) (Ashley et al., 2021). To ensure TIR1 expression throughout PVD development, we utilized two promoters: the unc-86 promoter, which drives expression in PVD immediately after differentiation, and the des-2 promoter, which initiates expression at the

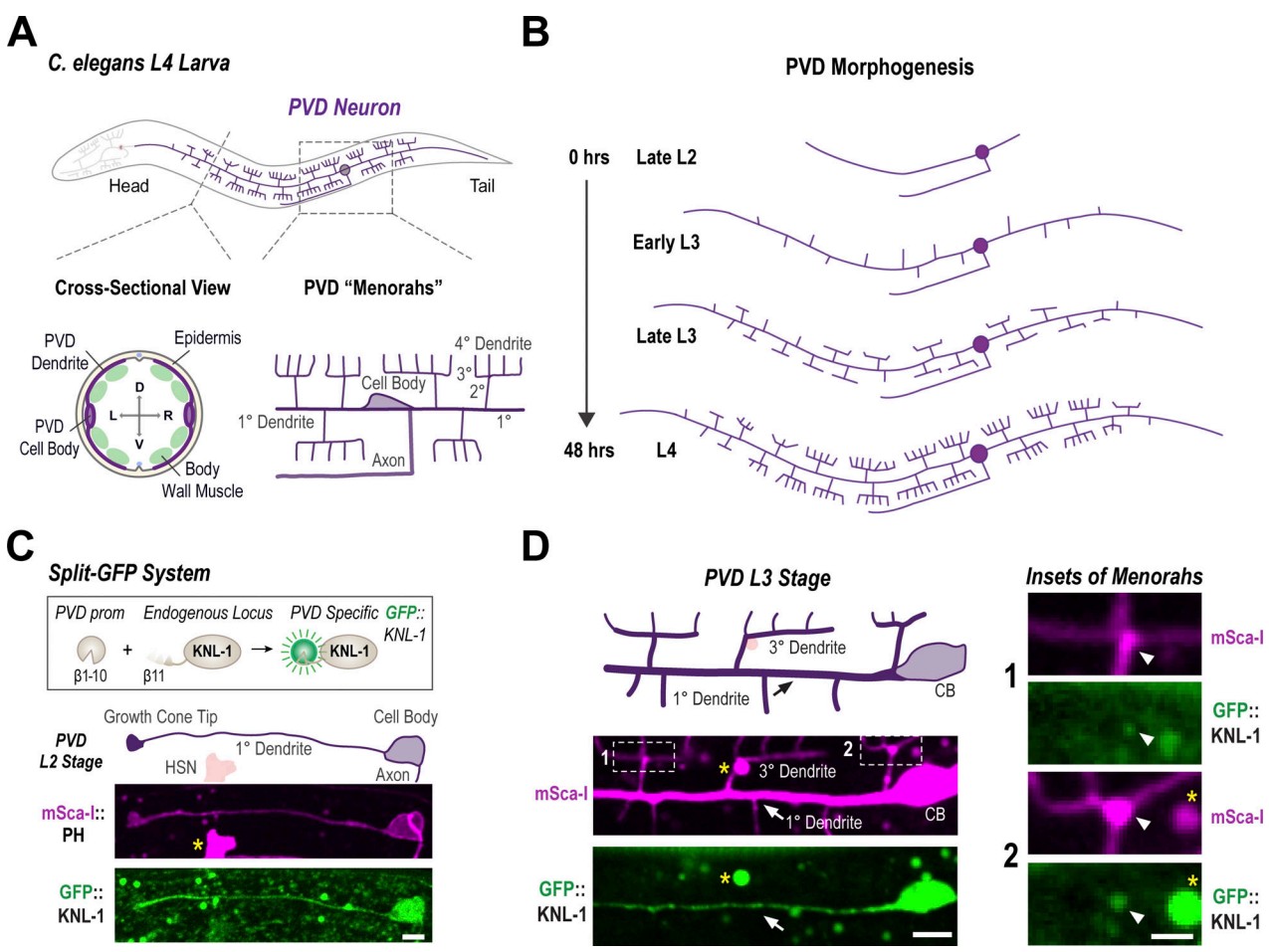

Figure 1. **KNL-1 localization in the PVD neuron. (A)** Schematic of the PVD neuron at the L4 larval stage (top). The bottom left inset shows a cross-sectional view of the PVD, located between the epidermis and body wall muscle (D, dorsal; V, ventral; L, left and R, right). The bottom right inset illustrates the discrete "menorah"-like dendrite branching pattern of the PVD neuron. The two primary (1°) branches emanate in both anterior and posterior directions from the cell body, while the higher-order branches emerge perpendicularly from the pre-existing ones. The axon of the PVD extends ventrally from the cell body and incorporates into the ventral nerve cord. **(B)** Schematic representing the PVD neuron morphogenesis during the larval stages of *C. elegans*. PVD development initiates at the L2 larval stage. By the late L2 stage, the anterior and posterior primary branches, along with the axon begin to extend. The primary (1°) dendrites are fully extended by the early L3 stage and secondary (2°) dendrites begin to emerge. During the late L3 and L4 stages, there is extensive branching as the tertiary (3°) and quaternary (4°) branches start growing. By the late L4 larval stage (~48 h at 20°C) the "menorah" structures are fully formed and dendritic branching completed. **(C)** Schematic of split-GFP system used to express KNL-1 in the PVD neuron during the development of the neuron (top). The images below show membrane labeled with mScarlet-I (mSca-I) fused to the PH domain of rat PLC1δ1 (magenta) and GFP:KNL-1 (green) within the developing dendrite at the L2 larval stage. The yellow star represents GFP::KNL-1 in the cell body of HSN, another neuron where the *unc-86* promoter is active. Scale bar, 5 μm. **(D)** Schematic (top) showing the anterior menorahs of PVD at L3 stage. Images (below) show the distribution of mSca-I (magenta) and GFP::KNL-1 (green). Arrow highlights 1° dendrite. The images are maximum intensity projections of four z-planes spaced 0.5 μm apart. Scale bar, 5 μm. Insets (right) highlight the 3° dendrites. Arrowheads highlight 3° branch points. Yellow stars represent autofluorescence granules. Scale bar, 2 μm.

late L2 and early L3 stages. We refer to this double TIR1 expression degron system as the PVD degron throughout the remainder of the manuscript. To degrade KNL-1, we synchronized the worms as L1 larvae and exposed them to auxin at the L2 larval stage when PVD development is initiated (Fig. S2 A). We observed a significantly reduced GFP::KNL-1 signal in the PVD cell body at the late L2 stage (Fig. 2 A). To assess the effect of KNL-1 degradation on PVD development, we visualized dendrite morphology in L4 larvae expressing a plasma membrane (pleckstrin homology domain, PH) marker. In both control animals, which expressed the TIR1 transgenes in untagged KNL-1 background, and KNL-1 degrader animals the 1° dendrites of the PVD neuron extended properly. However, we observed a

significant disruption of the higher-order branching pattern following KNL-1 degradation (Fig. 2 B and Fig. S2 G). Menorahs consist of dendrite branches that are arranged at right angles with respect to each other. Specifically, each menorah contains a 2° dendrite that branches off at a right angle from the 1° dendrite, which then gives rise to an orthogonal 3° branch that forms the base of menorah. Finally, several smaller quaternary (4°) dendrites emerge at right angles to the 3° dendrite to complete the menorah. The number of 2° dendrite branches was not significantly different between control and KNL-1 degrader neurons (Fig. S2 B). However, KNL-1 degrader neurons exhibited an increased number of 4° dendrite branches (Fig. 2 C) and a higher occurrence of overlaps between 3° dendrites

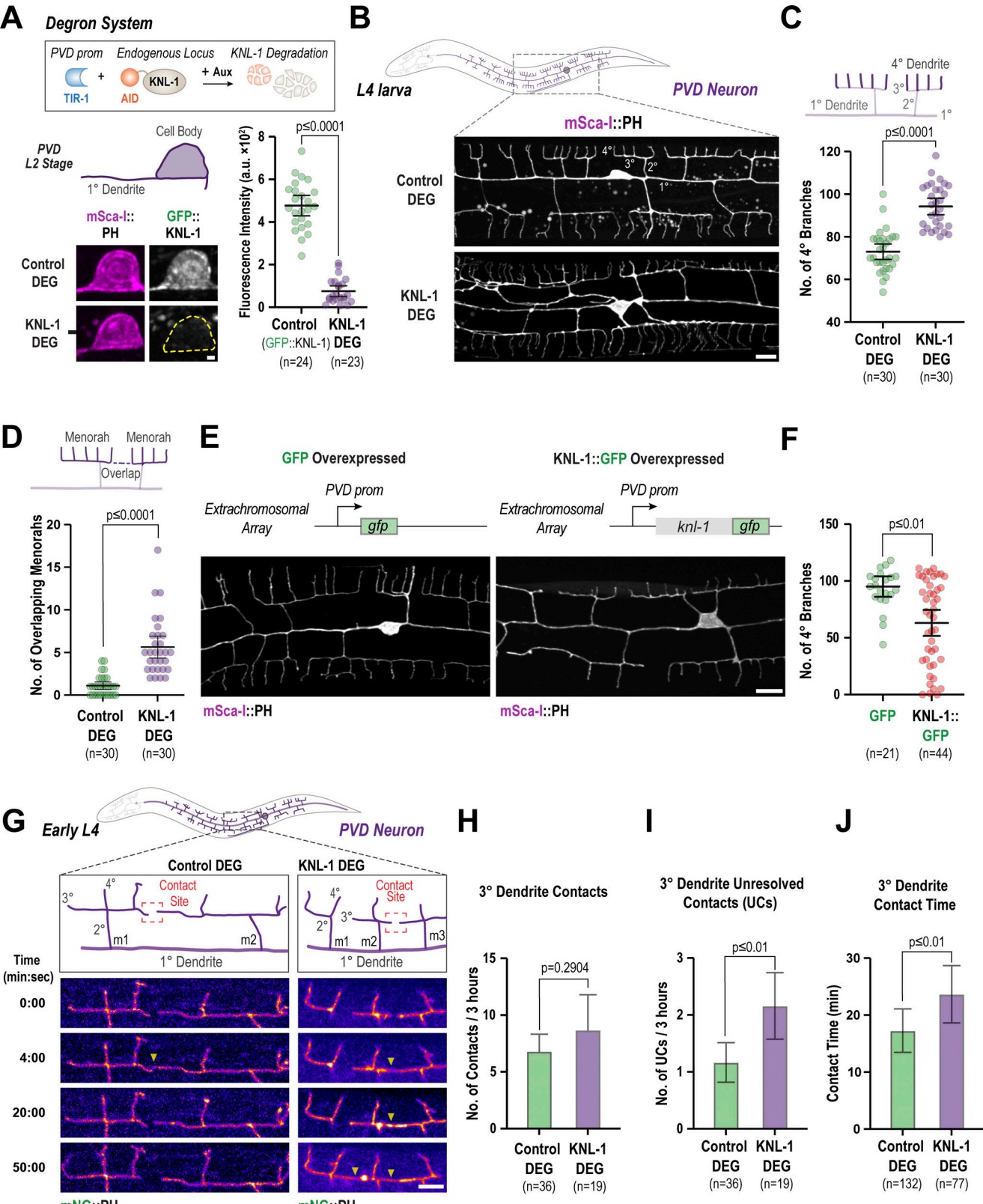

Figure 2. **KNL-1 is required for proper dendrite branching of the PVD neuron. (A)** Approach employed for the degradation of endogenously tagged AID::KNL-1 in the PVD neuron (top). Images below show GFP::KNL-1 signal (gray) within the PVD cell body (magenta) at the late L2 larval stage, in the absence (Control DEG) and presence of auxin (KNL-1 DEG). The graph on the right shows the mean fluorescence intensity of the GFP signal in the cell body for control and KNL-1 DEG conditions. *n* represents the number of animals. Error bars indicate mean ± 95% confidence interval (CI). P values from unpaired *t* test. Scale bar, 1 μm. **(B)** Images of PVD dendrite organization at the L4 larval stage in the control and KNL-1 DEG animals. These images show both the anterior and posterior dendrites extending from the cell body within a 100 μm region. Scale bar, 10 μm. **(C and D)** Quantification of PVD dendrite architecture determining

the number of 4° branches and overlaps between menorahs in control and KNL-1 DEG. *n* represents the number of animals. Error bars represent mean ± 95% CI. P values from unpaired *t* test (C) or Mann–Whitney test (D). **(E)** Images of PVD dendrites in GFP (control) and KNL-1::GFP overexpressing animals. In both conditions the transgenes were overexpressed as an extrachromosomal array under a PVD specific promoter. *n* represents the number of animals. Scale bars, 10 μm. **(F)** Quantification of the number of 4° branches in the PVD in GFP and KNL-1::GFP overexpressing animals. *n* represents the number of animals. Error bars represent mean ± 95% CI. P values from Mann–Whitney test. **(G)** Time-lapse imaging of 3° dendrite branch dynamics in control and KNL-1 DEG animals. Yellow arrowheads denote the contact sites between 3° dendrites of two adjacent menorahs. Scale bar, 5 μm. **(H)** Quantification of number of contacts made by tertiary dendrites of the PVD neuron over a 3-h time interval in the indicated conditions. *n* represents number of animals. Error bars represent mean ± 95% CI. P values from Mann–Whitney test. **(I)** Quantification of number of unresolved contacts (UCs) made by tertiary dendrites of the PVD neuron over a 3-h time interval in the indicated conditions. *n* represents the number of animals. Error bars represent mean ± 95% CI. P values from Mann–Whitney test. **(J)** Quantification of time spent in contact by 3° dendrites of two adjacent menorahs. *n* represents the number of contacts. Error bars represent mean ± 95% CI. P values from Mann–Whitney test.

---

(Fig. 2 D). Furthermore, we observed a greater number of membrane protrusions emerging from the 1° dendrites (Fig. S2 C). These ectopic protrusions were distinct from the 2° dendrites that had already formed the characteristic menorah-like structure by this stage. The dendritic morphology defects were consistently observed throughout the PVD neuron, as both anterior and posterior dendrites exhibited similar morphological disruptions upon KNL-1 depletion (Fig. S2, D and E).

We next overexpressed KNL-1 by generating extrachromosomal arrays that contain multiple copies of transgenic KNL-1::GFP under a PVD-specific promoter. Compared with control animals that contained a GFP-expressing plasmid in the array, animals with KNL-1::GFP arrays exhibited a decrease in higher-order branching. Specifically, the number of 2° and 4° dendrite branches was significantly reduced (Fig. 2, E and F; and Fig. S2 F). We conclude that PVD-specific degradation and overexpression of KNL-1 have opposite effects on higher-order dendritic branching.

We also investigated the effect of degrading the Mis12 and the Ndc80 complexes within the KMN network. Degradation of KNL-3 (a component of the Mis12 complex) or the NDC-80 subunit of the Ndc80 complex disrupted PVD dendrite architecture in a similar manner to KNL-1 degradation (Fig. S2, B, C, and G–I). Taken together, these results reveal that PVD-localized KNL-1 functions in the context of the KMN network to restrict dendritic branching. We concluded that the kinetochore–microtubule coupling machinery has a critical role in ensuring proper PVD dendrite arborization.

### KNL-1 regulates contact-dependent repulsion between dendrite branches

A prominent effect of KNL-1 degradation is an increase in the number of overlapping neighboring menorahs, a phenotype typically observed in mutants that affect dendritic self-avoidance (Fig. 2 G) (Smith et al., 2012; Liao et al., 2018; Sundararajan et al., 2019a; Hsu et al., 2020). Dendritic self-avoidance is a process in which the growing 3° dendrites of adjacent menorahs undergo contact-dependent repulsion to prevent overlap between sister dendritic branches (Smith et al., 2012). The increase in aberrant overlapping menorahs observed following KNL-1 degradation (Fig. 2 D) suggests that KNL-1 may play a role in resolving contacts between the growing tips of neighboring 3° dendrites. We tracked the growth dynamics of 3° dendritic tips during the early L4 stage to determine whether KNL-1 influences the spatial separation of the menorahs through modulation of

dendritic self-avoidance mechanisms. At this developmental stage, 3° dendrites project anteriorly or posteriorly and continue to grow until they encounter another 3° dendrite tip oriented in the opposite direction (Fig. 2 G). Upon contact, these adjacent 3° dendrite tips instantaneously retract. Consistent with previous studies (Smith et al., 2010, 2012; Liao et al., 2018), 3° dendrites of neighboring menorahs retracted within 10–20 min of mutual contact in control neurons (Fig. 2, G–J). In KNL-1 degrader neurons, the frequency with which neighboring 3° dendrites established contact remained unchanged (Fig. 2 H) but the contact duration was longer (Fig. 2 J). The presence of persistent contacts after KNL-1 degradation resulted in a higher frequency of unresolved contacts, indicating that KNL-1 ensures timely retraction of 3° dendrites during menorah formation (Fig. 2 I). These observations suggest that in addition to its role in restricting 4° dendrite branch number, KNL-1 activity is required to promote contact-dependent dendritic self-avoidance of 3° dendrites. Thus, the KNL-1 function is essential for spatial patterning of dendrite branches in the PVD neuron.

### KNL-1 degradation impairs the sensory behavior of the PVD and results in premature neurodegeneration

Proper dendrite arborization is essential for the sensory function of the PVD neuron (Liu and Shen, 2011). We therefore examined the effect of KNL-1 loss on two distinct sensory behaviors, proprioception and harsh touch response (Albeg et al., 2011; Way and Chalfie, 1989; Tao et al., 2019). Proprioception pertains to awareness of body position and movement, and the PVD dendrites sense mechanical stimuli and provide feedback to the body wall muscles to generate a stereotypical sinusoidal locomotory pattern (Fig. 3 A). To determine the body posture of animals, we visualized their movement on food and measured the wavelength and amplitude of their sinusoidal tracks. As a positive control, we included animals lacking DMA-1, a transmembrane receptor that is essential for PVD dendrite formation and is required for proper locomotion (Tao et al., 2019). Similar to Δ*dma-1*, the locomotory pattern of KNL-1 degrader animals showed reduced amplitude and wavelength, indicating impaired proprioception (Fig. 3, B and C). We next compared the ability of control and KNL-1 degrader animals to respond to high-threshold mechanical stimuli applied to the midbody (harsh touch). In *C. elegans*, PVD in conjunction with touch receptor neurons is required for harsh touch response (Way and Chalfie, 1989). Therefore, we evaluated PVD's harsh touch response in *mec-4* mutant

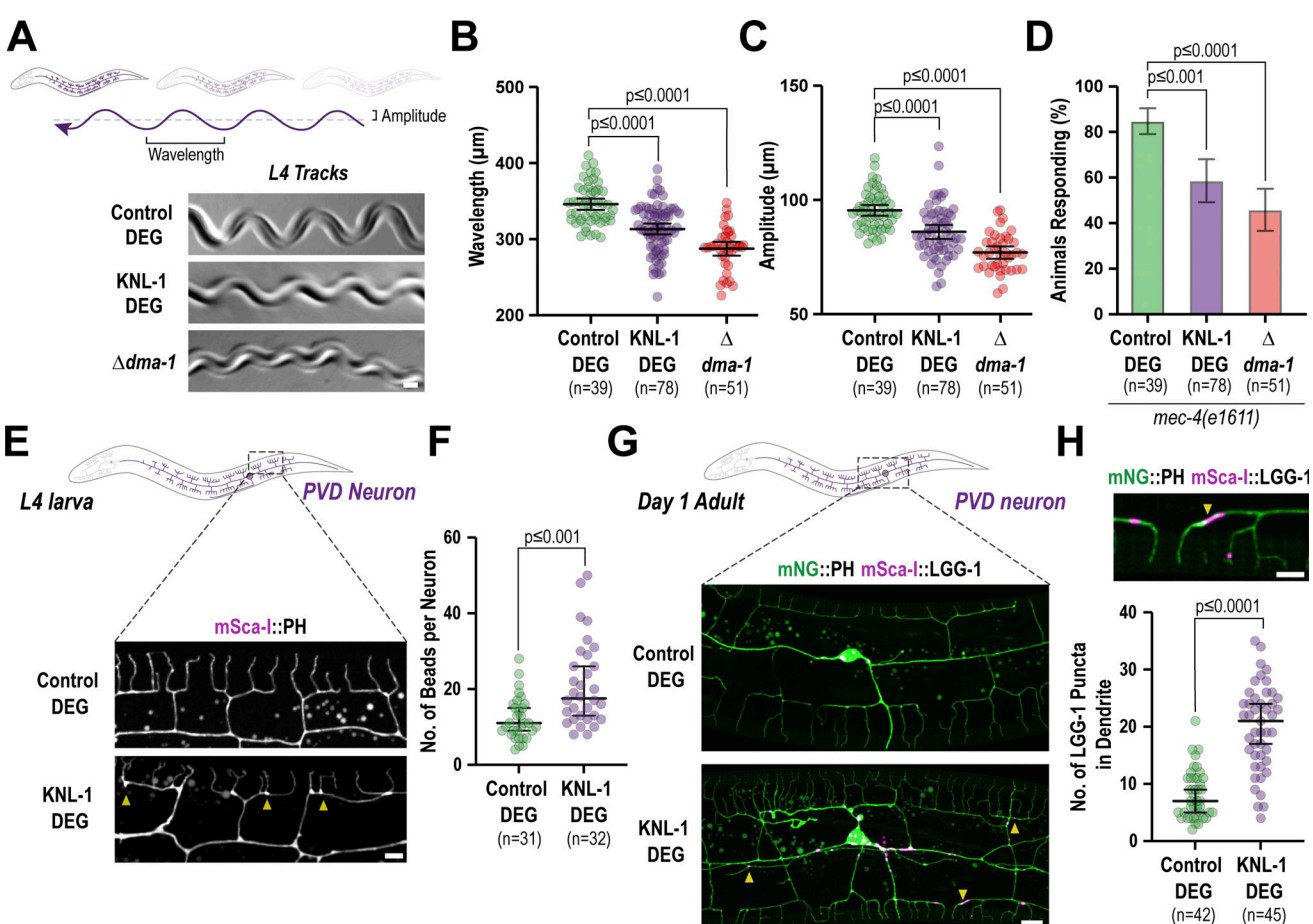

Figure 3. **KNL-1 is important for proprioceptive behavior and PVD neuron maintenance. (A)** Cartoon representation of animal movement, where the arrow represents the sinusoidal track that the animal leaves behind, defined by the wavelength and amplitude (top). Images show the tracks of control, KNL-1 DEG and Δ*dma-1* animals. The control worms form sinusoidal tracks while the amplitude and wavelength of these tracks were diminished in KNL-1 DEG and Δ*dma-1* animals. Scale bar, 20 μm. **(B and C)** Quantification of the wavelength and amplitude of the tracks by animals in indicated conditions. *n* represents the number of animals. Error bars represent mean ± 95% CI. P values from Kruskal–Wallis test followed by Dunn's test (B) or one-way ANOVA test followed by Tukey's test (C). **(D)** The plot represents the number of animals responding to harsh touch in the indicated conditions. The experiment was conducted in *mec-4(e1611)* mutant background, which lacks touch receptor neurons. *n* represents the number of animals. Error bars represent mean ± 95% CI. P values from Kruskal–Wallis test followed by Dunn's test. **(E)** Images of PVD dendrite organization at the L4 stage in control and KNL-1 DEG animals. Yellow arrowheads indicate the bead-like blebs in the plasma membrane of PVD neurons. Scale bar, 5 μm. **(F)** Quantification of the bead-like blebs in control and KNL-1 DEG animals. *n* represents the number of animals. Error bars represent mean ± 95% CI. P values from Mann–Whitney test. **(G)** Images show PVD cell body and dendrites using mNeonGreen(mNG)::PH marker (green) and autophagosomes using mScarlet-I(mSca-I)::LGG-1 marker (magenta) in the indicated conditions. Yellow arrowheads indicate the appearance of LGG-1 puncta along the dendrites of KNL-1 DEG animals. Scale bar 10 μm. **(H)** A zoomed image of the PVD dendrite showing the plasma membrane in green and autophagosomes in magenta. Arrowhead highlights mSca-I::LGG-1 puncta (top). The plot below is the quantification of mSca-I::LGG-1 puncta within the dendrites in both control and KNL-1 DEG animals. *n* represents the number of animals. Error bars represent mean ± 95% CI. P values from unpaired *t* test. Scale bar 5 μm.

background, which eliminates the touch receptor function (Liu and Shen, 2011). In contrast to control animals, a significant percentage of KNL-1 degrader animals displayed a defective touch response (Fig. 3 D). Although proprioception is regulated locally by PVD dendrites via neuropeptide release, harsh touch response requires axon-derived signals (Tao et al., 2019). KNL-1 is also found in axonal structures (Ouzounidis et al., 2024) and, it is therefore plausible that the effect of KNL-1 degradation on harsh touch response may be partly attributable to its impact on axonal and synaptic connectivity. Overall, the results of sensory assays indicate that KNL-1 ensures appropriate behavioral responses linked to the PVD neuron.

In addition to the sensory defects, we noticed morphological aberrations in the PVD architecture which were indicative of neurodegeneration (Pan et al., 2011; E et al., 2018). Specifically, at the L4 stage, the dendrites of KNL-1 degrader neurons contained more bead-like structures compared with control neurons (Fig. 3, E and F). Beading or bubble-like lesions along axons and dendrites are morphological defects that have been previously shown to be associated with age-dependent decline of neurons in *C. elegans* (E et al., 2018; Pan et al., 2011). In PVD dendrites, extensive beading does not appear until late into adulthood (day 6) in control neurons. The presence of substantial beading at the L4 stage suggested that KNL-1 degrader neurons may degenerate prematurely.

Age-dependent degeneration of PVD dendrites is mediated by the autophagy pathway (E et al., 2018), and autophagosomes, marked by mScarlet-I::LGG-1 and expressed from the PVD-specific *des-2* promoter, were enriched in dendrites of KNL-1 degrader neurons at the day 1 adult stage (Fig. 3, G and H). These observations suggest that KNL-1 degradation predisposes the PVD neuron to premature degeneration. Taken together, the above findings highlight that KNL-1 function is essential for PVD function and health.

**Loss of KNL-1 does not alter the neuronal polarity of the PVD**
Next, we explored the mechanism by which KNL-1 impacts PVD structure and function. Our previous work has shown that the microtubule-binding modules within the KMN network are essential to its postmitotic function in embryonic neurons (Cheerambathur et al., 2019; Ouzounidis et al., 2024). The microtubule cytoskeleton plays a critical role in establishing neuronal polarity and maintenance of the polarized distribution of proteins in the dendrite and axonal compartments of the PVD neuron (Maniar et al., 2011). Specifically, disruptions to the minus-end out organization of the anterior primary dendrite of the PVD neuron have been linked to neuronal polarity defects (Maniar et al., 2011; He et al., 2020; Dey et al., 2024).

We first tested whether KNL-1 degradation disrupts the distribution of RAB-3, a small GTPase associated with synaptic vesicles in axons, and DMA-1, a transmembrane receptor, that localizes exclusively to dendrites (Fig. 4 A). Previous studies had shown that loss-of-function mutants of microtubule-associated proteins result in the mislocalization of RAB-3 and DMA-1 to the dendritic and axonal compartments of PVD, respectively (Maniar et al., 2011; Dey et al., 2024; Eichel et al., 2022). We observed no change in the distribution of RAB-3 or DMA-1 following KNL-1 degradation, indicating that the polarized distribution of the axonal and dendritic proteins remains intact in the absence of KNL-1 (Fig. 4, B and C; and Fig. S3, A and B).

We also monitored microtubule organization in the anterior primary dendrite in control and KNL-1 degrader animals by following the dynamics of EB1[EBP-2], which binds to growing microtubule plus ends. EB1[EBP-2]::GFP was expressed under a PVD-specific promoter, and its trajectories were determined using time-lapse imaging in L4 larvae (Fig. 4 D). In control neurons, the majority of the EB1[EBP-2]::GFP puncta moved toward the cell body, consistent with previous observations (Harterink et al., 2018; Taylor et al., 2015; He et al., 2020, 2022) that microtubules are oriented with their minus end distal in the anterior 1° dendrite (Fig. 4, D and E). There was no significant difference in the direction of movement of EB1[EBP-2]::GFP comets in KNL-1 degrader neurons (Fig. 4, D and E). Taken together, our results demonstrate that the microtubule orientation in the anterior primary dendrite and PVD neuronal polarity are not impacted by KNL-1 degradation. Notably, this finding is consistent with our previous work, which showed that the axonal microtubule organization and polarized sorting of axonal proteins are not affected in the amphid sensory neurons of *C. elegans* when KNL-1 is absent (Ouzounidis et al., 2024).

**KNL-1 degradation affects microtubule dynamics within the anterior primary dendrite**
In *Drosophila* ddaE sensory neurons, kinetochore proteins impact microtubule polymer behavior within dendrites (Hertzler et al., 2020). Specifically, a reduction in kinetochore protein levels led to an increased number of polymerizing microtubule-plus ends. We examined whether KNL-1 degradation affected microtubule polymer dynamics in the PVD dendrites. We first measured the turnover of microtubules in neurons expressing GFP::TBA-1. As reported earlier, microtubules are predominantly present in 1° dendrites, and GFP::TBA-1 signal can also observed be in 2° and 3° dendrite branches (Fig. S3 C) (Maniar et al., 2011; Tang et al., 2019; Sundararajan et al., 2019a; Zhao et al., 2022). We performed fluorescence recovery after photobleaching (FRAP) experiments on microtubules labeled with endogenous GFP::TBA-1 using the split-GFP system in the anterior 1° dendrite (Fig. 4 F and Fig. S3 D). In both control and KNL-1 degrader animals, the GFP::TBA-1 signal within the anterior 1° dendrite recovered with the same kinetics (Fig. 4 F and Fig. S3 D). This suggests that the overall dynamics of the microtubule network in the 1° dendrite remain unchanged in KNL-1 degrader animals under our imaging conditions.

Next, we employed the EB1[EBP-2]::GFP probe to measure the microtubule plus end growth dynamics in the anterior 1° dendrite (Fig. 4, G–J). We observed, within our imaging time frame, no significant change in the number of EB1[EBP-2]::GFP comets or growth duration between control and KNL-1 degrader animals but a significant increase in the comet velocity and the growth length traversed by the EB1[EBP-2]::GFP comets. Taken together, these experiments suggest that although global microtubule turnover remains largely unaffected, specific aspects of microtubule plus end behavior in the anterior 1° dendrite are altered upon KNL-1 depletion.

**KNL-1 limits F-actin assembly in the PVD cell body and dendrites**
The actin cytoskeleton is a key driver of dendrite branching (Jan and Jan, 2010; Ouzounidis et al., 2023). During our morphological analysis using the membrane marker, we noticed that the cell body of the PVD neurons in KNL-1 degrader appeared enlarged and deformed (Fig. S4, A and B). Notably, similar deformations in cell bodies have been reported in response to disruptions in RhoGTPase and actin regulatory pathways in *C. elegans* (Demarco et al., 2012; Leeuwen et al., 1997; Zhao et al., 2022), suggesting potential defects associated with the actin cytoskeleton in KNL-1 degrader animals.

To visualize the actin cytoskeleton, we generated a strain expressing Lifeact, the actin-binding peptide of the yeast protein Abp140, fused to mKate2 under a PVD-specific promoter (Riedl et al., 2008). We imaged F-actin and the plasma membrane of the PVD neuron at the L4 stage when 4° dendritic branches are formed. In L4 control neurons, Lifeact::mKate2 signal is enriched in 3° and 4° dendrites relative to 1° dendrites and the cell body (Fig. 5 A), consistent with previous observations (Zou et al., 2018; Tang et al., 2019; Shi et al., 2021; Zhao et al., 2022) that F-actin is more abundant in newly formed dendrites. In KNL-1 degrader neurons, the mean Lifeact::mKate2 signal

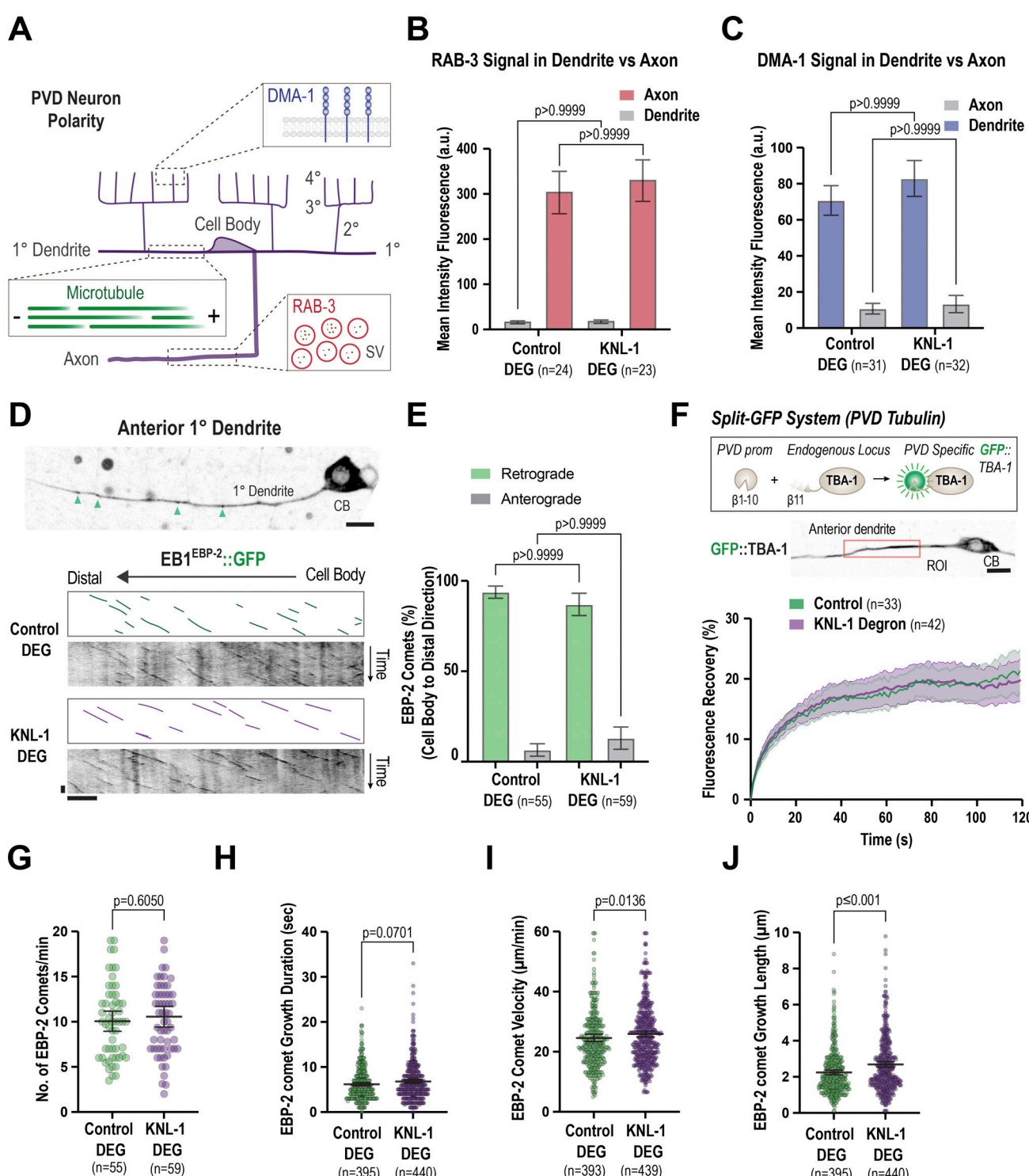

Figure 4. **PVD neuronal polarity and microtubule organization are not affected by postmitotic degradation of KNL-1. (A)** Schematic showing the intracellular organization of the PVD anterior dendrite and axon. DMA-1 is a dendrite-specific receptor and RAB-3 is found in the membranes of synaptic vesicles (SV) that accumulate in the axon of PVD. Microtubules in the primary anterior dendrite are organized with minus ends facing the dendrite tip and the plus ends point toward the cell body. **(B)** Quantification of RAB-3 fluorescence intensity in the dendrite and axon of the PVD neuron in the indicated conditions. *n* represents the number of animals. Error bars represent mean ± 95% CI. P values from one-way ANOVA test followed by Tukey's test. **(C)** Quantification of DMA-1 fluorescence intensity in the 4° dendrites and axons in indicated conditions. *n* represents number of animals. Error bars represent mean ± 95% CI. P values from Kruskal–Wallis test followed by Dunn's test. **(D)** EB1EBP-2::GFP dynamics in the anterior primary dendrite. Image of the PVD primary anterior dendrite at the L4 larval stage (top). The green arrowheads indicate EB1EBP-2 comets in the dendrite. Scale bar, 5 µm. Below are kymographs of EB1EBP-2::GFP along the dendrite in control and KNL-1 DEG animals (bottom). Scale bar, 10 s and 10 µm, respectively. **(E)** Quantification of EB1EBP-2::GFP comet direction in control and KNL-1 DEG. *n* represents number of animals. Error bars represent mean ± 95% CI. P values from two-way ANOVA test followed by Šídák's test. **(F)** Schematic of the split-GFP system used to express endogenous GFP::TBA-1 specifically in the PVD (top). The image below represents

GFP::TBA-1 in the anterior dendrite and the cell body of the PVD. The red rectangle denotes the region of 15 µm along the primary dendrite where photobleaching was performed. Scale bar 5 µm. The graph depicts the fluorescence recovery of GFP::TBA-1 in control and KNL-1 DEG. *n* represents the number of animals. Lines and shaded regions represent mean ± 95% CI, respectively. ROI, region of interest. CB, cell body. **(G–J)** Quantification of EB1^EBP-2^::GFP comet dynamics in control and KNL-1 DEG. *n* represents the number of animals (G) and the number of comets (H–J) respectively. Error bars represent mean ± 95% CI. P values from unpaired *t* test (G) or Mann–Whitney test (H–J).

intensity significantly increased around the periphery of the cell body, inside the cell body, and in proximal anterior 1° dendrites (Fig. 5, B–E), and closer examination revealed prominent F-actin filaments in the cell body (Fig. 5, B and D). Thus, degradation of KNL-1 during PVD development increases F-actin levels in the cell body and the cell body-proximal region of neurites.

Consistent with the role of KNL-1 in regulating F-actin dynamics, we also observed that KNL-1 degradation led to an increase in the number of dynamic membrane protrusions along

1° dendrites. The formation of 2° dendrite branches begins with the extension of actin-rich filopodia-like projections orthogonal to the 1° dendrite. These projections undergo extension and retraction until a subset stabilizes to form the 2° dendrites (Smith et al., 2010). Compared with control animals, KNL-1 degrader animals exhibited a significantly higher number of dynamic membrane protrusions from the 1° dendrites (Fig. 5, F and G). Since the extension of microtubules into filopodial extensions is essential for stabilizing and forming neurites in in vitro mammalian neuronal cultures (Dent et al., 2007), we investigated

Figure 5. **The PVD actin cytoskeleton is affected by KNL-1 degradation. (A)** Images showing actin distribution (gray at the top, magenta at the bottom) in the PVD cell body and dendrites (green). Actin is visualized using Lifeact::mKate2, and the plasma membrane is labeled with mNeonGreen(mNG)::PH. Scale bar, 10 µm. **(B)** Images highlighting actin distribution in the cell body (CB) and proximal anterior primary dendrite in control and KNL-1 DEG animals. Scale bar, 5 µm. **(C)** Box and whisker plot showing quantification of the mean Lifeact::mKate2 fluorescence intensity along a three-pixel wide line along the cell body (CB) periphery. *n* represents the number of animals. Whiskers represent minimum and maximum while the bar represents the first, second, and third quartiles. P values from Mann–Whitney test. **(D)** Box and whisker plot showing quantification of maximum Lifeact::mKate2 fluorescence intensity inside the cell body. *n* represents the number of animals. Whiskers represent minimum and maximum whilst bar represents first, second, and third quartiles. P values from unpaired *t* test. **(E)** Box and whisker plot showing quantification of mean Lifeact::mKate2 fluorescence intensity of a cross section of the 1° dendrite 10 µm anterior to the cell body along a three-pixel-wide line. *n* represents the number of animals. Whiskers represent minimum and maximum whilst bar represents first, second, and third quartiles. P values from Mann–Whitney test. **(F)** Images of the anterior primary dendrite labeled with mNG::PH in control and KNL-1 DEG, in late L3 larval stage animals. Orange arrowheads indicate filopodial like membrane protrusions. Scale bar, 5 µm. **(G)** Quantification of the number of membrane protrusions appearing from the anterior primary dendrite over a 3-h time interval. *n* represents the number of animals. Error bars represent mean ± 95% CI. P values from unpaired *t* test.

whether the elevated number of dendritic protrusions in KNL-1 degrader animals might be related to an increased presence of microtubules. We imaged GFP::TBA-1 in control and KNL-1 degrader animals but could not detect any GFP::TBA-1 signal within the membrane protrusions in either condition (Fig. 5 F; and Fig. S4, D and E). Although this suggests that these protrusions may predominantly lack microtubules, the possibility of transient or a low number of microtubules, undetectable under our current imaging conditions, cannot be excluded. Overall, these observations support the hypothesis that KNL-1 plays a role in regulating F-actin–related processes in the PVD neuron.

**Latrunculin A (Lat-A) treatment or co-depletion of CYK-1 formin, an actin assembly factor, alleviates dendrite branching abnormalities associated with KNL-1 loss**

Given the prominent role of KNL-1 in regulating higher-order dendrite branch number and menorah spacing in the PVD neuron, we sought to elucidate the underlying mechanism. We noticed that the mean intensity of the Lifeact::mKate2 signal in 4° dendrites was higher in KNL-1 degrader animals compared with the control, indicating increased F-actin levels in the higher order dendrites upon loss of KNL-1 (Fig. S4, F and G). Prior studies have implicated actin polymerization pathways in both dendrite branch outgrowth and contact-dependent retraction during dendritic self-avoidance (Liao et al., 2018; Sundararajan et al., 2019a; Hsu et al., 2020). This suggested that alterations to actin polymerization dynamics might be related to the higher-order branch abnormalities and self-avoidance defects associated with KNL-1 loss.

To explore this hypothesis, we treated both control and KNL-1 degrader animals with low doses of Lat-A, which inhibits actin filament assembly, for 2 h during the late L3-L4 developmental stages (Fig. S5 A). By this stage, 2° and 3° dendrites are already formed and the neuron starts to develop 4° dendrite branches, while 3° dendrites from adjacent menorahs undergo contact-dependent repulsion during dendritic self-avoidance. We assessed different Lat-A concentrations to find a regime that would perturb F-actin assembly without affecting the 2° and 4° dendrite branch numbers or the spacing between the menorahs in control animals (Fig. 6, A–C and Fig. S5 B). In KNL-1 degrader animals exposed to Lat-A, we found a significant reduction in quaternary dendrites and overlapping menorahs compared with those treated with DMSO (Fig. 6, A–C) as well as a decrease in prominent F-actin filaments in the cell body and a corresponding drop of the mean Lifeact::mKate2 signal around the cell body periphery (Fig. S5, C and D). These results suggest that KNL-1 modulates actin filament dynamics to regulate the number of quaternary branches and the spacing between the menorahs in the PVD neuron.

Actin filament assembly initiated by nucleators such as the Arp2/3 complex, which generate branched F-actin networks, and formins, which generate linear F-actin networks, contributes to dendrite branch patterning (Ouzounidis et al., 2023). We next investigated whether the effect of KNL-1 loss on PVD dendrite branching could be suppressed by perturbations of actin assembly factors such as Arp2/3 or formins. There are

seven formins in *C. elegans* (Mi-Mi et al., 2012). We focused on CYK-1, an ortholog of human mDia3, previously shown to be enriched at mitotic kinetochores of T98G glioblastoma cells, where it contributes to the stability of kinetochore-attached microtubules (Cheng et al., 2011). We engineered an AID fusion to ARX-2, a component of the Arp2/3 complex, and CYK-1, and performed co-depletions with KNL-1 to examine the effects on PVD dendrite arborization. We could not directly assess the effect of the Arp2/3 complex due to the essential role of Arp2/3 in initiating branches (Fig. 6, D and E; and Fig. S5, E and F). However, co-depletions of CYK-1 with KNL-1 ameliorated the increased quaternary branching and overlapping menorahs observed after KNL-1 degradation (Fig. 6, D–F). Notably, CYK-1 degradation alone did not result in PVD dendrite branching defects, nor did CYK-1 co-depletion rescue the increase in ectopic membrane protrusions from 1° dendrites observed after KNL-1 degradation (Fig. 6, D–F and Fig. S5 F). We also assessed whether co-depletions of CYK-1 and KNL-1 altered the higher levels of F-actin observed upon KNL-1 loss across the dendrite. Although co-depletions of CYK-1 and KNL-1 did not affect the elevated Lifeact::mKate2 signal in the cortical regions of the cell body, we observed that the mean intensity of Lifeact::mKate2 was significantly reduced in the anterior 1° and 4° dendrites when both CYK-1 and KNL-1 were absent, compared with KNL-1 degradation alone (Fig. S5, G–K). These results suggest that KNL-1 may play a role in counteracting actin assembly during dendrite branch growth. Overall, the above experiments are indicative of a role for KNL-1 in regulating linear F-actin assembly to promote proper arborization patterning in the PVD neuron.

**Ectopic plasma membrane targeting of KNL-1 induces F-actin assembly**

To explore the potential role of KNL-1 in regulating F-actin assembly during dendrite development, we designed a gain-of-function assay. It is well-established that Rho family GTPases, which are localized to the plasma membrane, modulate actin assembly to control dendrite branching (Scott and Luo, 2001). Our assay involved targeting KNL-1, C-terminally tagged with TagBFP, to the PVD plasma membrane using a myristoyl motif (MYR::KNL-1::TagBFP) (Fig. 7 A). Ectopic expression of MYR::KNL-1::TagBFP in PVD, but not MYR::TagBFP or KNL-1::TagBFP, resulted in the formation of actin-rich clusters around the cortical regions of the cell body (Fig. 7, B and C). These actin-rich clusters contained distinct foci of MYR::KNL-1::TagBFP surrounded by Lifeact::mKate2 and membrane (Fig. S6 A). Additionally, filopodia-like protrusions emerged from these actin clusters and displayed dynamic extension and retraction behavior (Fig. 7 D; and Videos 1 and 2). These findings suggest that plasma membrane–localized KNL-1 promotes F-actin assembly.

In mitosis, the N-terminus of KNL-1 acts as a signaling hub by serving as a docking site for protein phosphatase 1 (PP1) and the spindle assembly checkpoint proteins BUB-3/BUB-1, while the C-terminus of KNL-1 binds to the core microtubule coupler, the Ndc80 complex (Fig. 7 A) (Espeut et al., 2012). Our previous study identified that both the N- and C-termini of KNL-1 are essential for its neuronal function (Ouzounidis et al., 2024). To determine which half of KNL-1 is responsible for

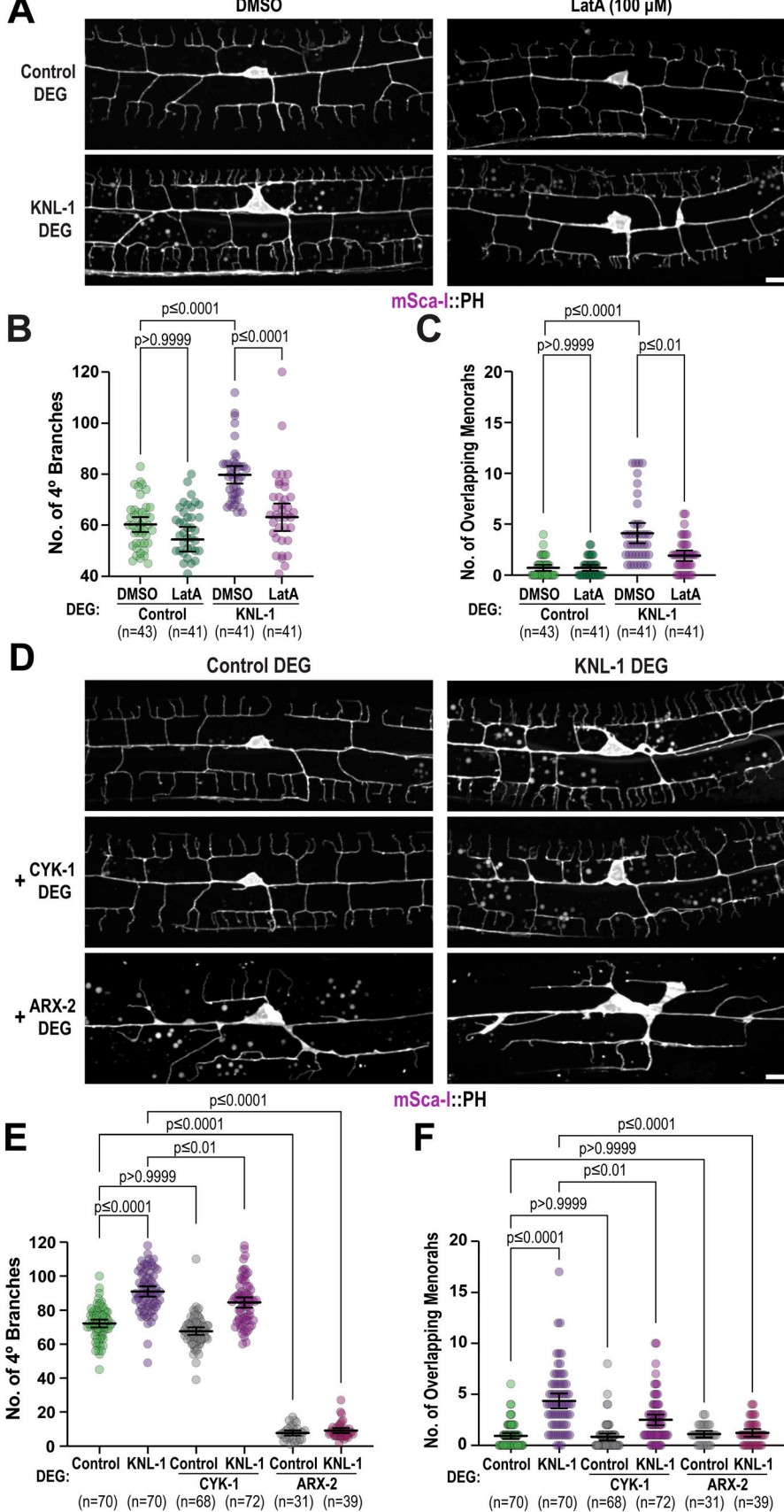

Figure 6. **Modulating actin dynamics alleviates dendrite branching abnormalities associated with KNL-1 loss. (A)** Images of PVD dendrite organization at the L4 larval stage in the control and KNL-1 DEG animals upon acute 2-h treatment with either DMSO or 100 µM Lat-A. These images show both the anterior and posterior dendrites extending from the cell body within a 100 µm region. Scale bar, 10 µm. **(B and C)** Quantification of the PVD dendrite architecture by determining the number of 4° branches and overlaps between menorahs for the indicated conditions. *n* represents number of animals. Error bars represent mean ± 95% CI. P values from one-way ANOVA test followed by Tukey's test (B) or Kruskal–Wallis test followed by Dunn's test (C). **(D)** Images of PVD dendrite organization at the L4 larval stage in the control and KNL-1 DEG animals in combination with CYK-1 DEG or ARX-2 DEG as indicated. These images show both the anterior and posterior dendrites extending from the cell body within a 100 µm region. Scale bar, 10 µm. **(E and F)** Quantification of PVD dendrite architecture including number of 4° branches and overlaps between menorahs in indicated conditions. *n* represents number of animals. Error bars represent mean ± 95% CI. P values from one-way ANOVA test followed by Tukey's test (E) or Kruskal–Wallis test followed by Dunn's test (F).

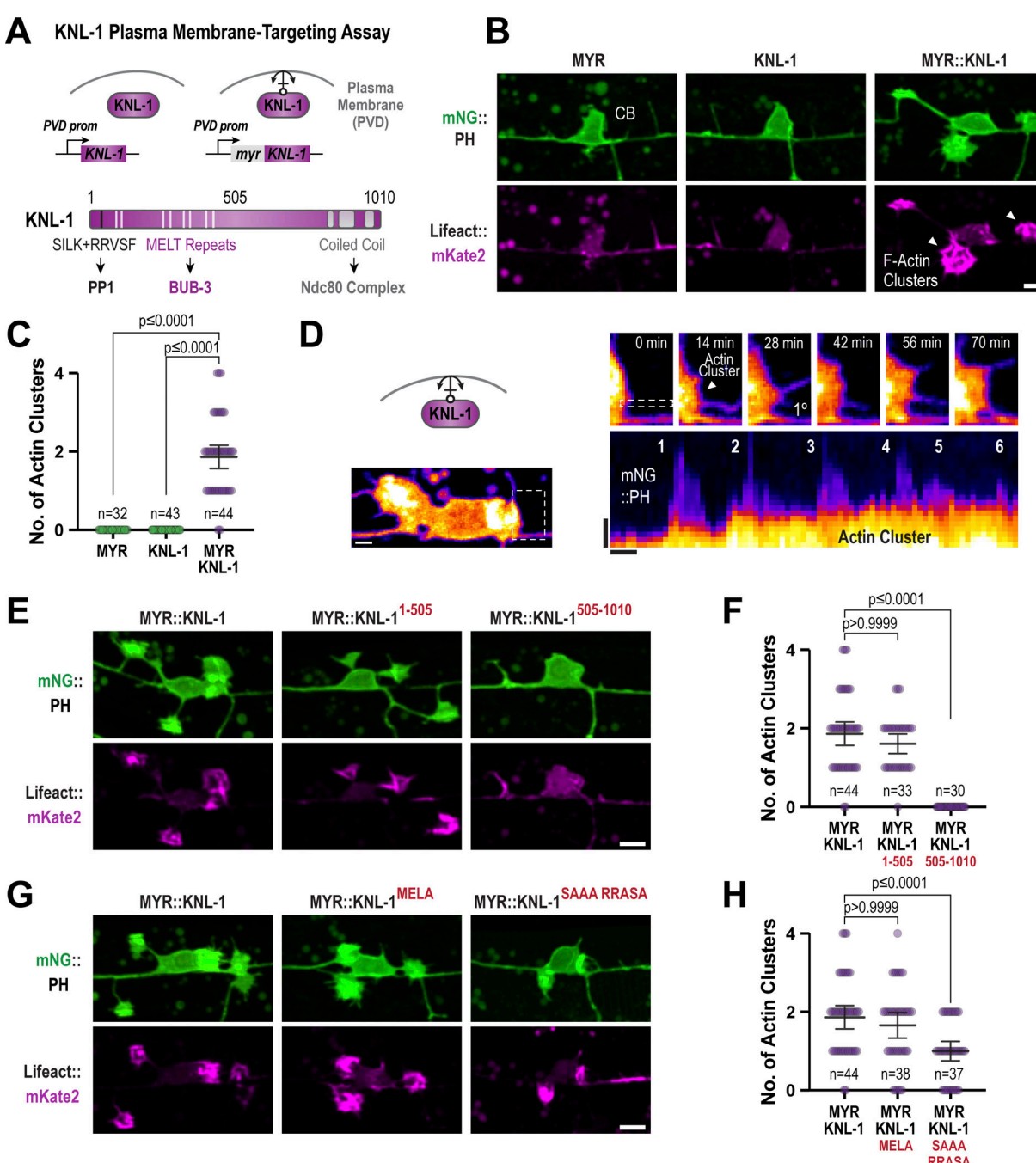

Figure 7. **Ectopic tethering of KNL-1 to the PVD plasma membrane results in the formation of actin clusters. (A)** Top: Cartoon of the approach employed for the ectopic tethering of KNL-1 to the plasma membrane of the PVD neuron using a myristoylation domain (MYR). Bottom: Schematic illustrating the functional domains within KNL-1. **(B)** Images showing the morphology of the PVD cell body and proximal primary dendrite at late L2 larvae stage (green) and actin (magenta) in MYR::TagBFP (MYR), KNL-1::TagBFP (KNL-1), and MYR::KNL-1::TagBFP (MYR::KNL-1). Actin is visualized using Lifeact::mKate2, and the plasma membrane is labeled with mNeonGreen(mNG)::PH. The white arrowheads indicate the actin clusters. Scale bar, 5 µm. **(C)** Quantification of the number of actin clusters in the PVD dendrites in the indicated conditions. n represents the number of animals. Error bars represent mean ± 95% CI. P values from Kruskal–Wallis test followed by Dunn's test. **(D)** Time-lapse imaging of actin cluster membrane dynamics in MYR::KNL-1 animals (top) Scale bar, 2 µm. Kymograph of mNG::PH at the membrane protrusion indicated by the dashed selection (bottom). Scale bar, 1 µm. and 5 min, respectively. **(E)** Images showing the morphology of PVD membrane at late L2 larvae stage (green) and actin (magenta) in animals expressing MYR:: KNL-1$^{1–505}$ or MYR::KNL-1$^{505–1010}$. Scale bar, 5 µm. **(F)** Quantification of the number of actin clusters in the PVD dendrites in indicated conditions. n represents number of animals. Error bars represent mean ± 95% CI. P values from Kruskal–Wallis test followed by Dunn's test. **(G)** Images showing morphology of PVD membrane at late L2 larvae stage (green) and actin (magenta) in animals expressing MYR::KNL-1$^{MELA}$ and MYR::KNL-1$^{SAAA+RRASA}$. Scale bar, 5 µm. **(H)** Quantification of the number of actin clusters in the PVD dendrites in indicated conditions. n represents number of animals. Error bars represent mean ± 95% CI. P values from Kruskal–Wallis test followed by Dunn's test.

F-actin assembly, we generated animals expressing MYR::KNL-1$^{1-505}$ or MYR::KNL-1$^{505-1,010}$. MYR::KNL-1$^{1-505}$ was sufficient to generate actin-rich clusters, suggesting that KNL-1's interaction with the microtubule-binding Ndc80 complex is dispensable for its F-actin assembly function (Fig. 7, E and F; and Fig. S6 B).

To elucidate the mechanism by which the N-terminus of KNL-1 promotes F-actin assembly, we expressed MYR::KNL-1$^{SAAA+RRASA}$, which disrupts PP1 recruitment to KNL-1, and MYR::KNL-1$^{MELA}$, in which mutation of the nine MELT repeats to ADAA diminishes BUB-3 binding (Fig. 7 A). MYR::KNL-1$^{MELA}$ generated a similar number of actin clusters as MYR::KNL-1, while MYR::KNL-1$^{SAAA+RRASA}$ showed a reduced number of actin clusters (Fig. 7, G and H). This suggests that KNL-1–mediated F-actin assembly involves the recruitment of PP1 activity.

## Discussion

Dendrite morphogenesis is a highly orchestrated process that requires precise regulation of the actin and microtubule cytoskeleton (Lefebvre, 2021). Various effectors of cytoskeletal remodeling in dendrites have been identified, but how effector functions are coordinated to generate the unique dendrite pattern of each neuron remains largely undefined. Taking advantage of the remarkable stereotypical morphology of the PVD neuron and using a PVD-specific protein-degradation system, we identified a role for KNL-1 and the associated KMN network, the microtubule-coupling machine that drives chromosome segregation during cell division, in dendrite arborization. Specifically, our work shows that KNL-1, the scaffold for microtubule and spindle assembly checkpoint machinery at the kinetochore, controls PVD dendrite branch organization and dynamics. Loss of KNL-1 led to an excess of 4° branches, overlapping menorahs, and an attenuation of contact-dependent retraction events between 3° dendrites during dendritic self-avoidance. This observation implies that KNL-1 activity, while dispensable for dendrite branch growth, contributes to restricting the terminal branch number and maintaining proper spacing between the branches. Both cell-autonomous and non-cell-autonomous mechanisms that regulate cellular processes such as membrane fusion and the cytoskeleton are involved in maintaining quaternary branch number and dendritic self-avoidance within the PVD dendrite (Oren-Suissa et al., 2010; Liao et al., 2018; Smith et al., 2012; Zou et al., 2018; Tang et al., 2019). In the future, it will be important to define how KNL-1 interacts with established intrinsic and extrinsic dendritic mechanisms to achieve the optimal number and spacing of PVD dendritic arbors.

Our analysis revealed that KNL-1 influences both microtubule and actin cytoskeletal dynamics within the PVD neuron. KNL-1 function appears to differ significantly from that of other known microtubule-associated proteins that affect PVD morphology, whose inhibition primarily affects microtubule orientation within the primary dendrite, or neuronal polarity (He et al., 2020, 2022; Taylor et al., 2015; Yan et al., 2013; Dey et al., 2024; Maniar et al., 2011). Loss of KNL-1 showed a modest but significant effect on microtubule plus-end growth in the 1° dendrites. Similar effects were observed in *D. melanogaster*

larval sensory neurons upon the knockdown of Ndc80 and Mis12 subunits, both components of the KMN network (Hertzler et al., 2020). The similarity in branching abnormalities in KNL-1 and Ndc80 complex depletions suggests that the interaction of KNL-1 with microtubules could be important to its function in the PVD. However, it remains unclear how alterations in microtubule plus-end growth properties affect 4° dendrite branching and dendritic self-avoidance, the two most predominant defects observed in neurons lacking KNL-1 and Ndc80 complex. The sparsity of microtubules in the 2° and 3° dendrite branches relative to the 1° dendrite makes it challenging to visualize the microtubule markers that measure microtubule turnover (GFP::TBA-1) and plus-end growth (EB1$^{EBP-2}$::GFP) in the higher-order branches (Fig. 4 D). It is likely that a more localized disruption of microtubule organization or dynamics may occur in 3° dendrite branches in the absence of KMN components, which we are unable to capture using our current imaging tools. Thus, defining the precise relationship between KMN microtubule regulation and dendrite branching will require the development of more refined molecular tools, such as EBP-2 and tubulin probes with increased photostability to study microtubule behavior in the higher-order branches, and the use of advanced imaging modalities.

An unexpected finding of our study is the effect of KNL-1 on actin filament dynamics to regulate the dendrite arborization of the PVD neuron. KNL-1 degradation led to elevated F-actin levels across the dendrite and an increase in filopodia-like membrane protrusions. Furthermore, increased quaternary branching and loss of dendritic self-avoidance in KNL-1 degrader animals were suppressed by altering actin polymerization using Lat-A and by co-depletion of the formin CYK-1, suggesting that KNL-1 plays a role in controlling actin filament dynamics. Knockdown of Mis12, another KMN component, in rat hippocampal neurons, also resulted in increased filopodial protrusions, supporting the idea that other KMN proteins might also affect the actin cytoskeleton (Zhao et al., 2019). While a direct connection between the KMN network and the actin cytoskeleton has not yet been described, several clues suggest a potential link between the actin machinery and kinetochores in the context of chromosome-microtubule attachment during cell division. Prior work has implicated the CYK-1 homolog mDia3 in the formation of kinetochore-microtubule attachments during mitosis (Cheng et al., 2011; Yasuda et al., 2004). Moreover, the actin cytoskeleton is crucial for the formation of kinetochore-microtubule fibers during mammalian oocyte meiosis, and F-actin cables generated by formin and spire, another nucleator of linear actin filaments, have been shown to interact with kinetochores during porcine oocyte meiosis (Harasimov et al., 2023; Mogessie and Schuh, 2017).

The observation that plasma membrane-targeted KNL-1 can induce actin assembly is intriguing, especially given that KMN network proteins were previously known only to interact with microtubules. The ability of KNL-1 N-terminus alone to induce the formation of actin-rich clusters at the plasma membrane suggests that KNL-1 has an intrinsic ability to regulate actin assembly and that this does not depend on the microtubule-binding Ndc80 complex. However, the increased actin levels

in dendrites after KNL-1 removal suggests a potential inhibitory role of KNL-1 on a specific actin assembly pathway. An important avenue for future investigation involves understanding the mechanism underlying KNL-1's F-actin assembly function and how it relates to dendrite branching mechanisms. Previous studies in the PVD neuron have shown that distinct modes of F-actin assembly dictate branch growth and retraction, tightly controlled by upstream signaling pathways (Sundararajan et al., 2019a; Liao et al., 2018). Furthermore, it has been proposed that the competition within cells for a limited supply of actin monomers could dictate which F-actin assembly pathway predominates (Kadzik et al., 2020). It is possible that KNL-1 facilitates the local buildup of a particular F-actin structure, and that removing KNL-1 allows for an alternative F-actin structure to assemble, which in turn could affect dendrite branch dynamics. Additional work targeting some of the actin assembly factors will be needed to determine the nature of the actin structures induced by the plasma membrane–targeted KNL-1.

Additionally, the reduced efficiency of plasma membrane targeted-KNL-1 protein phosphatase 1 docking mutants to generate actin clusters suggests that KNL-1 could influence upstream signaling pathways that control the actin cytoskeleton. Notably, PP1 function has been linked to neuronal cytoskeleton regulation, dendrite spine formation, and axon guidance mechanisms (Kastian et al., 2023; Monroe and Heathcote, 2013; Hofmann et al., 1998), but how it impacts actin cytoskeleton or dendrite branching is not known. Another possibility is that KNL-1 affects actin dynamics by modulating the activity of specific Rho GTPases which are known to mediate the assembly and disassembly of actin filaments in response to external cues during dendrite branching (Jan and Jan, 2010). Interestingly, Cdc42, a Rho GTPase family member found at centromeres, has been implicated in kinetochore-microtubule attachment and could be a potential candidate for further investigation (Lagana et al., 2010). Given that actin cytoskeletal dynamics are influenced by the microtubule cytoskeleton at structures such as focal adhesions (Seetharaman and Etienne-Manneville, 2019), it is plausible that KMN components may mediate crosstalk between microtubules and F-actin during dendrite branching.

One notable defect following KNL-1 degradation, the occurrence of overlapping menorahs, offers insights into the mechanistic role of KNL-1 during high-order branching. Non-overlapping arbors result from contact-dependent self-avoidance mediated by cell surface receptor-ligand interactions between dendrite branches of the same neuron (Lefebvre et al., 2015). Recent studies suggest that the formation of non-overlapping menorahs within the PVD neuron results from a balance of F-actin–dependent dendrite growth and retraction (Liao et al., 2018; Sundararajan et al., 2019a). Our time-lapse analysis shows that the ability of tertiary dendrites to retract upon contact is compromised in the absence of KNL-1. An emerging view is that a switch from a linear to a branched actin network underlies the transition from extension to retraction of the growing 3° dendrites (Liao et al., 2018; Sundararajan et al., 2019a). However, the precise molecular mechanism that triggers this transition is not known. Intriguingly, genetic evidence suggests that mutations in regulators of both linear

and branched actin polymerization impair contact-dependent retraction (Liao et al., 2018; Sundararajan et al., 2019a; Hsu et al., 2020). Our findings strongly indicate that KNL-1 plays a central role in contact-dependent retraction, and our co-depletion experiments with KNL-1 and CYK-1 support the idea that KNL-1 promotes self-avoidance via suppressing potentially linear actin filament formation. Future work is needed to identify how KNL-1 targets the specific actin machinery responsible for triggering retraction. Collectively, these observations suggest that the function of KNL-1 in PVD neurons is to promote the balance between linear and branched F-actin filaments required for proper dendrite arborization.

In addition, our work shows that beyond dendrite arborization, KNL-1 function is essential for proper sensory behavior. Our prior work shows a role for KNL-1 in the development of both dendrite and axonal structures (Cheerambathur et al., 2019; Ouzounidis et al., 2024). Local neuropeptide release from dendrites mediates proprioception, while axon-derived signals are required for harsh touch response (Tao et al., 2019). Although our analysis did not uncover any apparent axonal morphological defects in KNL-1 DEG animals, we cannot exclude the possibility that axonal defects or synaptic connectivity issues, together with dendritic morphology changes, contribute to the observed sensory behavior defects. Furthermore, KNL-1's activity is also required for the normal lifespan of the PVD neuron. Reduced dendrite arborization and dendrite degeneration are characteristic features of neuronal aging and neurodegenerative diseases, with cytoskeletal dysfunction implicated in these processes (Emoto et al., 2016). While the mechanisms connecting dendrite morphology to dendrite and neuronal lifespan remain unclear, our work highlights the importance of kinetochore protein's modulation of cytoskeletal dynamics to these important aspects of neuronal biology.

## Materials and methods

### *C. elegans* strains
*C. elegans* strains were maintained in nematode growth media (NGM) plates seeded with the *Escherichia coli* OP50 strain at 20°C. The genotypes of the *C. elegans* strains used in this study are described in Table S1.

### Recombinant plasmid DNA
All plasmids used in this study were made using the Gibson Assembly method (Gibson et al., 2009) and the DNA sequences were confirmed by Sanger sequencing.

### Generation of transgenic *C. elegans* strains
Single-copy transgenic integrations were engineered using the transposon-based Mos1-mediated Single Copy Insertion method (Frøkjaer-Jensen et al., 2008). The transgenes were cloned into pCFJ151 for insertion into chromosome I(*oxTi185*), II(*ttTi5605*), IV(*oxTi177*), or V(*oxTi365*) mos1 insertion sites. Information on the promoters used for the expression of the transgenes can be found in Table S2. A mix of repair plasmids that contains the transgene of interest, positive selection marker, transposase plasmid and three fluorescent markers for negative selection

(pCFJ90 [Pmyo-2::mCherry], pCFJ104 [Pmyo-3::mCherry], and pGH8 [Prab-3::mCherry]) was injected into appropriate *C. elegans* strains that contain the mos1 insertion sites. Transformants were selected based on their ability to rescue the mobility defect of the parental strains and successful integrants were identified by PCR genotyping.

The transgenic strains containing extrachromosomal arrays (*dhaEx1 and dhaEx3*) were generated by injecting plasmid construct mix containing either 50 ng/μl pDC1031or pDC1032, 2 ng/μl pCFJ90, and 50 ng/μl of empty pBluescript SK as carrier DNA into young adult hermaphrodites of the strain DKC410 that expressed the transgene P*des-2*::mScarlet-I::PH::*unc-54*::3′UTR from *oxTi177* mos1 locus on Chr IV.

### CRISPR/Cas9-mediated genome editing

Endogenous tagging of various genes (see Table S3) at the N- or C-terminus and generation of the *dma-1* deletion and *mec-4(e1611)* allele mimic that contains the A713V mutation were done using CRISPR/Cas9 methods (Dickinson et al., 2015; Paix et al., 2015). The specific method, guide RNA sequences, and homology arm sequences used to generate each strain are described in Table S3. Briefly, a DNA mix or Cas9 RNA mix, containing the respective repair template, guide RNA sequences, Cas9 and selection markers were injected into N2 animals, except for *dha185* allele, for which the DKC1139 strain was used. Recombinant strains were identified by appropriate selection method and by genotyping PCR and the sequences were confirmed using Sanger sequencing of the edited genomic region.

### Synchronization protocol for *C. elegans* to follow PVD morphogenesis

To track PVD morphogenesis, *C. elegans* animals were synchronized at the L1 arrest stage through the following procedure. Gravid adults were bleached on NGM plates using a solution containing 1 M sodium hydroxide and 2.5% sodium hypochlorite to extract eggs. The eggs were then incubated for 24 h on NGM plates without food at 20°C, allowing them to develop into L1 larvae and enter the L1 arrest stage. The arrested L1 animals were washed off the plates with M9 buffer, transferred to a centrifuge tube, and pelleted through centrifugation at 1,000 RPM for 1 min. The arrested L1 larvae were then placed onto NGM plates seeded with OP50 bacteria, designated as T = 0 h, and grown at 20°C. Under these conditions, the developmental stages following the transfer onto to NGM plate with food corresponded to: L2 (20 h), late L3 (36 h), and late L4 (48 h).

### Auxin degradation

The auxin treatment was conducted by growing animals on NGM plates supplemented with 4 mM auxin (K-NAA, N160; Phytochrome Laboratories), following the protocol described in Ashley et al. (2021). For efficient degradation of AID-tagged proteins, we expressed TIR1 under two distinct promoters within the PVD neurons: *unc-86*pro and *des-2*pro. To ensure targeted degradation of AID-tagged proteins specifically in PVD neurons after birth, synchronized animals were transferred to NGM plates containing auxin and OP50 *E. coli* at the late L2 stage (20 h after synchronization). In all experiments, except the

time-lapse imaging of PVD dendrite dynamics using mNeon-Green::PH membrane marker, TIR1 transgene expressing animals exposed to auxin were used as control. For time-lapse imaging experiments recording PVD dendrite dynamics, animals expressing both the AID fusion and TIR1 transgenes, but not exposed to auxin, were used as control.

### Behavioral assays

To conduct the locomotion assays, L4 stage worms were incubated for 1 h at 20°C on a freshly seeded NGM plate prepared the day before. Subsequently, the tracks left by the worms were recorded using a Nikon SMZ18 microscope equipped with a 2× P2-SHR Plan Apo 1× objective and a Photometrics Cool SNAP camera. For each worm, five tracks were analyzed, and this process was repeated for 15 animals per condition. The measurement of track characteristics included determining the track wavelength, which was defined as the distance between two successive peaks, and the track amplitude, which was calculated as half of the height between two opposing peaks. In total, five measurements were taken for each track to ensure accuracy and consistency in the analysis. Harsh touch response was assessed by prodding non-moving gravid day 1 adult *C. elegans* animals with a platinum wire to the midsection of the body, following previously established methods (Way and Chalfie, 1989). The animals were prodded using a long and flattened tip of a wire platinum pick either at or just behind the vulva region. Each individual animal was subjected to this prodding test three times, with 10-minute intervals between trials. Animals that backed up upon prodding were scored as sensitive to harsh touch. For each condition, 15 *C. elegans* animals were tested, and this process was repeated four times. All *C. elegans* animals used for this assay were in the *mec-4(e1611)* mutant mimic background which allowed for the specific isolation of the harsh touch response of the PVD neuron (Chatzigeorgiou et al., 2010). The harsh touch response analysis was carried out blind to genotypes.

### Lat-A drug treatment

The Lat-A treatment was performed on animals that were previously treated with auxin following the auxin degradation protocol (see above). The animals were treated in acute 100 μM Lat-A or DMSO solutions for 2 h in a humidity chamber at 20°C (36 h after synchronization). After acute treatment, the animals were transferred back to fresh NGM plates containing auxin and OP50 *E. coli*.

### Fluorescence microscopy

To image PVD neurons expressing the appropriate markers, L2, L3, or L4 stage, worms were anesthetized in 5 mM Levamisole and mounted in M9 on a 2% agarose pad. Images were acquired using a spinning disk confocal imaging system with a Yokogawa spinning disk unit (CSU-W1), a Nikon Ti2-E fully motorized inverted microscope, and a Photometrics Prime 95B camera at 20°C. In most experiments, a CFI60 60 × 1.4 NA HP Plan Apochromat lambda Oil (Nikon) objective was used to image the PVD neurons except when imaging Lifeact::mKate2 marker within the PVD neuron, where a CFI60 100 × 1.4 NA HP Plan

Apochromat lambda Oil (Nikon) objective was employed. Imaging acquisition was performed using NIS-Elements software. To visualize the cell body and the dendrite branches at various developmental stages, z-stacks of either 0.5 or 1.0 μm thickness were acquired from the PVD L/R neuron, whichever was closest to the side facing the coverslip/objective.

## Time-lapse imaging

In all experiments involving time-lapse imaging, samples were prepared, and images were acquired as described above. Specifically, for time-lapse imaging of PVD dendrite growth dynamics utilizing the mNeonGreen::PH membrane marker, synchronized L3 stage *C. elegans* animals were anesthetized in 5 mM Levamisole and mounted on 2% agarose pads. A region ~100 μm from the cell body along the anterior dendrite was chosen for this analysis. Time-lapse sequences were captured, comprising 9–11 z-planes with a spacing of 0.9 μm, at a rate of 1 frame per 2 min for 3 h at 20°C.

Acquisition for most time-lapses was performed using a spinning disc confocal imaging system with a Yokogawa spinning disk unit (CSU-W1), a Nikon Ti2-E fully motorized inverted microscope, with a CFI60 60 × 1.4 NA HP Plan Apochromat lambda Oil (Nikon) objective and Photometrics Prime 95B camera.

To image EB1[EBP-2]::GFP dynamics, synchronized L4 stage *C. elegans* animals were anaesthesized in 5 mM levamisole and mounted in M9 on 2% agarose pads. Movies comprising seven z-planes with a spacing of 0.5 μm within a 50 μm region adjacent to the cell body along the anterior dendrite were recorded. The movies were acquired at a rate of 1 frame per 1 s with a CFI60 100 × 1.4 NA HP Plan Apochromat lambda Oil (Nikon) objective and a spinning disc confocal imaging system as described above.

To image membrane dynamics in KNL-1 membrane tethering assays using the mNeonGreen::PH membrane marker, late L2 stage animals were anesthetized in 5 mM levamisole and mounted in M9 on 2% agarose pads. Time-lapse sequences were captured, comprising seven z-planes with a spacing of 0.75 μm at a rate of one frame per 1 min for 2 h.

## FRAP experiments

For the FRAP experiments involving GFP::TBA-1 and Lifeact::mKate2, *C. elegans* animals were anesthetized in 5 mM Levamisole and mounted in M9 on 2% agarose pads. FRAP was performed along the primary dendrite within a region adjacent to the cell body. Before photobleaching, a single z-slice of the anterior dendrite, 50 μm from the cell body was acquired at one frame per 1 s using the aforementioned spinning disc confocal imaging system with a CFI60 Plan Apochromat lambda 100× Oil (Nikon) objective. Following the acquisition of three prebleach frames, a 1.5 μm region, positioned 20 μm away from the cell body was subjected to photobleaching using a 405 nm laser. The fluorescence recovery after photobleaching was recorded at a rate of one frame per 1 s for a duration of 2 min at 20°C.

## Image analysis

All image analysis was performed using Image J (Fiji). In summary, the z-stacks were transformed into a maximum intensity projection, and the subsequent analysis was conducted on these maximum intensity projections to obtain the specified measurements as detailed below, unless stated otherwise. All the measurements were plotted using GraphPad Prism (GraphPad Software).

## Fluorescence intensity profile of GFP::KNL-1 along the dendrite

To visualize the distribution of GFP::KNL-1 along the dendrite, a 1-pixel-wide segmented line was drawn along the primary anterior dendrite using Image J (Fiji). For quantifying the colocalization of mScarlet-I with GFP::KNL-1 at PVD branch points, line scans were performed across the first menorah anterior to the cell body. For each animal, a two-pixel-wide segmented line was traced along the branch point of either in 1°–2° branch point or 2°–3° branch point, 2 μm on either side of the branch point. The maximum and minimum fluorescence intensity profiles along the segmented lines were recorded using Fiji's "Plot Profile" function for all wavelengths. The values were then normalized to the maximum and minimum intensity values, $\left(\frac{x - minimum}{maximum - minimum}\right)$.

## Quantification of GFP::KNL-1 puncta distribution in dendrite branch points

To quantify the GFP::KNL-1 positive branch points, only the menorahs within the first 50 μm anterior to the cell body were analyzed. For each animal, the number of branch points that had colocalized KNL-1 puncta in a 50 μm region anterior to the cell body was measured in reference to the total number of branch points within this region.

## Quantification of GFP::KNL-1 degradation

To quantify KNL-1 degradation within the cell body of the PVD neurons expressing GFP::KNL-1 and GFP::KNL-1::AID, the region corresponding to the cell body was defined by employing a polygon selection tool in Image J (Fiji) to outline the cell body boundary in mScalet-I::PH marker. The mean intensity of GFP::KNL-1::AID within the cell body was measured and corrected for background by subtracting the background fluorescence intensity, measured from a corresponding area outside the neuron.

## Quantification of PVD dendrite morphology parameters

Various parameters related to dendrite morphology were individually assessed for both posterior dendrite and the proximal anterior dendrite, located 100 μm from the soma, in *C. elegans* animals expressing the mNeonGreen::PH membrane marker. The secondary (2°) dendrites were identified as branches orthogonal to the primary (1°) dendrite that exhibited higher-order branching, including tertiary (3°) and/or quaternary (4°) branches. The 4° dendrites were recognized as orthogonal branches to the 3° dendrites. Overlapping menorahs were characterized as two adjacent 3° dendrites lacking a gap as described previously (Smith et al., 2012). Membrane protrusions were defined as structures extending from the 1° dendrite with a length of at least twice their width and devoid of higher-order branches. To determine the cell body area, a region was defined along the perimeter of the cell body using the polygon selection

tool in Image J (Fiji), and the area under the selected region was then quantified. While quantifying PVD dendrite morphology parameters for strains with extrachromosomal arrays expressing GFP and KNL1::GFP, only animals with normalized GFP intensity values within the cell body that exceeded half of the maximum recorded intensity were considered for analysis. To establish this threshold, we initially measured the maximum fluorescent intensities of GFP or KNL-1::GFP within the region demarcated by a polygon selection outlining the cell body boundary in Image J (Fiji). The values were corrected for background by subtracting the background fluorescence intensity, measured from a corresponding area outside the neuron.

### Quantification of PVD dendrite dynamics

The duration of contact between 3° dendrites, number of contacts, and number of unresolved contacts were quantified by analyzing MIPs of z-stacks obtained from the time-lapse movies using Image J(Fiji). These measurements were carried out within a 100 μm region along the PVD neuron, positioned anterior to the cell body. Contacts between 3° dendrites were defined as instances where two adjacent 3° dendrites were in direct contact without any visible gap, following the criterion described previously (Smith et al., 2010). Only the contacts that occurred during the time-lapse acquisition were included in the quantification. Unresolved contacts were defined as contacts between 3° dendrites that remained without a gap by the end of the time-lapse. Additionally, for Fig. 5, F and G; and Fig. S4 C, filopodial protrusions were defined as any membrane extensions originating from the primary dendrite that were <2.5 μm in length.

### Quantification of membrane dynamics in KNL-1 tethering assays

To assess membrane dynamics, kymographs were generated using KymographClear 2.0a and KymographDirect ImageJ plugins (Mangeol et al., 2016) from a segmented line with a width of three pixel and 4 μm at the membrane of an actin cluster.

### Analysis of membrane blebbing

To quantify membrane blebbing, we examined the entire anterior dendrite by analyzing maximum intensity projections of *C. elegans* animals expressing the mScarlet-I::PH membrane marker at the L4 stage. Bead-like structures observed on the membrane were categorized as "blebbing events" based on an analysis of dendrite width. Specifically, we measured the dendrite width, compared any variations in width, and used an empirically determined threshold of 1.5 times the width of the flanking membrane as a criterion for defining beads or blebbing events.

### Counting of mScarlet-I::LGG-1 puncta

mScarlet-I::LGG-1 was found in the dendrites, axon initial segment, and cell body, but for our analysis, we only considered the mScarlet-I::LGG-1 puncta along the dendrites. To quantify the number of mScarlet-I::LGG-1 puncta, we applied a threshold arbitrarily (set at >10× the local diffuse background signal) to the fluorescence intensity profile plot along the dendrites of day 3 adult animals.

### Analysis of mKate2::RAB-3 and DMA-1::mKate2 fluorescence intensity

mKate2::RAB-3 is primarily concentrated in the axon of the PVD neuron; however, the mKate2 signal was present throughout the neuron, effectively outlining the morphology of the PVD neuron. To evaluate the distribution of mKate2::RAB-3, we quantified the mean fluorescence intensity of mKate2::RAB-3 puncta along both the length of the axon of each neuron and a corresponding length in the proximal anterior dendrite. To do this, we initially drew a segmented line with a width of three pixels along the axon and a matching line with similar width and length along the primary dendrite. Subsequently, we recorded the mean fluorescence intensity along these segmented lines using Image J (Fiji). To account for background fluorescence, a parallel segmented line was traced outside the neuron to obtain a background intensity measurement, and this value was then subtracted from the mean fluorescence intensity.

For DMA-1::mKate2, which is enriched in the quaternary dendrites, we quantified the mean fluorescence intensity of the puncta along in 4° dendrites of the first anterior menorah adjacent to the cell body using membrane counter marker (mNeonGreen::PH). Specifically, a three-pixel-wide segmented line was traced along the 4° dendrites of that menorah, and the mean fluorescence intensity was measured and averaged for each menorah. The measured fluorescence intensity values were corrected for background by subtracting the background fluorescence intensity, measured along a corresponding segmented line outside the neuron. Similarly, the mean fluorescence intensity of the puncta was quantified along the axon.

### EBP-2 dynamics analysis

To quantify EBP-2 dynamics, kymographs were generated using KymographClear 2.0a ImageJ plugin (Mangeol et al., 2016) from a segmented line with a width of 2 pixel and 50 μm anterior to the cell body. From these kymographs, comet velocity, density, growth length, and growth duration were quantified.

### FRAP analysis

To measure fluorescence intensity after photobleaching for GFP::TBA-1, a region of interest with a diameter of 1.5 μm was chosen using the elliptical selection tool in Image J (Fiji). At each time point, we measured the fluorescence intensity within this elliptical region and then adjusted it by subtracting the background fluorescence intensity obtained from a comparably sized region located outside the neuron. Additionally, to account for bleaching during image acquisition, we performed a correction by dividing the value of the bleached area by the value of the unbleached area, compensating for this loss.

### Quantification of tubulin in the filopodia protrusions

To quantify the tubulin distribution in the filopodia protrusions segmented lines of two pixels were drawn across all the filopodia-like protrusions (<2.5 μm length) within 50 μm anterior of the cell body using a plasma membrane marker (mNeon-Green::PH) to trace it. The mean GFP::TBA-1 fluorescence intensity along the filopodia was measured and corrected for background by subtracting the background fluorescence

intensity, measured along a corresponding segmented line outside the neuron.

### Quantification of actin in the PVD neuron

To evaluate the distribution of actin around and within the PVD neuron cell body, Lifeact::mKate2 z-stacks composed of ~18 × 1 µm planes were projected as a maximum intensity projection. To determine the circumference of the cell body, a segmented line of 3-pixel width was drawn around the cell body, starting at the anterior side, and tracing along the cell body boundary in a clockwise direction using Image J (Fiji). A corresponding line of the same width was placed outside the neuron to calculate the background fluorescence intensity. The mean Lifeact::mKate2 fluorescence intensity along the cell body circumference was measured and corrected for background by subtracting the background fluorescence intensity, measured from a corresponding area outside the neuron.

To assess the distribution of actin filaments within the cell body, the region corresponding to the cell body was outlined using a polygon selection tool in Image J (Fiji). Within this defined area, the standard deviation, maximum, and minimum intensity of Lifeact::mKate2 were determined. The values were corrected for background by subtracting the background fluorescence intensity, measured from a corresponding area outside the neuron.

To quantify the actin distribution in the 1° dendrite, a segmented line of 2 pixels wide was drawn across the dendrite 15 µm anterior to the cell body. The mean Lifeact::mKate2 fluorescence intensity along at the 1° dendrite was measured and normalized to the background.

To quantify the actin distribution in the 4° dendrite, segmented lines two pixels wide were drawn across all the 4° dendrites of the first dorsal anterior menorah using a plasma membrane marker (mNeonGreen::PH) to trace it. The mean Lifeact::mKate2 fluorescence intensity along at the 4° dendrites was measured, and the average intensity for each menorah was calculated and corrected for background by subtracting the background fluorescence intensity, measured from a corresponding area outside the neuron.

### Quantification of actin clusters in KNL-1 membrane tethering assays in the PVD neuron

To quantify the membrane protrusions with enriched actin in the PVD (referred to as actin clusters), in the KNL-1 membrane tethering mutants, superimposed membrane marker (mNeonGreen::PH) was used. These clusters were considered if they were localized outside the cell body and had an width twice as large as the width of the 1° dendrite.

### Quantification of KNL-1:TagBFP transgene expression in the cell body of the PVD neuron

To quantify the expression of KNL-1::TagBFP, a z-stack composed of ~18 × 1 µm planes was projected as a maximum intensity projection in the KNL-1::TagBFP channel and a membrane counter marker (mNeonGreen::PH). The cell body was traced using the membrane marker and polygon selection tool in Image J (Fiji). The mean KNL-1::TagBFP fluorescence intensity was measured and normalized to maximum and minimum fluorescence intensities in an area 100 × 100 µm surrounding the cell body, $\left(\frac{x - minimum}{maximum - minimum}\right)$.

### Quantification and statistical analysis

Details of the methods employed to extract and quantify various parameters in microscopy datasets are described in the image analysis section. The statistical tests used to determine significance are described in the figure legends. The data normality was assessed using a Shapiro–Wilk test. For normally distributed data either an unpaired $t$ test (for comparisons between two groups), an ordinary one-way ANOVA with a follow up Tukey's multiple comparison test (for comparisons between three and more groups), or a two-way ANOVA with a follow-up Šídák's multiple comparison test (for comparisons with two or more independent variables) were performed. For data sets that did not pass the normality test either a Mann–Whitney test (for comparisons between two groups) or a Kruskal–Wallis test with a follow-up Dunn's multiple comparison test (for comparisons between three and more groups) were performed. All statistical comparisons were done in GraphPad Prism (GraphPad Software).

### Online supplemental material

Fig. S1 shows results related to Fig. 1 (description of GFP::KNL-1 localization during L3 stages). Fig. S2 shows results related to Fig. 2 (effect of degradation of Ndc80 complex and Mis12 complex components on PVD architecture). Fig. S3 shows results related to Fig. 4 (DMA-1 localization and microtubule organization). Fig. S4 shows results related to Fig. 5 (F-actin distribution and filopodia dynamics). Fig. S5 shows results related to Fig. 6 (effects of Latrunculin and co-depletion of actin nucleators in suppression KNL-1 branching deffects). Fig. S6 shows results related to Fig. 7 (KNL-1 expression in membrane tethering mutants). Videos 1 and 2 are related to Fig. 7. Time-lapse imaging of the cell body of PVD expressing MYR::TagBFP and MYR::KNL-1::TagBFP. Table S1 lists *C. elegans* genotypes used in this study. Table S2 lists promoter sequences used for transgene expression. Table S3 lists CRISPR/Cas9 method, single guide RNAs (sgRNAs), and sequences targeted for each allele used in this study.

### Data availability

Data are available in the primary article and the supplementary materials. Original data, *C. elegans* strains, and plasmids generated in this study are available upon request from the corresponding author.

## Acknowledgments

We thank Reto Gassmann, Bill Earnshaw, Esther Zanin, and Rebecca Green for their valuable feedback on the manuscript. We thank Frank McNally, Department of Molecular and Cellular Biology, University of California, Davis, Davis, CA, USA and Martin Harterink, Department of Biology, Utrecht University, Utrecht, The Netherlands for sharing *C. elegans* strains, Ana Carvalho, Instituto de Investigacao e Inovacao em Saude (i3S),

Universidade do Porto, Porto, Portugal for the Lifeact::mKate2 plasmid, and Dave Kelly and Toni McHugh for microscopy and image analysis support.

Some strains were provided by the Caenorhabditis Genetics Center (CGC), which is funded by the National Institutes of Health Office of Research Infrastructure Programs (P40 OD010440). This work was supported by a Sir Henry Dale Fellowship to D.K. Cheerambathur jointly funded by the Wellcome Trust and the Royal Society (208833/Z/17/Z), a Sir Henry Wellcome Postdoctoral Fellowship (215925) awarded to B. Prevo and core funding for the Wellcome Centre for Cell Biology (203149), and the Discovery Research Platform for Hidden Cell Biology (226791). H. Alves Domingos and V.R. Ouzounidis were supported by the studentships from the Darwin Trust, Edinburgh. Open Access funding provided by University of Edinburgh.

Author contributions: H. Alves Domingos: Conceptualization, Data curation, Formal analysis, Investigation, Methodology, Project administration, Validation, Visualization, Writing - original draft, Writing - review & editing, M. Green: Conceptualization, Data curation, Formal analysis, Investigation, Methodology, Validation, Visualization, Writing - original draft, Writing - review & editing, V.R. Ouzounidis: Conceptualization, Data curation, Formal analysis, Investigation, Methodology, Resources, Supervision, Visualization, C. Finlayson: Investigation, Resources, B. Prevo: Funding acquisition, Investigation, Methodology, Resources, Visualization, Writing - original draft, Writing - review & editing, D.K. Cheerambathur: Conceptualization, Data curation, Formal analysis, Funding acquisition, Investigation, Methodology, Project administration, Resources, Software, Supervision, Validation, Visualization, Writing - original draft, Writing - review & editing.

Disclosures: The authors declare no competing interests exist.

Submitted: 23 November 2023

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

# Supplemental material

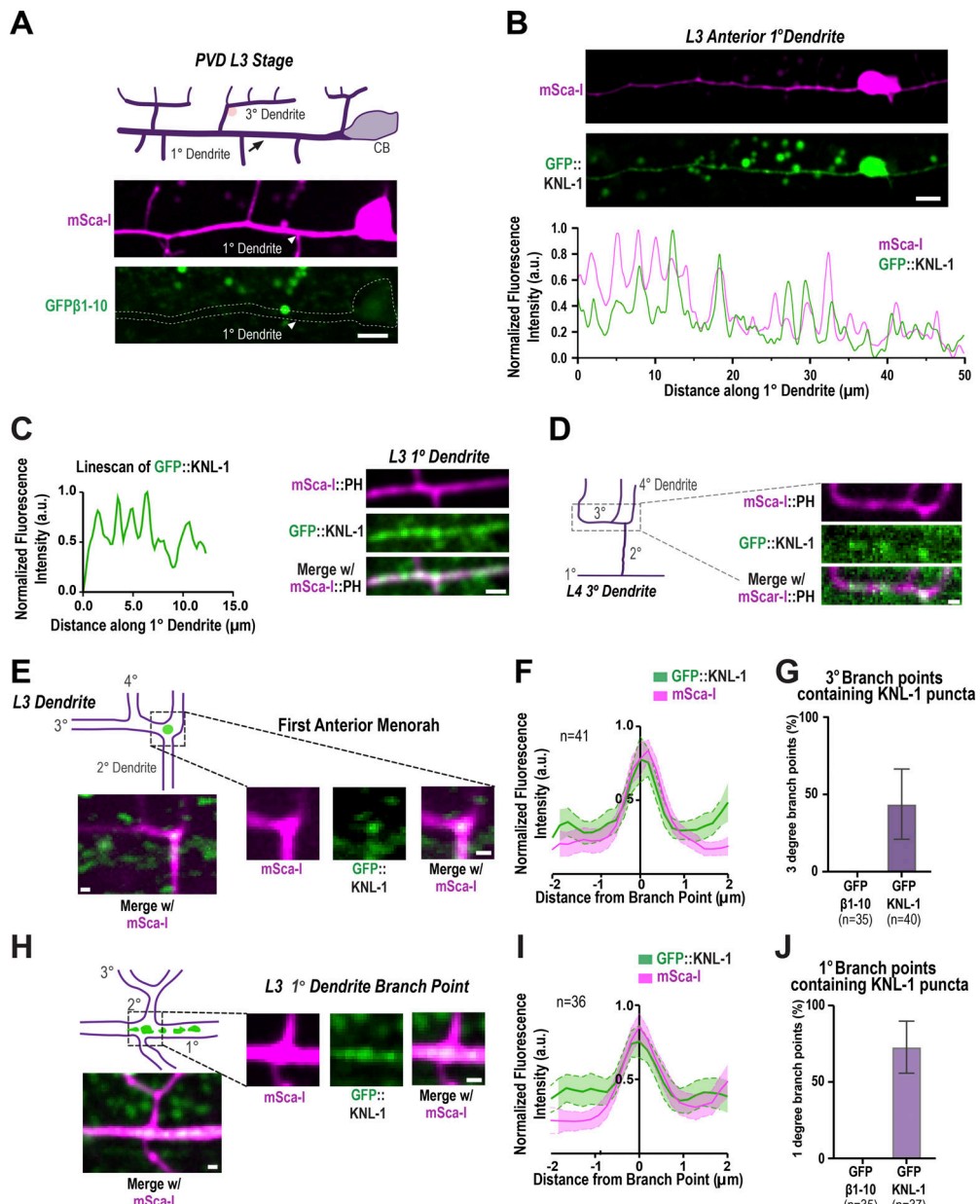

Figure S1.    **Related to Fig. 1. PVD morphogenesis and KNL-1 localization. (A)** Images show GFP1-10β control across anterior dendrite at the L3 stage. White arrow highlights 1° dendrite, which shows no detectable signal for GFP β1-10 in the absence of the complementing β11 strand. Pink circle highlights an autofluorescence granule. The images are maximum intensity projections of four z-planes spaced 0.5 µm apart. Scale bar, 5 µm. **(B)** Images (top) and linescan of the fluorescence intensity (bottom) showing GFP::KNL-1 and mScarlet-I (mSca-I) cytoplasmic probe distribution along the primary dendrite 50 µm anterior to the cell body at the L3 stage. The images are maximum intensity projections of three z-planes spaced 0.5 µm apart. Scale bar, 10 µm. **(C)** Image showing GFP::KNL-1 puncta distribution along 1° dendrite at the L3 stage (right). Scale bar, 1 µm. Graph showing linescan of the fluorescence intensity of GFP::KNL-1 (left). **(D)** Image shows the distribution of GFP::KNL-1 along the 3° dendrite at the early L4 stage. Scale bar, 2 µm. **(E)** Left: Schematic (top) and images (bottom) shows the localization of GFP::KNL-1 in the first menorah anterior to the cell body. Right: Insets of GFP::KNL-1 puncta (green) and mSca-I (magenta) at the 2° and 3° branch points. The images are maximum intensity projections of two z-planes spaced 0.5 µm apart. Scale bar, 1 µm. Only a subset of the branch point contains GFP::KNL-1 signal. **(F)** Quantification of GFP::KNL-1 and the mSca-I signal at the intersections of 2° and 3° in 41 animals including the ones without GFP::KNL-1. Dotted lines and shaded region represent mean ± SEM. Branch points show an increased intensity of both the cytoplasmic mSca-I and GFP::KNL-1 signal. **(G)** Quantification of the distribution of GFP::KNL-1 puncta at the branch points between 2° and 3° compared with the GFP β1-10, which was used as a negative control. GFP::KNL-1 positive branch points were analyzed only within the menorahs 50 µm anterior to the cell body. n represents the number of animals. Error bars indicate mean ± SD. **(H)** Left: Schematic (top) and images (bottom) show the localization of GFP::KNL-1 in the first menorah anterior to the cell body. Right: Insets of GFP::KNL-1 puncta (green) and mSca-I (magenta) at the 1° and 2° branch point. The images are maximum intensity projections of two z-planes spaced 0.5 µm apart. Scale bar, 1 µm. Only a subset of the branch point contains GFP::KNL-1 signal. **(I)** Quantification of GFP::KNL-1 and the mSca-I signal at the intersections of 1° and 2° in 36 animals including the ones without GFP::KNL-1. Dotted lines and shaded region represent mean ± SEM. Branch points show an increased intensity of both the cytoplasmic mSca-I and GFP::KNL-1 signal. **(J)** Quantification of the distribution of GFP::KNL-1 puncta at the branch points between 1° and 2° compared to the GFP β1-10 which was used as a negative control. GFP::KNL-1 positive branch points were analyzed only within the menorahs 50 µm anterior to the cell body. n represents the number of animals. Error bars indicate mean ± SD.

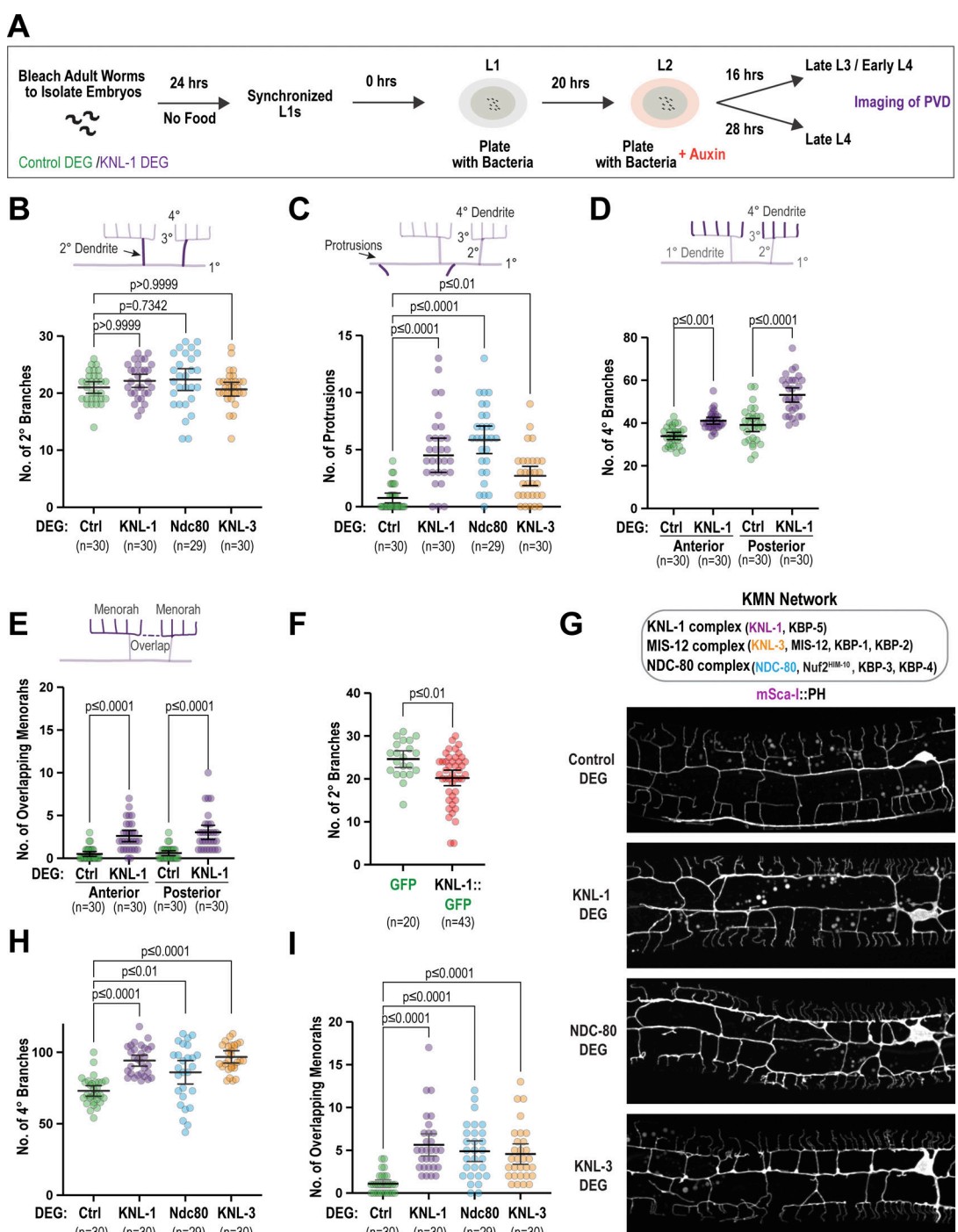

Figure S2. **Related to** Fig. 2. **Effects of degradation of KMN network on PVD morphology. (A)** Schematic indicating the experimental design for degrading AID::KNL-1. To obtain synchronized L1 animals, embryos from control or AID::KNL-1 adult worms expressing PVD-specific TIR1 transgene were isolated by treating the worms with bleach and hatched for 24 h on plates without food. The arrested L1 larvae were placed on a plate with food until they reached the L2 stage (20 h). The late L2 stage animals were transferred to plates with food and auxin and grown until they reached L3 (16 h) or L4 stage (28 h). The late L4 stage animals, which have fully developed menorahs, were imaged to analyze the PVD morphology. **(B and C)** Quantification of the number of 2° branches and the number of protrusions for the indicated conditions. n represents the number of animals. Error bars represent mean ± 95% CI. P values from Kruskal–Wallis test followed by Dunn's test. **(D and E)** Quantification of PVD dendrite architecture including number of 4° branches and overlaps between menorahs in control and KNL-1 DEG in anterior and posterior dendrites. n represents number of animals. Error bars represent mean ± 95% CI. P values from one-way ANOVA test followed by Tukey's test (D) or Kruskal–Wallis test followed by Dunn's test (E). **(F)** Quantification of the number of 2° branches in the PVD in GFP and KNL-1:: GFP overexpressing animals. n represents the number animals. Error bars represent mean ± 95% CI. P values from Mann–Whitney test. **(G)** Schematic of the KMN network complexes (KNL1/Mis12/Ndc80) and their components (top). Images of PVD dendrite organization in control, KNL-1 DEG, NDC-80 DEG & KNL-3 DEG animals with mScarlet-I(mSca-I)::PH membrane marker (bottom). Scale bar, 10 μm. **(H and I)** Quantification of the number of 4° branches and overlaps between menorahs for the indicated conditions. n represents the number of animals. Error bars represent mean ± 95% CI. P values from Kruskal–Wallis test followed by Dunn's test.

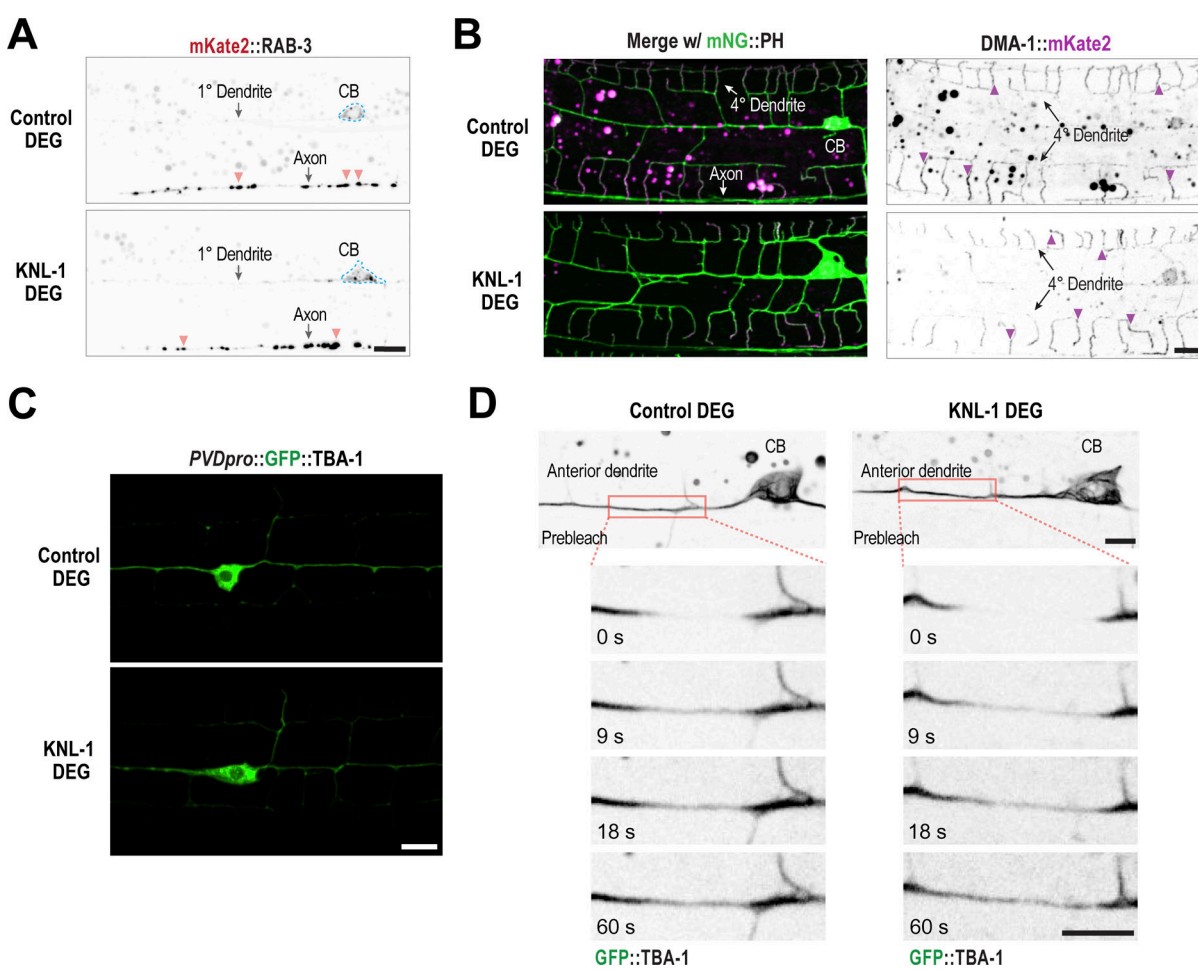

Figure S3.    **Related to** Fig. 4**. DMA-1 distribution, microtubule organization, and stability after degradation of KNL-1 in the PVD. (A)** Localization of the synaptic marker RAB-3 in control and KNL-1 DEG. Orange arrowheads indicate example RAB-3 puncta. RAB-3 puncta are predominantly enriched in the axon in both conditions. Scale bar, 5 µm. CB, cell body. **(B)** Localization of the dendrite guidance receptor, DMA-1 in control and KNL-1 DEG at the L4 larval stage. (Left) Merged images of DMA-1::mKate2 with mNeonGreen(mNG)::PH membrane marker and (Right) DMA-1::mKate2 for the indicated conditions. DMA-1 is restricted to dendrite structures and no axonal signal for DMA-1 is visible. Purple arrowheads indicate example DMA-1 puncta. Scale bar, 5 µm. **(C)** Images display expression of GFP::TBA-1 driven by *des-2* promoter in control and KNL-1 DEG animals. GFP::TBA-1 is predominantly localized to the 1° dendrite relative to the 2° and 3° dendrite branches. Scale bar, 10 µm. **(D)** Still images of the PVD anterior primary dendrite expressing endogenous GFP::TBA-1 in control and KNL-1 DEG animals (top). Scale bar, 5 µm. The dashed rectangles indicate regions that were photobleached in the FRAP experiments. Pre-bleach and post-bleach images of GFP::TBA-1 in control and KNL-1 DEG animals (bottom) Scale bar, 5 µm.

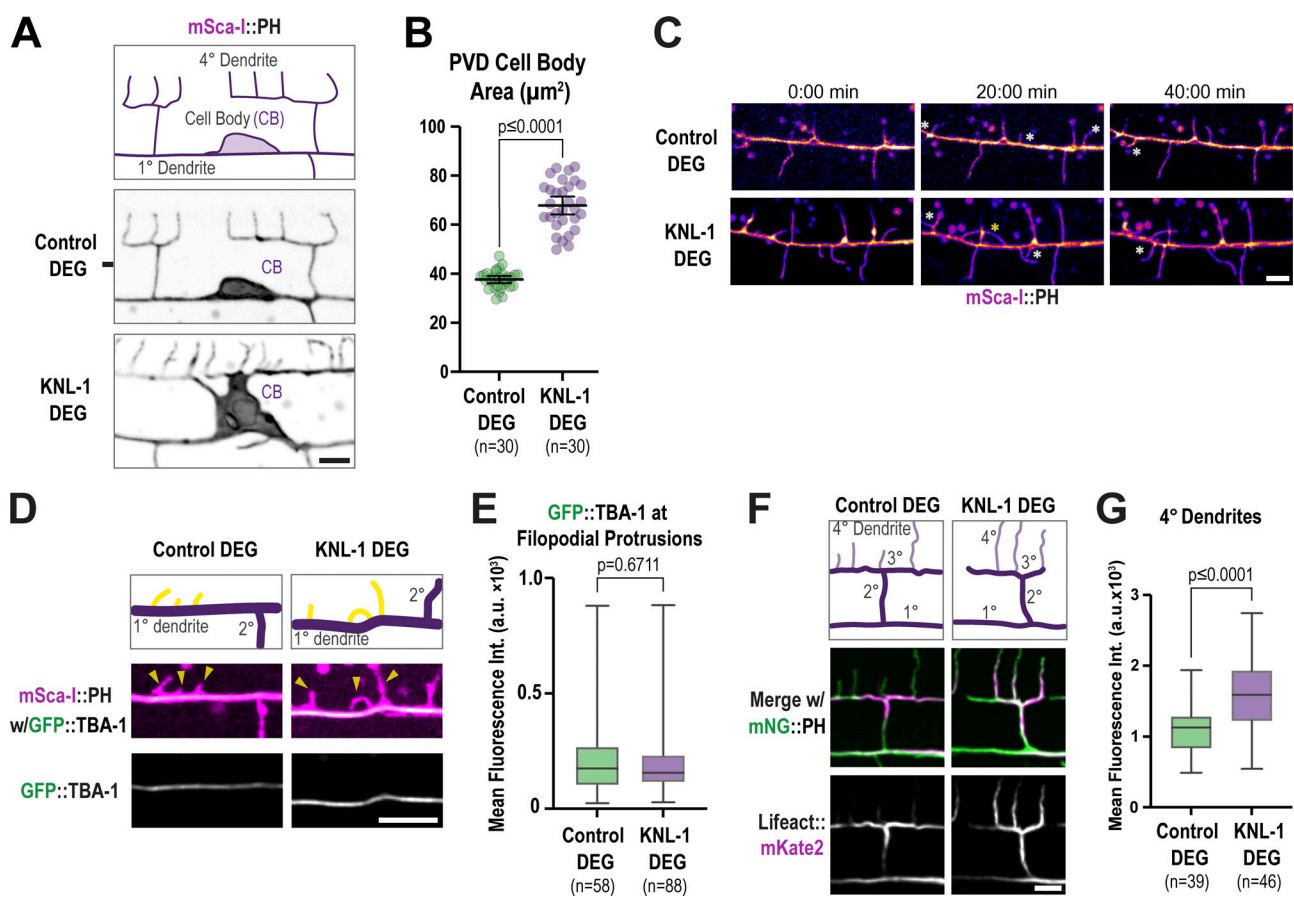

Figure S4. **Related to Fig. 5. Analysis of membrane and F-actin dynamics in control and KNL-1 degraded animals. (A)** Images of the PVD cell body (CB) and proximal regions in control and KNL-1 DEG labelled using mScarlet-I(mSca-I)::PH membrane marker at the L4 stage. Scale bar, 5 µm. **(B)** Quantification of the area of the cell body in control and KNL-1 DEG animals. *n* represents number of animals. Error bars represent mean ± 95% CI. P values from Mann–Whitney test. **(C)** Snapshots of time-lapse imaging of 1° dendrite branch dynamics using mNeonGreen(mNG)::PH membrane marker in control and KNL-1 DEG animals at the late L3 stage. Yellow arrowheads indicate already formed protrusions. White asterisks indicate newly extending protrusions. Yellow asterisks indicate the misguided protrusions that run parallel to the 1° dendrite. Scale bar, 5 µm. **(D)** Images of the membrane of the primary dendrite and the microtubule cytoskeleton labeled using mSca-I::PH membrane marker and endogenous GFP::TBA-1, respectively, in control and KNL-1 DEG animals at the L3 stage. Yellow arrowheads highlight membrane protrusions. Scale bar, 5 µm. **(E)** Box and whisker plot showing quantification of the mean intensity of GFP::TBA-1 within the membrane protrusions along the primary dendrite 50 µm anterior to the cell body. *n* represents the number of animals. Whiskers represent minimum and maximum while bar represents first, second, and third quartiles. P values from Mann–Whitney test. **(F)** Schematic and images of the anterior dorsal menorah proximal to the cell body showing the actin distribution at the 4° dendrites in control and KNL-1 DEG animals. Actin and membrane are labeled using Lifeact::mKate2 and mNG::PH, respectively. Scale bar, 5 µm. **(G)** Box and whisker plot showing quantification of Lifeact::mKate2 fluorescence intensity along a two-pixel wide line along 4°dendrites of the first anterior menorah. *n* represents the number of animals. Whiskers represent minimum and maximum whilst bar represents first, second, and third quartiles. P values from unpaired *t* test.

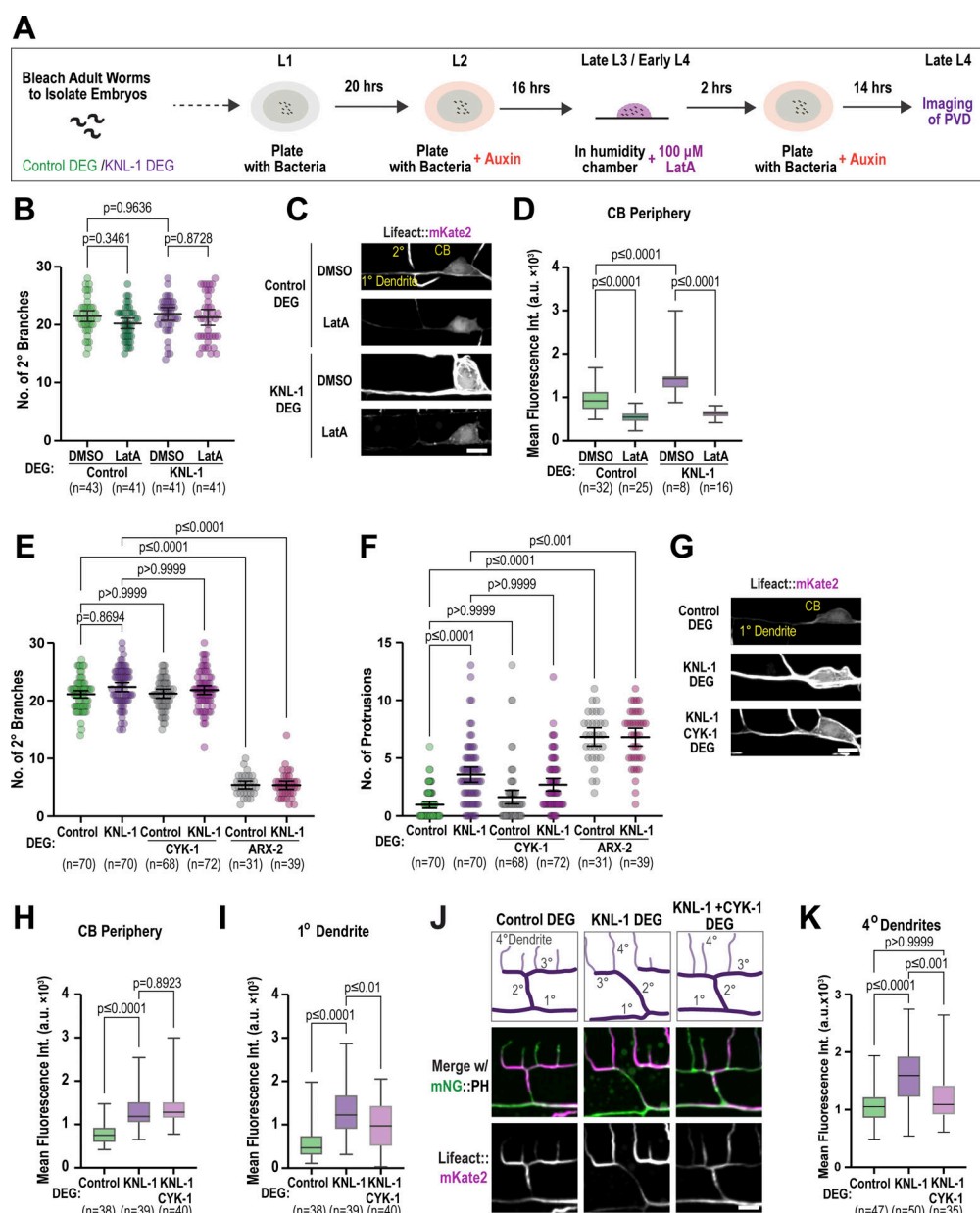

Figure S5.   **Related to Fig. 6. Effect of Lat-A and actin nucleators on PVD branching and F-actin distribution. (A)** Schematic showing the experimental design of Lat-A treatment. To obtain synchronized L1 animals and for auxin treatment the protocol in Fig. S2 A was followed. Once the animals reached late L3 stage (16 h after auxin), they were exposed to 100 µM of Lat-A or DMSO in a humidity chamber for 2 h. The animals were then transferred to plates with food and auxin and grown until they reached late L4 stage. **(B)** Quantification of number of 2° branches for the indicated conditions. n represents number of animals. Error bars represent mean ± 95% CI. P values from one-way ANOVA test followed by Tukey's test. **(C)** Images showing actin (Lifeact::mKate2) in the cell body (CB) and proximal anterior primary dendrite in indicated conditions. Scale bar, 5 µm. **(D)** Box and whisker plot showing quantification of Lifeact::mKate2 fluorescence intensity along a three-pixel-wide line along the cell body (CB) periphery in control and KNL-1 DEG when treated with DMSO or 100 µM Lat-A. n represents the number of animals. Whiskers represent minimum and maximum whilst bar represents first, second, and third quartiles. P values from one-way ANOVA test followed by Tukey's test. **(E and F)** Quantification of the number of 2° branches and protrusions for the indicated conditions. n represents the number of animals. Error bars represent mean ± 95% CI. P values from Kruskal–Wallis test followed by Dunn's test. **(G)** Images highlighting actin distribution in the cell body (CB) and proximal anterior primary dendrite for the indicated conditions. Scale bar, 5 µm. **(H)** Box and whisker plot showing quantification of Lifeact::mKate2 fluorescence intensity along a three-pixel wide line along the circumference of the cell body (CB) in control, KNL-1 DEG, and KNL-1 CYK-1 DEG. n represents the number of animals. Whiskers represent minimum and maximum whilst bar represents first, second, and third quartiles. P values from Kruskal–Wallis test followed by Dunn's test. **(I)** Box and whisker plot showing quantification of Lifeact::mKate2 fluorescence intensity of a cross section of the 1° dendrite 10 µm anterior of the cell body along a three-pixel-wide line in indicated conditions. Whiskers represent minimum and maximum while bar represents first, second, and third quartiles. P values from one-way ANOVA test followed by Tukey's test. **(J)** Schematic and images of the anterior dorsal menorah proximal to the cell body, showing the actin distribution at the 4° dendrites for the indicated conditions. Actin and membrane are labeled using Lifeact::mKate2 and mNeonGreen(mNG)::PH, respectively. Scale bar, 5 µm. **(K)** Box and whisker plot showing quantification of the Lifeact::mKate2 fluorescence intensity along a two-pixel-wide line positioned along 4° dendrites of the first anterior menorah. n represents the number of animals. Whiskers represent minimum and maximum whilst bar represents first, second, and third quartiles. P values from Kruskal–Wallis test followed by Dunn's test.

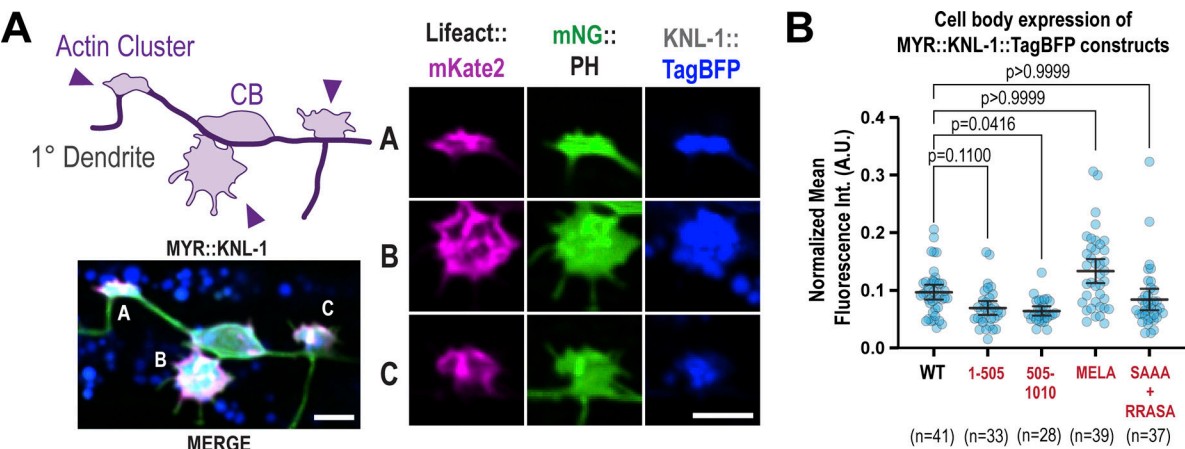

Figure S6. **Related to** Fig. 7. **Localization of MYR::KNL-1 and expression of MYR::KNL-1 variants in the PVD. (A)** Schematic with arrowheads highlighting the actin clusters (left above) and image of MYR::KNL-1::TagBFP induced actin clusters (left below). Insets on the right show images of Lifeact::mKate2 (magenta), the plasma membrane labeled with mNeonGreen(mNG)::PH (green) and MYR::KNL-1::TagBFP (blue) in three different actin clusters. Scale bar, 5 μm. **(B)** Quantification of KNL-1::TagBFP fluorescence intensity in the cell body normalized to maximum and minimum intensities in indicated conditions. *n* represents the number of animals. Error bars represent mean ± 95% CI. P values from Kruskal–Wallis test followed by Dunn's test.

Video 1. **Time-lapse sequence of a control (MYR::Tag::BFP) PVD neuron expressing mNeonGreen::PH membrane marker.** Each movie frame is a maximum intensity projection of a 5 × 0.75 μm z-stack acquired on a spinning disk confocal microscope. Playback speed is 300× real-time. Total duration 1 h. Size scaled up six times compared with the original.

Video 2. **Time-lapse sequence of a control (VMYR::KNL-1::Tag::BFP) PVD neuron expressing mNeonGreen::PH membrane marker.** Each movie frame is a maximum intensity projection of a 5 × 0.75 μm z-stack acquired on a spinning disk confocal microscope. Playback speed is 300× real-time. Total duration 1 h. Size scaled up six times compared with the original.

**Provided online are Table S1, Table S2, and Table S3. Table S1 lists** *C. elegans* **strains. Table S2 lists promoter sequences used for transgene expression. Table S3 lists CRISPR/Cas9 method, sgRNAs, and sequences targeted for each allele used in this study.**

