## [Peer Review File · The Journal of Cell Biology]

The Kinetochore Protein KNL-1 Regulates the Actin Cytoskeleton to Control Dendrite Branching

Henrique Alves Domingos, Mattie Green, Vasileios Ouzounidis, Cameron Finlayson, Bram Prevo, and Dhanya Cheerambathur

Corresponding Author(s): Dhanya Cheerambathur, University of Edinburgh

Review Timeline:

Submission Date:	2023-11-23
Editorial Decision:	2024-01-10
Revision Received:	2024-07-23
Editorial Decision:	2024-08-25
Revision Received:	2024-11-03
Editorial Decision:	2024-11-07
Revision Received:	2024-11-11

Monitoring Editor: Louis Reichardt

Scientific Editor: Dan Simon

Transaction Report:

DOI: <https://doi.org/10.1083/jcb.202311147>

January 10, 2024

Re: JCB manuscript #202311147

Dr. Dhanya K Cheerambathur
University of Edinburgh
Wellcome Centre for Cell Biology
School of Biological Sciences
Kings Buildings
Edinburgh EH9 3BF
United Kingdom

Dear Dr. Cheerambathur,

Thank you for submitting your manuscript entitled "The Kinetochore Protein KNL-1 Regulates the Actin Cytoskeleton to Control Dendrite Branching" to Journal of Cell Biology. I am attaching to this letter the evaluations by three reviewers.

Each reviewer shares an assessment that the phenotypes described in the manuscript are very intriguing. None of the three reviewers, however, believe that your mechanistic efforts have resulted in definitive results demonstrating that the KMN complex is acting directly on the actin cytoskeleton and not primarily through microtubules. The reviewers each provide a list of questions and comments that I hope will be helpful for you and your lab members. As interesting as your phenotypes are, this journal simply cannot accept a manuscript that leaves three reviewers not convinced that your major mechanistic conclusion is definitively demonstrated. Consequently, while we would welcome a future manuscript that established clearly the mechanisms through which the KMN complex is controlling dendrite morphologies, we must reject this manuscript.

I am sorry that I do not have better news to send to you and hope that this decision does not discourage your lab members. Of course, this decision does not imply any lack of interest in your work and we look forward to future submissions from your lab.

Thank you for your interest in Journal of Cell Biology.

Sincerely yours,

Louis F. Reichardt, PhD
Monitoring Editor
Journal of Cell Biology

Dan Simon, PhD
Scientific Editor
Journal of Cell Biology

Reviewer #1 (Comments to the Authors (Required)):

The manuscript of Green et al is a very interesting description of phenotypes that arise when components of the KMN complex (particularly KNL-1) are degraded in a much-studied sensory neuron in *C. elegans*. It represents a significant advance over a previous study in *C. elegans* (by Dr. Cheerambathur) that first described a post-mitotic role for kinetochore proteins in *C. elegans* neurons. That paper built a very strong case for a legitimate post-mitotic role in neurodevelopment, but could not offer any detailed or mechanistic analysis of why the dendrites were abnormal in the particular sensory neurons that studied. By switching to a different sensory neuron and using cell-specific degradation of KNL-1, they were able to dig more deeply and found several interesting features, including incorrect branch spacing, increased ectopic protrusions from the primary dendrite, changes in the distribution and stability of actin, especially close to the cell body. The work is overall well conducted and with adequate attention to statistics and rigor, but there are a number of places that require more attention, as detailed below, and some unnecessarily sweeping statements in the abstract and discussion regarding the primacy of an actin phenotype that are ill-substantiated. Though it is a very good paper - the main point, that KMN complex is acting directly on the actin cytoskeleton and not primarily on microtubules is supported only partially and indirectly. No direct interaction is demonstrated and this is in contrast to prior work in *Drosophila* dendrites that pointed to microtubule interactions.

1. The signal from the ScarletPH domain is not adequately explained - is there more membrane at the branchpoint simply because it is a branchpoint and therefore has more surface to volume, or does it represent some internal clustering of membranes - which is what the images actually suggest?

2. The line scans along dendrites are not adequately explained in text or methods or figure legends. Each dendrite in each animal has multiple menorahs with 2o and 3o branches. Were all of them scanned in each animal or only a select few that had detectable scarlet and GFP signals? What exactly then does n=14 animals mean? If there has been some selection for visible GFP or scarlet at the branchpoint, then the conclusion is less robust. Figure S1 also undercuts the argument that there is a special relationship of KNL-1 and branchpoints. There are many more KNL-1 hotspots along the primary dendrite than there are axon branches in B and S1C shows two branches - one with and one without a KNL-1 hotspot, and the cartoon similarly indicates hotspots all over and not necessarily correlated with a branch. Bottom line is that I'm not persuaded the hotspots are enriched at branchpoints to a statistically significant extent.
3. In Fig. 2, please define better the distinction between a 2o branch and an ectopic projection from the primary neuron. Are they only called 2o if they continue to develop 3o branches? Is that a meaningful distinction?
4. When branches appear to overlap or make contact, do they really make contact and is this really a failure of repulsion? In the *Drosophila* sensory dendrite literature there are several phenotypes that are mistakenly interpreted as a failure of self-avoidance when in fact they are misguidance phenotypes where the dendrites fail to stay in the proper layer of epidermis and therefore are out of plane with one another and only appear to overlap. See Kim et al 2012 PMC3470655 . In the present case, are the dendrite branches staying at the boundary of the epidermis and muscle cells or wandering out of that plane?
5. Does the change in filipodial number in Figure 4 represent increased initiation of the outgrowth or increased stabilization? Images were recorded at 1 frame every 20 min, which makes it impossible to distinguish these mechanisms. Initial outgrowth might indeed suggest an actin-dependent mechanism, but increased stabilization could well be due to a change in microtubule behavior. They do not report whether these filipodia contain a detectable microtubule.
6. Figure 5H (y axis): Punctae is not a word. Puncta is the plural of punctum. (very minor point)
7. The Discussion is interesting and nicely reviews what indications there are in the literature that kinetochores and actin filaments may at times interact. They do consider (wisely IMHO) in the middle of p16 that the changes in actin distribution may be entwined with or secondary to changes in microtubules, In a few places, however, it makes rather sweeping claims that are not adequately supported by the experiments.
 1. That the failure of repulsion/avoidance explains all the dendrite morphogenesis phenotypes. Even setting aside the concern mentioned above, contact repulsion is not likely to explain all the additional filipodia forming off the primary dendrite, nor the increase in the number of 4o branches.
 2. That loss of KNL-1 affects the actin cytoskeleton and not the microtubule cytoskeleton. There are clearly changes in the abundance and stability of actin, but the evidence that there is no change in microtubules is much weaker. They only measure EB-3 comets in the primary dendrite (which does not have a strong phenotype), don't measure parameters such as lifetime of growing tip or invasion of filipodia) and ignore their previous conclusion that the microtubule binding site of Ndc80 is necessary for proper dendrite development in a different set of sensory neurons.
 3. They cite Hertzler et al. incorrectly as saying that microtubule growth dynamics were not altered in fly sensory neurons. In fact, Hertzler finds increased growing + ends in those dendrites; that article explicitly studies increased microtubule dynamics. This neuron in *C. elegans* may have a different use of the kinetochore, but you can't gloss over the difference by an inaccurate citation.
 4. They state the microtubules are unaltered in 3o and 4o dendrites in Figure S3, but I can't see those microtubules at all, there is no quantification of them and no information on their dynamics.

Reviewer #2 (Comments to the Authors (Required)):

Summary

In this manuscript, Green et al use the highly branched *C. elegans* PVD neuron to examine the cellular basis of dendritic patterning. Specifically, they examine terminal branching, self-avoidance, and maintenance regulation by the kinetochore protein KNL-1 and interactors. Prior studies of KNL-1 in worms and kinetochore proteins in flies and mammals support a wider role for kinetochores in neural development. The current study expands these findings to the dendritic arborization of PVD neurons. First, KNL-1 inhibits terminal branches, and small branch initiation from the primary dendrite, thereby controlling the PVD dendrite pattern. Second, KNL-1 degraded neurons show specific deficits in self-avoidance. Third, worms show defective proprioception and nociception. Fourth, dendrites show hallmarks of degeneration and potentially premature aging. As a link between chromosomes and spindle microtubules during cell division, the KMN network, which includes KNL-1, might be expected to regulate neuronal microtubule dynamics during development. However, the authors found no evidence to support this hypothesis. Rather, KNL-1 degradation appears to limit F-actin assembly and dynamics, especially in the cell body and proximal dendrites, potentially accounting for some of the dendrite phenotypes observed. While the authors describe a number of interesting new contexts for kinetochore protein function in neural development, their analysis does not yet offer satisfying new insight into processes underlying the observed phenotypes and thus the role of KNL-1 in normal development.

Major points

1. Kinetochore proteins have known roles in neural development in worms, flies, and mammals. This paper will spawn many follow up studies in this system that will provide new insights into their roles, but the authors take a survey approach, and this paper does not go into depth to reveal the basis for any of the phenotypes that are described. For example, does KNL-1

interface with known regulators of self-avoidance or branch fusion such as the UNC-6 pathway, FMI-1, MIG-14, or EFF-1? This analysis could be especially interesting and insightful given the apparent similarity of the phenotypes in tertiary dendrites. The proposed premature ageing phenotype is not further explored except to look at the abundance of a molecular correlate of ageing in dendrites. Altogether the superficial analysis of various phenotypes reduced enthusiasm for this manuscript in its current form.

2. The overall conclusion that phenotypes are not due to effects of microtubule dynamics is not sufficiently supported. The authors show that there are no obvious defects in microtubule trafficking or polarity along the anterior primary PVD dendrite. However, other than the ectopic branchlets along the primary arbor, KLN-1 phenotypes are observed in terminal parts of arbors, which were not studied in detail. Imaging analysis was also performed in later-stage arbors, which is presumably after morphological phenotypes would first develop. Lastly, it was difficult to assess the distribution of TBA-1::GFP in dendrites in Figure S3 and arguments were entirely qualitative. As a result, the argument that phenotypes upon KLN-1 degradation are not due to effects on microtubules was not convincing.

3. Although the observations are careful and most phenotypes are quantified, my general take-away was that KNL-1 is required for many different aspects of PVD dendrite morphology. This is an important starting point, but I still don't know how or why these phenotypes emerge in any of the cases.

Minor points

4. The resolution of the images in Figure 1B is poor, and it is therefore hard to visualize the differences between the red and green channels.

5. Different parts of the arbor are examined in certain experiments. Is there any regional specificity to the dendrite phenotypes along the A-P axis of the arbor?

6. DMA-1 expression analysis is not convincing or maybe the image is not representative because DMA-1 signal looks different in control and KNL-1 DEG. If the authors measure mean intensity along the 4th order dendrites, they could be missing aspects of distribution. Perhaps a line-scan of signal intensity along the arbor could be informative.

7. The authors show that KNL-1 DEG causes behavioral phenotypes and argue that these are linked to actin misregulation and dendrite arborization phenotypes. Have the authors eliminated possible effects on axon arborization or terminal connectivity, or synaptic transmission independent of dendrites phenotypes? In Figure 1B it looks like KNL-1 is present at least in the proximal axon. Without this determination, conclusions about the functional effects of arbor defects seem more a matter of the paper's focused perspective on dendrites and should be qualified by stating other possible causes.

8. The authors argue that loss of KNL-1 leads to premature ageing due to the presence of beading and accumulation of autophagosomes. It seems that the phenotype could just be indicative of poor neuron health without invoking age-related explanations.

Reviewer #3 (Comments to the Authors (Required)):

In "The kinetochore protein KNL-1 regulates the actin cytoskeleton to control dendrite branching" the authors build on their previous work showing that KMN network proteins function post-mitotically in neurons in *C. elegans*. Previously they had only observed changes in neurons that use a cilium as a dendrite. They now extend this work to show that neurons with branched dendrites (PVD) have structural and functional deficits. The portions of the manuscript that describe the PVD phenotype from degradation and overexpression of KNL-1 are convincing. In Figure 2 structural changes in PVD are shown and in Figure 5 behaviors known to involve PVD are altered in animals in which KNL-1 is reduced in PVD. However, the mechanistic parts of the story are not convincing; problems with assays and interpretations are detailed below.

Fig 1- Complementation of endogenous KNL-1 tagged with GFP11 by cell-type specific expression of GFP 1-10 in PVD is used to show KNL-1 presence and localization in post-mitotic PVD neurons. In some contexts GFP1-10 can have weak fluorescence on its own so this data would be strengthened by including control images of GFP1-10 expressed in PVD without KNL-1-GFP11. This tagging strategy is also used to conclude KNL-1 localizes at branch points. However this finding is not demonstrated convincingly. It would be helpful to have overviews that show the localization of KNL-1 at different stages. Also, what percent of branch points have KNL-1 puncta? Is this higher than chance based on spacing of puncta along the dendrite? Also, the average increase in fluorescent intensity of KNL-1 at branch points is not any different than the membrane marker (Figure 1D) although in the images shown the membrane marker looks fairly evenly distributed while the KNL-1 is not.

Fig 2- The gain and loss of function effects on quaternary branching are very convincing and satisfyingly opposite. For the last point in the figure- the additional fusion between dendrites, does this always happen when there is extra terminal growth? Or is it a specific change in this background. It would be helpful to know whether other overgrowth genotypes have

similar extra fusion.

Fig 3 and S3-

One of the major conclusions highlighted in the abstract is that KNL-1 does not act in PVD via the microtubule cytoskeleton. The data supporting this conclusion is presented in Figure 3 and S3. As the microtubule-binding domain of Ndc80 was shown to be important in ciliated neurons in *C. elegans*, and many kinetochore proteins were shown to regulate microtubule dynamics in *Drosophila* dendrites, this is surprising.

Previously it was shown that reduction of KMN network proteins increases the number of microtubule plus ends per unit length in *Drosophila* neurons. It looks as though this might also be the case here (Fig S3E) as the three highest values are in the KNL-1 DEG background. However only 11 cells were assayed, so more data is needed to see if the distribution is different or to make a strong conclusion that it is not. In the EB1 kymographs shown (3D) it looks like length of time growing may be slightly longer in the KNL-1 DEG scenario. It always requires a very high bar to show that something is not affected, especially when previous studies have implicated these proteins in microtubule regulation so more data on number of growing microtubules and other aspects of microtubule behavior seems warranted.

It is unclear what FRAP of tubulin represents in 3F and S3- is this just diffusion of free tubulin back into the region since the overall percent recovery is low? If so, this parameter would be unrelated to microtubule behavior but would only report on size of free tubulin as diffusion would largely be controlled by size. This assay is said to represent microtubule stability, but it seems much more likely that on a 60s time scale only diffusion of free tubulin is being assessed rather than anything about stability of polymerized microtubules.

The pattern of DMA-1 in the control and KNL-1 DEG neurons shown in Fig S3A actually looks quite different although it is described as the same. It seems very enriched near tips of control dendrites and much more evenly distributed in the KNL-1 DEG neurons.

Overall the data presented in both Figure 3 and S3 is problematic, in some cases an conclusions are drawn that are not supported by the type of assay used (Figure 3D) and in other cases the data presented does not seem like it supports the conclusion that control and KNL1- DEG neurons are similar (Figure S3A and S3E).

Fig 4

One of the central points of the paper is that KNL-1 regulates the actin cytoskeleton. Support for this conclusion derives from the use of the Lifeact reporter shown in Figure 4. The distribution of the reporter appears quite different in the control and KNL-1 DEG neurons, but in both cases seems to label only a subset of actin as many of the terminal actin-based branches do not contain the reporter. This is consistent with previous reports indicating that only a subset of actin structures are labeled with Lifeact. Much of the data in this figure deals with changes in the cell body that are difficult to relate to dendrite structure. It is also unclear whether these changes (for example cell body size) are related to expression of the Lifeact reporter as they are not mentioned earlier in the paper when different visualization methods are used (cell bodies shown in fig 2A are very similar in both backgrounds).

Lifeact is referred to as a way to monitor F-actin, but it also binds G-actin so will label the soluble portion as well. Lifeact has been reported to dissociate from actin too rapidly to be used in FRAP assays to monitor actin dynamics (<https://www.tandfonline.com/doi/full/10.1080/19490992.2014.1047714>), so it is unlikely that the data shown in H and I is related to turnover of F-actin as described in the results.

Overall, the data on actin dynamics is difficult to interpret and the use of FRAP with Lifeact is not appropriate as a measure of actin stability or turnover.

Fig 5

The behavioral data meshes well with the structural changes in PVD observed in Figure 2. Premature beading of neurons is another interesting phenotype.

Overall the phenotypic descriptions are nice, but the mechanistic experiments are weak and conclusions related to these are not well supported.

Minor points

It would be helpful to explain the rationale behind using different PVD promoters in different experiments, and to be consistent with nomenclature (in some places promoter name is given and in some places it just says PVD promoter).

Spelling errors: (just the misspelled word is listed- should be able to search)

Influences

Additionally

Mechansims

Nevertheless

Comments to All Reviewers:

We would like to thank the reviewers for their positive assessment of our work and their constructive feedback. We are glad to see that all the reviewers found our work describing the novel role of KMN network in dendrite branching interesting.

We acknowledge the valid concern raised by reviewers that the data in our initial submission may not adequately support the idea that the dendrite branching phenotype observed in KNL-1 degrader animals is due to a direct effect on the actin cytoskeleton.

Since our initial submission, we have substantially strengthened the link between KNL-1-mediated actin regulation and PVD dendrite branching in the following ways:

- 1) Our new data demonstrates that the PVD branching abnormalities associated with KNL-1 loss can be mitigated by the actin-depolymerizing drug Latrunculin A (**new Fig. 6A-6C & S5A-S5D**). This finding supports the hypothesis that KNL-1 facilitates dendrite architecture by modulating F-actin assembly.
- 2) Our epistasis analysis between KNL-1 and actin nucleators provides further insights into how KNL-1 could modulate the actin cytoskeleton during dendrite branching (**new Fig. 6D-6F & S5E-S5K**). We have identified CYK-1, a *C. elegans* mDia3 formin orthologue and linear actin filament nucleator, as a genetic interactor of KNL-1 in the PVD neuron. Co-depletion of CYK-1 and KNL-1 suppresses the excess quaternary branches and dendritic self-avoidance defects seen with the tertiary dendrites in KNL-1 degrader animals.
- 3) We developed a gain-of-function assay demonstrating that plasma membrane-targeted KNL-1 can directly induce the formation of actin-rich clusters within PVD neurons (**new Fig. 7 & S6**). Importantly, this actin regulation by KNL-1 resides in its N-terminus, indicating that KNL-1's interaction with the microtubule-binding Ndc80 complex is not a prerequisite for its ability to initiate actin assembly.

Additionally, we recognize that our manuscript may have inadvertently given the impression that the microtubule-related function of the KMN network is not essential for dendrite branching. Our primary objective was to highlight an additional major role in actin regulation for KNL-1, a non-microtubule binding component within the KMN network. We agree with the reviewers that the microtubule-related function of KMN could contribute to both actin regulation and dendrite branching. In our initial submission, we intended to leverage state-of-the-art experiments that were routinely employed to investigate microtubule function in the PVD neuron context to differentiate the phenotypes resulting from KNL-1 loss from those of known microtubule regulators in PVD neurons. In doing so, we may have unintentionally misled readers into thinking that we are discounting the microtubule-related functions of the KMN network in dendrite branching. To address this issue in our revised manuscript, we have 1) expanded and included a more comprehensive

analysis of the PVD microtubule cytoskeleton after KNL-1 degradation, incorporating the reviewer's suggestions (**new Fig. 4G-4J**) and 2) revised the text to convey this concept more clearly.

Specific comments to the reviewers

Reviewer#1

The manuscript of Green et al is a very interesting description of phenotypes that arise when components of the KMN complex (particularly KNL-1) are degraded in a much-studied sensory neuron in C. elegans. It represents a significant advance over a previous study in C. elegans (by Dr. Cheerambathur) that first described a post-mitotic role for kinetochore proteins in C. elegans neurons. That paper built a very strong case for a legitimate post-mitotic role in neurodevelopment, but could not offer any detailed or mechanistic analysis of why the dendrites were abnormal in the particular sensory neurons that studied. By switching to a different sensory neuron and using cell-specific degradation of KNL-1, they were able to dig more deeply and found several interesting features, including incorrect branch spacing, increased ectopic protrusions from the primary dendrite, changes in the distribution and stability of actin, especially close to the cell body. The work is overall well conducted and with adequate attention to statistics and rigor, but there are a number of places that require more attention, as detailed below, and some unnecessarily sweeping statements in the abstract and discussion regarding the primacy of an actin phenotype that are ill-substantiated. Though it is a very good paper - the main point, that KMN complex is acting directly on the actin cytoskeleton and not primarily on microtubules is supported only partially and indirectly. No direct interaction is demonstrated and this is in contrast to prior work in Drosophila dendrites that pointed to microtubule interactions.

We thank the reviewer for the positive assessment of our work and have addressed the points raised below.

1. The signal from the ScarletPH domain is not adequately explained - is there more membrane at the branchpoint simply because it is a branchpoint and therefore has more surface to volume, or does it represent some internal clustering of membranes - which is what the images actually suggest?

We share the reviewer's curiosity about the increased signal from the mScarlet-I::PH probe and acknowledge that the suggestions put forward by the reviewer are alternative scenarios that could explain the intensity. To investigate this further we generated a cytoplasmic mScarlet-I probe. Notably, similar to the PH fusion, the cytoplasmic probe also exhibited increased fluorescence at the branch points (**new. Fig. S1F and S1I**), suggesting that the increased intensity in mScarlet-I may indicate an increased membrane volume or be an artefact of the mScarlet-I probe forming ectopic aggregates.

Anecdotal reports from multiple research groups, including ours, have described the tendency of the mScarlet-I probe to form ectopic aggregates in *C. elegans* somatic cells and associate with endolysosomal structures.

Furthermore, as explained below, we cannot definitively conclude that GFP::KNL-1 is present at all branch points; we can only demonstrate that a subset of branch points contains GFP::KNL-1 (**new. Fig. S1E-S1J**). Consequently, we have de-emphasized the branch point localization in our revised manuscript. Any further characterization of branch point analysis will require additional localization experiments using a more innocuous probe, such as mKate2.

2. The line scans along dendrites are not adequately explained in text or methods or figure legends. Each dendrite in each animal has multiple menorahs with 2o and 3o branches. Were all of them scanned in each animal or only a select few that had detectable scarlet and GFP signals? What exactly then does n=14 animals mean? If there has been some selection for visible GFP or scarlet at the branchpoint, then the conclusion is less robust. Figure S1 also undercuts the argument that there is a special relationship of KNL-1 and branchpoints. There are many more KNL-1 hotspots along the primary dendrite than there are axon branches in B and S1C shows two branches - one with and one without a KNL-1 hotspot, and the cartoon similarly indicates hotspots all over and not necessarily correlated with a branch. Bottom line is that I'm not persuaded the hotspots are enriched at branchpoints to a statistically significant extent.

We have expanded the Fig. 1 and S1 to include a more comprehensive characterization of GFP::KNL-1 localization along the primary dendrite and in menorahs at the L3 stage (**new Fig. 1D and Fig. S1**) and methods section to provide further clarification on the linescans and analysis of branch points. Specifically, we added overview images of the GFP::KNL-1 1) across the anterior dendrite (**new Fig. 1D & Fig. S1A**), 2) in the primary dendrite with the corresponding linescans (**new Fig. S1B**) and 3) expanded our analysis on branch point localization (**Fig. S1E-S1J**).

Briefly, for branch point analysis, we performed linescans across the first menorah anterior to the cell body and the 'n' value corresponds to the number of animals (**Fig. S1F & S1I**). To quantify the GFP::KNL-1 positive branch points we only analyzed the menorahs within the first 50 μm anterior to the cell body (**new. Fig. S1G & S1J**). Within this regions, KNL-1 is present approximately 75% and 50%, respectively, of the branch points in the primary and tertiary dendrites. We are unable to conclusively determine whether this coincidence or if there is a functional relevance to it and has revised the text to reflect this.

3. In Fig. 2, please define better the distinction between a 2o branch and an ectopic projection from the primary neuron. Are they only called 2o if they continue to develop 3o branches? Is that a meaningful distinction?

The ectopic projection is a membrane protrusion that emerges from the primary dendrite and do not form a 3° branch or menorah. Our analysis reveals that these filopodial protrusions lack microtubules (**new Fig. S4D & S4E**), which distinguishes them from most typical neurites. We have updated our terminology to refer to these structures as membrane protrusions for accuracy and clarity in our descriptions within the main text and methods section.

4. When branches appear to overlap or make contact, do they really make contact and is this really a failure of repulsion? In the Drosophila sensory dendrite literature there are several phenotypes that are mistakenly interpreted as a failure of self-avoidance when in fact they are misguidance phenotypes where the dendrites fail to stay in the proper layer of epidermis and therefor are out of plane with one another and only appear to overlap. See Kim et al 2012 PMC3470655 . In the present case, are the dendrite branches staying at the boundary of the epidermis and muscle cells or wandering out of that plane?

Dendritic self-avoidance between the 3° branches of the PVD neuron is very well-established (PMID: 20537990, 23889932, 29673481, 31220078 & 32631831). The 3° branches grow along the anterior-posterior axis of the sublateral nerve cord, which lie beneath the hypodermis and near body wall muscles. Time-lapse imaging has revealed that multiple protein classes, such as ligand/receptor pairs and actin regulators, impact PVD self-avoidance, with mutations in these genes leading to a specific failure in the retraction of adjacent 3° dendrite tips after contact. (PMID: 20537990, 29673481 & 32631831). Also, the morphology of the PVD neuron is distinguished by its highly stereotypical linear and orthogonal organization. The 3° branches strictly follow the A-P axis of the sublateral nerve cord, undergo fasciculation with other neurons within this structure and thus unlikely to stray from this path into surrounding tissues. Additionally, we have carried out our time-lapse analysis carefully by closely tracking the dendritic tips to ensure that they arose from adjacent 3° dendritic branches.

5. Does the change in filipodial number in Figure 4 represent increased initiation of the outgrowth or increased stabilization? Images were recorded at 1 frame every 20 min, which makes it impossible to distinguish these mechanisms. Initial outgrowth might indeed suggest an actin-dependent mechanism, but increased stabilization could well be due to a change in microtubule behavior. They do not report whether these filipodia contain a detectable microtubule.

We quantified filopodia-like protrusions by measuring the number of projections emerging from the primary dendrite during a 3-hour time-lapse of the PVD neuron at the L3 stage. The time lapses were acquired at 1 frame every 2 minutes and not every 20 minute which was a typo in our initial manuscript, and we apologized for this error. By

this stage secondary dendrite forming menorahs have already been established. These dynamic protrusions are devoid of microtubules (**new Fig. S4D & S4E**), exhibit extension and retraction behaviour but do not stabilize into secondary branches. We have modified the text to clarify this.

6. Figure 5H (y axis): Punctae is not a word. Puncta is the plural of punctum. (very minor point)

This has been corrected.

7. The Discussion is interesting and nicely reviews what indications there are in the literature that kinetochores and actin filaments may at times interact. They do consider (wisely IMHO) in the middle of p16 that the changes in actin distribution may be entwined with or secondary to changes in microtubules, In a few places, however, it makes rather sweeping claims that are not adequately supported by the experiments.

1. That the failure of repulsion/avoidance explains all the dendrite morphogenesis phenotypes. Even setting aside the concern mentioned above, contact repulsion is not likely to explain all the additional filipodia forming off the primary dendrite, nor the increase in the number of 4o branches.

We agree with the reviewer that the repulsion/self-avoidance does not explain all the phenotypes and have reworked the text to highlight the effect of KNL-1 degradation on protrusions, quaternary branches and dendrite self-avoidance.

2. That loss of KNL-1 affects the actin cytoskeleton and not the microtubule cytoskeleton. There are clearly changes in the abundance and stability of actin, but the evidence that there is no change in microtubules is much weaker. They only measure EB-3 comets in the primary dendrite (which does not have a strong phenotype), don't measure parameters such as lifetime of growing tip or invasion of filipodia) and ignore their previous conclusion that the microtubule binding site of Ndc80 is necessary for proper dendrite development in a different set of sensory neurons.

We have expanded our analysis of microtubule behaviour in primary dendrites and included new measurements on EBP-2 comet extension (**new Fig.4G-4J**). The microtubule signal intensity in the higher order dendrites is low compared to the primary dendrite (**see Fig.S3D**) and concomitantly it is technically difficult to observe the EBP-2::GFP comets in the higher order dendrites. Also, our current imaging tools lack the necessary brightness and sensitivity to perform reliable time-lapse imaging of microtubule dynamics and track microtubule tip growth into filopodia within higher-order dendrites. We acknowledge the importance of addressing this issue and are

currently developing more photostable EBP-2 probes (EBP2::mNeonGreen) in the PVD neuron.

We did not intend to overlook our previous findings on the role of Ndc80 microtubule-binding domains. We have shown that Ndc80 degradation affects PVD dendrite architecture indicating that the Ndc80 complex is essential for proper dendrite branching (**Fig. S2C and S2G-S2I**). We agree with the reviewer that it is important dissect the contribution of Ndc80 microtubule-binding domains to branching and we are developing gene replacement tools to express Ndc80 mutant transgenes in the PVD neuron. While this is beyond the scope of our current analysis, investigating how the microtubule-binding activity of the Ndc80 complex intersect with KNL-1-mediated actin regulation remains key to understanding how KMN affects dendrite branching.

3. They cite Hertzler et al. incorrectly as saying that microtubule growth dynamics were not altered in fly sensory neurons. In fact, Hertzler finds increased growing + ends in those dendrites; that article explicitly studies increased microtubule dynamics. This neuron in C. elegans may have a different use of the kinetochore, but you can't gloss over the difference by an inaccurate citation.

We respectfully disagree with the reviewer regarding the perceived inaccuracy. We have consistently cited Hertzler et al. as evidence for the role of kinetochore proteins in modulating microtubule dynamics (see Page 5 ; introduction). When referring to Hertzler et al. in the discussion, we stated that "components of the KMN network did not cause considerable changes in microtubule growth dynamics or microtubule stability in the dendrites." This statement was based on data presented in Hertzler et al. Supplementary Figure. S6, which showed that "Kinetochore protein knockdown does not change many metrics of microtubule polymerization," and Supplementary Figure. S5, which indicated that "Kinetochore protein knockdown may affect microtubule turnover in dendrites but not axons." To avoid misconceptions, we have made sure to use precise terminology in the revised version of our manuscript.

4. They state the microtubules are unaltered in 3o and 4o dendrites in Figure S3, but I can't see those microtubules at all, there is no quantification of them and no information on their dynamics.

We admit that Fig.S3D does not convey that microtubule distribution is similar in the PVD dendrites of control and KNL-1 degrader animals. This figure, showing GFP::TBA-1 expressed under a PVD-specific promoter, aimed to provide an overview of microtubule distribution across dendrites. It confirmed previous findings from multiple groups that microtubules are primarily enriched in the primary dendrite compared to higher-order branches using probes that directly incorporate into microtubules (e.g., TBA-1 fluorescent fusion, as seen in this manuscript and others (PMID: 22101643, 30694177,

31220078 &36126047) or using the microtubule binding proteins MAPH1.1 as the microtubule marker (PMID: 32293562). We agree that without quantitation, we cannot confirm or deny differences, so we have removed this from our current manuscript. While higher-order dendrites (2° and 3° tertiary) do contain microtubules, their low signal intensity makes it technically challenging to perform quantitative analyses, such as FRAP assays. We fully support the reviewer's suggestion that it is important to quantify microtubule dynamics and stability in the higher order branches of control and KNL-1 degrader animals. Currently, we are developing alternative microtubule probes, such as mNeonGreen::MAPH1.1 that binds to microtubule polymer, to overcome these technical challenges and further investigate the relationship between KNL-1 and microtubule regulation.

Reviewer #2 (Comments to the Authors (Required)):

Summary

In this manuscript, Green et al use the highly branched *C. elegans* PVD neuron to examine the cellular basis of dendritic patterning. Specifically, they examine terminal branching, self-avoidance, and maintenance regulation by the kinetochore protein KNL-1 and interactors. Prior studies of KNL-1 in worms and kinetochore proteins in flies and mammals support a wider role for kinetochores in neural development. The current study expands these findings to the dendritic arborization of PVD neurons. First, KNL-1 inhibits terminal branches, and small branch initiation from the primary dendrite, thereby controlling the PVD dendrite pattern. Second, KNL-1 degraded neurons show specific deficits in self-avoidance. Third, worms show defective proprioception and nociception. Fourth, dendrites show hallmarks of degeneration and potentially premature aging. As a link between chromosomes and spindle microtubules during cell division, the KMN network, which includes KNL-1, might be expected to regulate neuronal microtubule dynamics during development. However, the authors found no evidence to support this hypothesis. Rather, KNL-1 degradation appears to limit F-actin assembly and dynamics, especially in the cell body and proximal dendrites, potentially accounting for some of the dendrite phenotypes observed. While the authors describe a number of interesting new contexts for kinetochore protein function in neural development, their analysis does not yet offer satisfying new insight into processes underlying the observed phenotypes and thus the role of KNL-1 in normal development.

We appreciate the reviewer's acknowledgment of the importance of our findings on KNL-1's role in dendrite branching. We also value the constructive feedback provided, which has guided us in expanding upon the mechanistic basis of our work.

Major points

1. Kinetochore proteins have known roles in neural development in worms, flies, and

mammals. This paper will spawn many follow up studies in this system that will provide new insights into their roles, but the authors take a survey approach, and this paper does not go into depth to reveal the basis for any of the phenotypes that are described. For example, does KNL-1 interface with known regulators of self-avoidance or branch fusion such as the UNC-6 pathway, FMI-1, MIG-14, or EFF-1? This analysis could be especially interesting and insightful given the apparent similarity of the phenotypes in tertiary dendrites. The proposed premature ageing phenotype is not further explored except to look at the abundance of a molecular correlate of ageing in dendrites. Altogether the superficial analysis of various phenotypes reduced enthusiasm for this manuscript in its current form.

As noted by the reviewer, kinetochore proteins have been shown to affect various aspect of neuron development. Until now, all our work on kinetochore function in *C. elegans* has primarily focused on depletion of the KMN proteins in the context of groups of neurons (PMID: 30827898 & 38656792) limiting our ability to investigate specific mechanisms. This manuscript takes a significant step forward and addresses this limitation by examining the effects of KNL-1 depletion at the single neuron level. Our findings show that KNL-1 degradation from PVD significantly impacts distinct aspects of its morphology, behaviour and overall health. Although our approach may be considered a "survey approach," we believe this detailed descriptive documentation was necessary to establish a solid foundation for further mechanistic investigations. We agree with the reviewer that the important next step is expanding our research to identify the underlying mechanisms responsible for the observed phenotypes. In the revised manuscript, we have made a concerted effort to build upon our initial observations and specifically add more insights into the molecular basis of KNL-1's actin regulation and its relevance to PVD arborization.

As an alternate approach, motivated by the reviewer's suggestion to identify how KNL-1 interacts with established dendritic self-avoidance pathways, we also initiated a series of epistasis experiments with mutants of genes involved in self-avoidance. We began by testing the effect of the interaction between MIG-14/Wntless, which modulates actin assembly to promote dendritic self-avoidance, and KNL-1. Our preliminary findings suggest that KNL-1 degradation in the *mig-14* mutant background did not result in significant changes, indicating that KNL-1 may function downstream of MIG-14 (reviewer

Fig.1). To further characterize the interaction between KNL-1 and MIG-14, we are currently performing additional analysis on branch and actin dynamics, as well as overexpression of MIG-14 in the KNL-1 DEG background. Moreover, we are systematically examining KNL-1's interactions with other genes involved in self-avoidance. While this comprehensive analysis goes beyond the scope of our current study, we believe our ongoing work will provide important insights into KNL-1's role within these pathways.

2. The overall conclusion that phenotypes are not due to effects of microtubule dynamics is not sufficiently supported. The authors show that there are no obvious defects in microtubule trafficking or polarity along the anterior primary PVD dendrite. However, other than the ectopic branchlets along the primary arbor, KLN-1 phenotypes are observed in terminal parts of arbors, which were not studied in detail. Imaging analysis was also performed in later-stage arbors, which is presumably after morphological phenotypes would first develop. Lastly, it was difficult to assess the distribution of TBA-1::GFP in dendrites in Figure S3 and arguments were entirely qualitative. As a result, the argument that phenotypes upon KLN-1 degradation are not due to effects on microtubules was not convincing.

We concur with the reviewer's observation that the phenotypes induced by KNL-1 DEG predominantly manifest during the later stages of PVD development, particularly in terminal branches. We attribute this, in part, to the limitations of the TIR1 degrader system, as the degron might not be active early enough to influence the initial phases of development. Our ongoing work aims to address this concern by exploring alternative methods to manipulate KNL-1 activity during earlier developmental stages, which would allow us to better understand its role throughout the entire PVD development process.

We acknowledge that our morphological analysis primarily examines the late L4 stage, focusing on the final effects on terminal branches. However, we have also conducted time-lapse analysis of the arbor formation during the late L3/early L4 stage, which has allowed us to associate some of the observed defects with the impairment of self-avoidance mechanisms (**Fig. 2G-2J**). Additionally, our new experiments investigating actin regulation target the L3 stage, during which terminal arbors are being constructed (**new Fig. 6A-6C & S5A-S5D**). Ideally, capturing the entire arbor formation process through time-lapse imaging would provide further insights. However, the terminal branch formation spans several hours, and technical challenges, such as phototoxicity, have limited our ability to obtain time lapses longer than four hours with adequate resolution. We are actively working to refine our imaging techniques and, as the reviewer has aptly noted, will prioritize capturing this critical time window in our future analysis.

We agree that Fig. S3C does not effectively demonstrate similar microtubule distribution in the PVD dendrites of control and KNL-1 degrader animals. This figure (**new Fig. S3D**), which displays GFP::TBA-1 expressed under a PVD-specific promoter, was

intended to offer a general view of microtubule distribution across dendrites. Consequently, we have removed any statements implying comparisons between control and KNL-1 DEG based on this image from our text.

It is important to clarify that we are not claiming KNL-1 has no effect on microtubules, but rather that the impact of KNL-1's function either directly or through microtubules is primarily on the actin cytoskeleton, the key driver of dendrite branching. In fact, our new data indicate that KNL-1 influences microtubule growth length in primary dendrites (**new Fig. 4G-4J**), and the phenotypic similarities between KNL-1 DEG and Ndc80 DEG suggest that the microtubule-related functions of the KMN complex are crucial for dendrite branching (**Fig. S2B, S2C & S2G-S2I**). We have significantly revised the text to convey our thoughts and interpretations clearly.

3. Although the observations are careful and most phenotypes are quantified, my general take-away was that KNL-1 is required for many different aspects of PVD dendrite morphology. This is an important starting point, but I still don't know how or why these phenotypes emerge in any of the cases.

We agree with the reviewer that the role of KNL-1 during PVD neuron development is likely multifaceted as KNL-1 degradation seems to affect aspects of dendrite branching, sensory behaviour and overall neuronal health. In our manuscript we have focused primarily on how KNL-1 affect dendrite branch number and spacing through its potential actin regulation mechanism. Building on our observation that KNL-1 affects F-actin, we characterized the role of actin regulation in the context of higher order branch (quaternary) formation and dendrite self-avoidance mediated by tertiary branches, which were predominantly affected in KNL-1 DEG (**new Fig. 6 & S5**). Our new data confirms our hypothesis that KNL-1 modulates terminal branch number and spacing by influencing actin filament dynamics (**new Fig. 6A-6C & S5A-S5D**). We have identified a potential interactor, CYK-1, which may contribute to this process (**new Fig. 6D-6F & S5E-S5K**). Additionally, we have shown that KNL-1 can be activated at the plasma membrane to induce actin assembly, and we have performed a structure-function analysis to pinpoint this activity within the non-microtubule-binding KNL-1 N-terminus (**new Fig. 7 & S6**).

Minor points 4.

The resolution of the images in Figure 1B is poor, and it is therefore hard to visualize the differences between the red and green channels.

We assume the reviewer is referring to Fig.1C, which shows the localization of GFP::KNL-1 at the L3 stage. We expanded significantly on the L3 stage localization data for GFP::KNL-1 in the revised manuscript. In Fig. 1C (**new Fig. 1D**), we replaced the original panels with new images that shows overview of GFP::KNL-1 across anterior dendrite at the L3 stage and provided zoomed-in examples of different regions (**new Fig.1D & S1**),

offering a more comprehensive view of GFP::KNL-1 distribution during this developmental stage.

5. Different parts of the arbor are examined in certain experiments. Is there any regional specificity to the dendrite phenotypes along the A-P axis of the arbor?

We do not observe any regional specificity to the dendrite phenotypes upon KNL-1 depletion. In response to the reviewers' comments, we have now included a detailed breakdown of the two primary phenotypes: 4° dendrite branches and increased overlaps between 3° dendrites for both anterior and posterior dendrites (**new Fig. S2D and S2E**).

6. DMA-1 expression analysis is not convincing or maybe the image is not representative because DMA-1 signal looks different in control and KNL-1 DEG. If the authors measure mean intensity along the 4th order dendrites, they could be missing aspects of distribution. Perhaps a line-scan of signal intensity along the arbor could be informative.

We employed DMA-1 and RAB-3 distribution as a proxy readout for neuronal polarity to study how KNL-1 DEG affects it. In order to enhance this analysis, we have generated a new strain where mKate2::DMA-1 is expressed alongside a counter marker (mNG::PH) that labels the PVD plasma membrane, enabling us to distinguish between axons and dendrites more effectively (**new Fig.4C & S3B**). This has allowed us to demonstrate that DMA-1 localizes specifically to the dendrite, with no DMA-1 presence in the axons. It is possible that the distribution of DMA-1 in the dendrites may vary in the KNL-1 DEG animals across different parts of the dendrite. Investigating this aspect is an important avenue for future studies, considering that DMA-1 directly regulates actin polymerization. However, we note that the DMA-1 marker used in our current manuscript is a slightly overexpressed version under the ser2prom3 promoter. To facilitate more detailed analysis of DMA-1 distribution along the dendrite, we are in the process of creating an endogenous tagged DMA-1 strain, which will be utilized for future experiments.

7. The authors show that KNL-1 DEG causes behavioral phenotypes and argue that these are linked to actin misregulation and dendrite arborization phenotypes. Have the authors eliminated possible effects on axon arborization or terminal connectivity, or synaptic transmission independent of dendrites phenotypes? In Figure 1B it looks like KNL-1 is present at least in the proximal axon. Without this determination, conclusions about the functional effects of arbor defects seem more a matter of the paper's focused perspective on dendrites and should be qualified by stating other possible causes.

As correctly pointed out by the reviewer, KNL-1 signal is present in the axons during the early stages. Although our analysis did not reveal any apparent axonal morphological

defects in KNL-1 DEG animals, we cannot exclude the possibility of axonal defects or synaptic connectivity issues. Both proprioception and harsh touch are sensed by dendrites, with proprioception being mediated locally by dendrites through neuropeptide release, while harsh touch response requires axon-derived signals (PMID: 31735664). Given these distinct mechanisms, we agree with the reviewer's point that not all behavioural defects can be attributed solely to dendritic disruptions. We have revised the text to incorporate this perspective and acknowledge the potential contribution of axonal defects especially to the harsh touch response.

8. The authors argue that loss of KNL-1 leads to premature ageing due to the presence of beading and accumulation of autophagosomes. It seems that the phenotype could just be indicative of poor neuron health without invoking age-related explanations.

We agree with the reviewer's assessment that without direct evidence (e.g. suppression in a *daf-2* allele background), it is difficult to conclusively link the morphological abnormalities in KNL-1 DEG animals to an aging-related process. In our revised manuscript, we have revised the relevant section and replaced the term "premature aging" with "premature degeneration," which more accurately describes the observed phenotypes without implying an explicit connection to aging.

Reviewer#3 In "The kinetochore protein KNL-1 regulates the actin cytoskeleton to control dendrite branching" the authors build on their previous work showing that KMN network proteins function post-mitotically in neurons in C. elegans. Previously they had only observed changes in neurons that use a cilium as a dendrite. They now extend this work to show that neurons with branched dendrites (PVD) have structural and functional deficits. The portions of the manuscript that describe the PVD phenotype from degradation and overexpression of KNL-1 are convincing. In Figure 2 structural changes in PVD are shown and in Figure 5 behaviors known to involve PVD are altered in animals in which KNL-1 is reduced in PVD. However, the mechanistic parts of the story are not convincing; problems with assays and interpretations are detailed below.

We thank the reviewer for the positive assessment of the phenotypic descriptions of the KNL-1 loss to PVD morphology and function. Following up on the reviewer's feedback in our revised manuscript we have significantly expanded the mechanistic aspects of KNL-1's role in actin regulation and its relationship to dendrite branching.

Fig 1- Complementation of endogenous KNL-1 tagged with GFP11 by cell-type specific expression of GFP 1-10 in PVD is used to show KNL-1 presence and localization in post-mitotic PVD neurons. In some contexts GFP1-10 can have weak fluorescence on its own so this data would be strengthened by including control images of GFP1-10 expressed in PVD without KNL-1- GFP11. This tagging strategy is also used to conclude KNL-1 localizes

at branch points. However this finding is not demonstrated convincingly. It would be helpful to have overviews that show the localization of KNL-1 at different stages. Also, what percent of branch points have KNL-1 puncta? Is this higher than chance based on spacing of puncta along the dendrite? Also, the average increase in fluorescent intensity of KNL-1 at branch points is not any different than the membrane marker (Figure 1D) although in the images shown the membrane marker looks fairly evenly distributed while the KNL-1 is not.

We and several others have routinely employed the split GFP system for cell type specific visualization of proteins in *C. elegans* (PMID: 28767038, 30952669 & 30827898). In our experience, the fluorescence arising from GFP1-10 alone is not significant over the autofluorescent background in the animals (**new Fig. S1A**). Also, our GFP 1-10 constructs are expressed as single-copy transgenes to prevent experimental artefacts arising from overexpression. To further confirm that the observed signal in the GFP channel is not due to weak fluorescence, we have included control images from a strain lacking endogenously tagged 7XGFP β 11::KNL-1, which showed that the signal in the primary dendrite and cell body of the 7XGFP::KNL-1 is significantly higher than in the control (**new Fig. S1A**).

As recommended by the reviewer we have added overview images of the 7XGFP::KNL-1 at the L3 stage (**new Fig. 1D & S1B**). Specifically, we added images of the GFP::KNL-1 across the anterior dendrite (**new Fig. 1D & Fig. S1A**), 2) in the primary dendrite with the corresponding linescans (**new Fig. S1B**) and 3) expanded our analysis on branch point localization (**new Fig. S1E-S1J**).

We cannot definitively conclude that GFP::KNL-1 is present at all branch points; we can only demonstrate that a subset of branch points contains GFP::KNL-1 (**new. Fig. S1E-S1J**). To quantify the GFP::KNL-1 positive branch points we only analyzed the menorahs within the first 50 μ m anterior to the cell body (**new. Fig. S1G & S1J**). Within this regions, KNL-1 is present approximately 75% and 50%, respectively, of the branch points in the primary and tertiary dendrites. We are unable to conclusively determine whether this is coincidence or if there is a functional relevance to it. Consequently, we have de-emphasized the branch point localization in our revised manuscript.

We have also compared the branchpoint puncta of GFP::KNL-1 with a cytoplasmic mScarlet-I probe (**new. Fig. S1E-S1I**). Notably, similar to the PH fusion, the cytoplasmic probe also exhibited increased fluorescence at the branch points (**new. Fig. S1F and S1I**), suggesting that the increased intensity in mScarlet-I may indicate an increased membrane volume or be an artefact of the mScarlet-I probe forming ectopic aggregates.

Fig 2- The gain and loss of function effects on quaternary branching are very convincing and satisfyingly opposite. For the last point in the figure- the additional fusion between dendrites, does this always happen when there is extra terminal growth? Or is it a specific

change in this background. It would be helpful to know whether other overgrowth genotypes have similar extra fusion.

The data presented in Fig. 2G-J suggest that the overlapping menorahs may result from reduced contact-dependent repulsion between 3° dendrites in neighbouring menorahs. Each menorah originates from a 2° dendrite, and 3° dendrites emerge orthogonally from the 2° dendrites. Both KNL-1 degrader animals and controls display a similar number of 2° dendrites (Fig. S2B), indicating that the number of 3° dendrites is also likely to be the same. This suggests that the increased overlap between 3° dendrites in KNL-1 degrader animals is not due to additional outgrowth of 3° dendrites. Instead, the observed overlap appears to be caused by an impaired ability of the 3° dendrites in KNL-1 degrader animals to perform contact-dependent retraction compared to the control. Similar effect on branch spacing have been reported for mutations in several genes, including those encoding ligand/receptor pairs and actin-assembly factors (PMID: 20537990, 29673481 & 31220078). We have revised the text and the panels in Fig 2G-2J to provide a clearer explanation of this process.

Fig 3 and S3- One of the major conclusions highlighted in the abstract is that KNL-1 does not act in PVD via the microtubule cytoskeleton. The data supporting this conclusion is presented in Figure 3 and S3. As it the microtubule-binding domain of Ndc80 was shown to be important in ciliated neurons in C. elegans, and many kinetochore proteins were shown to regulate microtubule dynamics in Drosophila dendrites, this is surprising. Previously it was shown that reduction of KMN network proteins increases the number of microtubule plus ends per unit length in Drosophila neurons. It looks as though this might also be the case here (Fig S3E) as the three highest values are in the KNL-1 DEG background. However only 11 cells were assayed, so more data is needed to see if the distribution is different or to make a strong conclusion that it is not. In the EB1 kymographs shown (3D) it looks like length of time growing may be slightly longer in the KNL-1 DEG scenario. It always requires a very high bar to show that something is not affected, especially when previous studies have implicated these proteins in microtubule regulation so more data on number of growing microtubules and other aspects of microtubule behavior seems warranted.

We would like to clarify that our intention was not to imply that the microtubule-binding function of the KMN is unimportant in PVD dendrites. In fact, our own prior data indicate that the microtubule-binding interface of the Ndc80 complex is essential for both axonal and dendritic functions of the KMN (PMID: 30827898 & 38656792), and the depletion of the Ndc80 complex results in similar dendrite branching disruptions as those observed with KNL-1 depletion (**Fig. S2C and S2G-S2I**). The experiments presented in Fig. 3 (**new revised as Fig. 4**) aimed to demonstrate that the effects of KMN removal from PVD neurons differ from the phenotypes associated with mutations in typical microtubule-

binding proteins (PMID: 22101643, 23482306, 26633194, 32293562, 36395330 & 38985513). To avoid any confusion, we have substantially revised the text and provided clearer interpretations of the experiments in the new Fig. 4.

Furthermore, in response to the reviewer's valuable suggestions, we have significantly increased our sample sizes to strengthen our quantifications (**new Fig. 4G-4J**) and broadened the range of microtubule parameters examined, such as EBP-2 comet length. We are particularly grateful to the reviewer for spotting the comet length differences between the control and KNL-1 DEG animals, which prompted us to further investigate and measure the EBP-2 comet length (**new Fig. 4I**). Our improved analysis reveals a slight but significant increase in EBP-2 comet velocity and a substantial increase in growth length. These findings demonstrate that KNL-1 depletion does indeed impact the microtubule cytoskeleton and could potentially contribute to the observed phenotypic abnormalities, including mis-regulation of actin. We are currently exploring the potential crosstalk between microtubules and actin mediated by the KMN network in the PVD. Additional studies that fall outside the current scope of this manuscript will be needed to understand the underlying mechanisms.

It is unclear what FRAP of tubulin represents in 3F and S3- is this just diffusion of free tubulin back into the region since the overall percent recovery is low? If so, this parameter would be unrelated to microtubule behavior but would only report on size of free tubulin as diffusion would largely be controlled by size. This assay is said to represent microtubule stability, but it seems much more likely that on a 60s-time scale only diffusion of free tubulin is being assessed rather than anything about stability of polymerized microtubules.

The FRAP data reveals that approximately 80% of GFP::TBA-1 does not recover within the 120-second imaging timeframe, which is consistent with previous observations and most likely reflects the stable microtubule population (PMID: 32293562). The remaining 20% mobile fraction may comprise freely diffusing molecules and possibly newly incorporated tubulin, while most GFP::TBA-1 is part of the microtubule polymer structure. Our analysis indicates that the immobile fraction of GFP::TBA-1 is similar between the control and KNL-1 DEG within the timescale of our imaging. The timescale for freely diffusing molecules of similar size in the PVD is on the order of a few seconds (PMID: 32293562). To maintain consistency in terminology, we will remove stability from the text and refer as microtubule turnover.

The pattern of DMA-1 in the control and KNL-1 DEG neurons shown in Fig S3A actually looks quite different although it is described as the same. It seems very enriched near tips of control dendrites and much more evenly distributed in the KNL-1 DEG neurons.

We employed DMA-1 and RAB-3 distribution as a proxy readout for neuronal polarity to study how KNL-1 DEG affects it. In order to enhance this analysis, we have generated a new strain where *mKate2::DMA-1* is expressed alongside a counter marker (*mNG::PH*) that labels the PVD neuron, enabling us to distinguish between axons and dendrites more effectively (**new Fig.4C & S3B**). This has allowed us to demonstrate that DMA-1 localizes specifically to the dendrite, with no DMA-1 present in the axons. It is possible that the distribution of DMA-1 in the dendrites may vary in the KNL-1 DEG animals across different parts of the arbor. Investigating this aspect is an important avenue for future studies, considering that DMA-1 directly regulates actin polymerization. However, we note that the DMA-1 marker used in our current manuscript is a slightly overexpressed version under the *ser2prom3* promoter which prevents us from reliably quantifying DMA-1 distribution spatially across the dendrites. To facilitate more detailed analysis of DMA-1 distribution along the dendrite, we are in the process of creating an endogenous tagged DMA-1 strain, which will be utilized for future experiments.

Overall the data presented in both Figure 3 and S3 is problematic, in some cases an conclusions are drawn that are not supported by the type of assay used (Figure 3D) and in other cases the data presented does not seem like it supports the conclusion that control and KNL-1- DEG neurons are similar (Figure S3A and S3E).

We have addressed the above concerns by addition of new data (**new Fig. 4I**), increasing the sample size (**new Fig. 4G, 4H & 4J**), providing controls (**new Fig. 4C & S3B**) and clarifying the text referring to the figure describing the effects of KNL-1 degradation on microtubule organization and dynamics in the PVD(**new Fig. 4 & S3**) .

Fig 4

One of the central points of the paper is that KNL-1 regulates the actin cytoskeleton. Support for this conclusion derives from the use of the Lifeact reporter shown in Figure 4. The distribution of the reporter appears quite different in the control and KNL-1 DEG neurons, but in both cases seems to label only a subset of actin as many of the terminal actin-based branches do not contain the reporter. This is consistent with previous reports indicating that only a subset of actin structures are labeled with Lifeact.

We recognize the concerns regarding the use of various actin probes, such as Lifeact. Lifeact has been widely employed to study actin filament dynamics in *C. elegans* across multiple contexts, including cell contractility and cytokinesis (PMID: 31221727, 35978196, 37665665) as well as the actin cytoskeleton within the PVD neurons (PMID: 29738713, 31220078, 34281597). Our observations with the *Lifeact::mKate2* show that the signal is consistently brightest near the tips of growing dendrites compared to internal dendritic regions, consistent with the distribution pattern described in previous studies employing F-actin visualization probes like *Lifeact::GFP* (PMID: 29738713,

31220078, 34281597) or Utrophin Calponin Homology domain (UtrCH) fusions in PVD neurons (PMID: 30694177). Furthermore, as observed previously using Lifeact or UtrCH probes, we observed Lifeact labelled F-actin in only a subset of quaternary branches, presumably those that are being newly formed. This selective labelling could explain why only a portion of the terminal branches displayed a Lifeact signal. Additionally, as the reviewer mentioned, this could also be attributed to Lifeact labelling a subset of actin structures. Nevertheless, our data demonstrate that the Lifeact::mKate2 signal at the cell body and the primary dendrite in the control and KNL-1 DEG neurons diminish upon Latrunculin treatment, suggesting that the probe is indeed labeling a form of polymerized actin (**new Fig. S5C & S5D**).

Much of the data in this figure deals with changes in the cell body that are difficult to relate to dendrite structure. It is also unclear whether these changes (for example cell body size) are related to expression of the Lifeact reporter as they are not mentioned earlier in the paper when different visualization methods are used (cell bodies shown in fig 2A are very similar in both backgrounds).

To further strengthen the connection between actin regulation and dendrite branching, we performed additional experiments demonstrating that modulating actin dynamics in the KNL-1 DEG background can suppress branch abnormalities caused by KNL-1 depletion (**new Fig. 6 & S5**). These new results strengthen our hypothesis that KNL-1 plays a role in actin cytoskeleton organization and dendrite branching. Furthermore, the cell body measurements were obtained using a plasma membrane probe and thus observed changes are not a consequence of Lifeact expression. We have revised the text to provide clearer information about these experimental details.

Lifeact is referred to as a way to monitor F-actin, but it also binds G-actin so will label the soluble portion as well. Lifeact has been reported to dissociate from actin too rapidly to be used in FRAP assays to monitor actin dynamics (<https://www.tandfonline.com/doi/full/10.1080/19490992.2014.1047714>), so it is unlikely that the data shown in H and I is related to turnover of F-actin as described in the results. Overall, the data on actin dynamics is difficult to interpret and the use of FRAP with Lifeact is not appropriate as a measure of actin stability or turnover.

We recognize the concerns raised by the reviewer regarding the use of Lifeact as a measure of actin turnover. Our experiments employing Latrunculin A demonstrate that the increased signal intensity observed in the primary dendrite of KNL-1 DEG decreases in the presence of Latrunculin A, similar to the cell body. However, we concur that the FRAP of Lifeact measurements might reflect the binding and unbinding of Lifeact probe to actin. In light of this, we have removed the FRAP data from the revised manuscript and

modified our conclusions. In future studies, we intend to utilize additional actin probes, such as UtrCH, in parallel with the Lifeact probe to further validate our findings.

Fig 5 The behavioral data meshes well with the structural changes in PVD observed in Figure 2. Premature beading of neurons is another interesting phenotype. Overall the phenotypic descriptions are nice, but the mechanistic experiments are weak and conclusions related to these are not well supported.

We would like to thank the reviewer for the positive comments regarding the phenotypic description. In response to the feedback provided by the reviewer, we have made a concerted effort to enhance the mechanistic aspect of our work, as detailed in our previous responses.

Minor points

It would be helpful to explain the rationale behind using different PVD promoters in different experiments, and to be consistent with nomenclature (in some places promoter name is given and in some places it just says PVD promoter).

We have revised the figures and text to include this information.

Spelling errors: (just the misspelled word is listed- should be able to search)

Influences

Additionally

Mechansims

Neverthles

We thank the reviewer for catching the spelling errors. They have been corrected in the new version.

August 25, 2024

Re: JCB manuscript #202311147R-A

Dr. Dhanya K Cheerambathur
University of Edinburgh
Wellcome Centre for Cell Biology
School of Biological Sciences
Kings Buildings
Edinburgh EH9 3BF
United Kingdom

Dear Dr. Cheerambathur,

Thank you for submitting your revised manuscript entitled "The Kinetochore Protein KNL-1 Regulates the Actin Cytoskeleton to Control Dendrite Branching." Your revised manuscript has been evaluated by each of the three original reviewers with much more positive evaluations, but two of the three reviewers have identified some remaining issues that require your attention and revision before submitting a final re-revised version for editorial evaluation of its appropriateness for final acceptance.

Considering the few, but significant comments of the first reviewer, it seems important to provide the p values, update Figure 2D, and either clarify the sensitivity of your ability to detect tubulin or temper your statement regarding its putative absence in protrusions from primary dendrites.

The third reviewer has a more extensive list of remaining concerns: 1. Reservations about the CYK-1 suppression experiment (paragraphs 1 and 5); 2. Concern that you are measuring turnover of tubulin, not microtubules with your FRAP experiment in Figure 4B; 3. Concerns about low n values; 4. Absence of some important comparisons, plus some more minor issues.

Our general policy is that papers are considered through only one revision cycle; however, given that the suggested changes are relatively minor we are open to one additional short round of revision. We believe that it is important for you to address each of the issues raised by the first and third reviewers, including increasing low n values. We expect to make a final decision editorially, but are optimistic that with some effort by you and your lab members that a final version will be acceptable. Please do not submit a revision prematurely without these issues convincingly addressed. Please submit the final revision along with a cover letter that includes a point by point response to the remaining reviewer comments.

Thank you for this interesting contribution to Journal of Cell Biology. You can contact me or the scientific editor listed below at the journal office with any questions at cellbio@rockefeller.edu.

Sincerely yours,

Louis Reichardt, PhD
Monitoring Editor
Journal of Cell Biology

Dan Simon, PhD
Scientific Editor
Journal of Cell Biology

Reviewer #1 (Comments to the Authors (Required)):

The revised manuscript is much stronger. The new data are informative and the claims of the manuscript more circumspect and in keeping with the data. In particular, the actin phenotype is better described and the experiments linking membrane-anchored KNL1 to actin recruitment are very interesting. There are certainly some aspects of the mechanism that remain opaque, but there is plenty of new information in this manuscript to justify its publication.

Some technical matters need to be fixed however.

1. Though the ms no longer claims that there is fusion of the 3o branches of the menorahs, Fig 2D cartoon is still labelled as fusion.
2. The figures are marked with * and ns rather than actual p values and this is not acceptable. In some cases, such as the quantification of comet velocity and comet number, one is NS and the other is * but the distinction might be just 0.051 vs 0.049 and that is an insignificant difference in "significance". The real p values are always needed so the reader can judge for

themselves the degree of confidence in the conclusion. In the case of the comets, where the shift in the means between all the parameters are very similar, I strongly suspect that the difference in so-called significance is not very meaningful and says more about the variability inherent in the biology and measurements than in the underlying biological phenomenon. In all the figures, a single * and ns need to be replaced with real values, even when it probably doesn't matter, as in Figure 6E,F.

3. The manuscript asserts that the excess protrusions from the 1o dendrite were devoid of microtubules, but that is only true if the sensitivity of the assay is sufficient to see single microtubules and the possibility of transient entry of microtubules into nascent dendrites or filipodia is harder to exclude.

Reviewer #2 (Comments to the Authors (Required)):

The authors have addressed the critiques with new analysis, new figure panels, changes in wording, and clarifications. A minor glitch arose in the revision that the reference to figure S1A appears to have been lost. Authors should either reintroduce or eliminate that panel.

Reviewer #3 (Comments to the Authors (Required)):

The authors have removed some of the problematic data and added new data to strengthen the manuscript. The suppression of the KNL-1 phenotype by Lat-A is a good addition. However, the suppression by CYK-1 is not entirely convincing and is oversold. There are also some remaining problems with data and description.

While one of the problematic FRAP experiments was removed, the description of the remaining one is still wrong (Figure 4F). It is introduced by saying: "We first measured the turnover of microtubules in neurons expressing GFP::TBA-1." On the time frame of seconds, this is not what is being measured. Microtubules in neurons typically turnover on the hours scale. They are measuring free tubulin and its diffusion- as they later go on to state. It is unclear how the percentage of free tubulin might relate (if at all) to microtubule turnover or stability, so from this data they cannot conclude "and that the overall dynamics of the microtubule network is remains unchanged." Now can they state "these experiments suggest that although global microtubule turnover remains largely unaffected". Please can the misleading statements about the assay's ability to monitor overall turnover and dynamics of microtubules be removed.

There are still a couple of places where low n's may obscure potential differences- for example, Figure 6B, a point is made about LatA conditions being found that do not perturb quaternary branching on their own, but the quaternary branching is trending low in the control background and the vehicle control n's are much lower than the n's for KNL-1 vehicle. Similarly, as in the previous iteration the comet number/min looks higher in KNL-1 than control in Figure 4H, but again it seems likely that low n's are obscuring the difference. The scale of the difference looks similar to the other microtubule changes in 4G and 4I, but those have many more n's because they are measured for each microtubule and so are statistically significant.

The authors display the two sets of suppression data paired up differently and with different statistical comparisons, and this means that a critical comparison is missing. In 6B in the control genetic background vehicle and LatA are compared, but the comparable genotypes in 6E would be Control (far left) and CYK-1 alone and these are not compared. However, in the text the authors say "CYK-1 degradation alone did not result in PVD dendrite branching defects," despite this comparison not being shown.

While the LatA suppression is convincing and a nice addition, the CYK-1 suppression is weak and may just represent the additive phenotype of CYK-1 alone (which look like it reduces branch number, but this is the missing comparison) and KNL-1. The suppression by CYK-1 is highlighted as a key point throughout and the data is just not strong enough to do that. Is the difference between the Control and KNL-1 in the CYK-1 background actually any less than in the control background? From the figure it does not look like it.

The ability of plasma membrane targeted KNL-1 to generate ectopic actin structures is intriguing and quite striking. However, it is difficult to relate it to the rest of the data that suggests the role of KNL-1 is to inhibit actin assembly. It would be good to at least address this discrepancy more directly in the discussion.

Minor points:

Different versions of graphing software were used to make graphs- for example in Figure 6 the right hand graphs have points aligned horizontally while left have graphs have them scattered- this is a little distracting.

Spelling: "obaserved"

Response to the Reviewers:

Reviewer #1:

The revised manuscript is much stronger. The new data are informative and the claims of the manuscript more circumspect and in keeping with the data. In particular, the actin phenotype is better described and the experiments linking membrane-anchored KNL1 to actin recruitment are very interesting. There are certainly some aspects of the mechanism that remain opaque, but there is plenty of new information in this manuscript to justify its publication.

We thank the reviewer for the positive assessment of our revised manuscript.

Some technical matters need to be fixed, however.

1. Though the ms no longer claims that there is fusion of the 3o branches of the menorahs, Fig 2D cartoon is still labelled as fusion.

We thank the reviewer for making us aware of this error, we have corrected this to “overlap” in Fig. 2D

*2. The figures are marked with * and ns rather than actual p values and this is not acceptable. In some cases, such as the quantification of comet velocity and comet number, one is NS and the other is * but the distinction might be just 0.051 vs 0.049 and that is an insignificant difference in "significance". The real p values are always needed so the reader can judge for themselves the degree of confidence in the conclusion. In the case of the comets, where the shift in the means between all the parameters are very similar, I strongly suspect that the difference in so-called significance is not very meaningful and says more about the variability inherent in the biology and measurements than in the underlying biological phenomenon. In all the figures, a single * and ns need to be replaced with real values, even when it probably doesn't matter, as in Figure 6E,F.*

Following the reviewer's recommendation, we have replaced all single * and ns with exact p-values. For the **, *** and **** asterisks, we have included the corresponding p-value thresholds $p \leq 0.01$, $p \leq 0.001$, and $p \leq 0.0001$, respectively.

3. The manuscript asserts that the excess protrusions from the 1o dendrite were devoid of microtubules, but that is only true if the sensitivity of the assay is sufficient to see single microtubules and the possibility of transient entry of microtubules into nascent dendrites or filipodia is harder to exclude.

We appreciate the reviewer's point concerning the potential presence of transient or low numbers of microtubules in filopodial protrusions, which may be undetectable with our current imaging capabilities. To reflect this possibility, we have revised the relevant sections of the manuscript accordingly.

Reviewer #2:

The authors have addressed the critiques with new analysis, new figure panels, changes in wording, and clarifications.

We thank the reviewer for the positive evaluation of our revised manuscript.

A minor glitch arose in the revision that the reference to figure S1A appears to have been lost. Authors should either reintroduce or eliminate that panel.

Thank you for bringing this oversight to our attention. We have now reintroduced the reference to Fig. S1A in the revised manuscript.

Reviewer #3:

The authors have removed some of the problematic data and added new data to strengthen the manuscript. The suppression of the KNL-1 phenotype by Lat-A is a good addition. However, the suppression by CYK-1 is not entirely convincing and is oversold. There are also some remaining problems with data and description.

We thank the reviewer for his/her insightful comments. Following up on the reviewer's feedback we have increased the sample sizes as suggested and have revised the manuscript to incorporate the recommended changes.

While one of the problematic FRAP experiments was removed, the description of the remaining one is still wrong (Figure 4F). It is introduced by saying: "We first measured the turnover of microtubules in neurons expressing GFP::TBA-1." On the time frame of seconds, this is not what is being measured. Microtubules in neurons typically turnover on the hours scale. They are measuring free tubulin and its diffusion- as they later go on to state. It is unclear how the percentage of free tubulin might relate (if at all) to microtubule turnover or stability, so from this data they cannot conclude "and that the overall dynamics of the microtubule network is remains unchanged." Now can they state "these experiments suggest that although global microtubule turnover remains largely unaffected". Please can the misleading statements about the assay's ability to monitor overall turnover and dynamics of microtubules be removed.

We respectfully disagree with the reviewer's assessment that the GFP::TBA-1 recovery merely represents free tubulin diffusion. Previous research on the PVD neuron, alongside studies of microtubule turnover in various biological processes, indicates that diffusion of molecules similar to that of tubulin or of tubulin typically occurs within a half-life of less than one second (PMID: 32293562, 17576796). However, the half-life for GFP::TBA-1 recovery in our FRAP experiments, assuming a one-phase association, is considerably longer, averaging 9.8 seconds in control and 9.7 seconds in KNL-1 degrader animals. These values are consistent with the high dynamic recovery rates observed in microtubule networks during mitosis (PMID: 17576796). Additionally, the presence of EBP-2 decorated microtubule plus ends supports the inference of ongoing microtubule dynamics. To

address the complex nature of these measurements, we acknowledge that an even more rigorous method such as using photoactivatable tubulin would provide clearer insights into microtubule turnover, although this is beyond the scope of the current manuscript.

Our primary intent of performing FRAP of GFP::TBA-1 was to compare microtubule behaviour between control and KNL-1 degrader animals. To avoid possible misinterpretations, we have revised the manuscript to only present the observed results within our experimental conditions, without making extrapolative statements about microtubule turnover. We have removed the phrases specifically highlighted by the reviewer and revised the text to merely report on the observed results under our specific experimental conditions. We hope that this will allow readers the opportunity to draw their own conclusions from the FRAP curves.

There are still a couple of places where low n's may obscure potential differences- for example, Figure 6B, a point is made about LatA conditions being found that do not perturb quaternary branching on their own, but the quaternary branching is trending low in the control background and the vehicle control n's are much lower than the n's for KNL-1 vehicle.

To address the reviewer's concerns regarding the potential for low sample sizes to obscure significant differences, we have increased the n values in our Latrunculin A (LatA) experiments (Fig. 6A-C and Fig. S5B). In the revised data the sample sizes are similar ranging from 41 to 43 animals across all conditions. The statistical analysis (one-way ANOVA followed by Tukey's multiple comparison for **Fig. 6B** and **Fig. S5E** and Kruskal-Wallis followed by Dunn's multiple comparison test for **Fig. 6C** and **Fig. S5F**) confirms that the differences between the LatA Control and the DMSO Control are not statistically significant, thus, reinforcing the validity of our conclusions that LatA conditions do not independently perturb quaternary branching.

Similarly, as in the previous iteration the comet number/min looks higher in KNL-1 than control in Figure 4H, but again it seems likely that low n's are obscuring the difference. The scale of the difference looks similar to the other microtubule changes in 4G and 4I, but those have many more n's because they are measured for each microtubule and so are statistically significant.

We have significantly increased the sample size for the EBP-2 comets/min analysis presented in **Fig. 4H**, with 55 animals for the control group and 59 for the KNL-1 degrader group, pooled across more than three independent experimental replicates. This sample size is consistent with state-of-the-art EBP-2 analysis in various organisms, including *C. elegans*. For similar measurements, the number of neurons analyzed in *C. elegans* neurons including the PVD neuron typically ranges from 15-60 (see Harterink et al., 2018, JCS (PMID: 30254025) and Puri et al., 2021, JCB (PMID: 34137792)). Furthermore, Hertzler et al., 2020 (PMID: 32673176), who showed the impact of kinetochore on EB-1 dynamics in *Drosophila* larval sensory neurons, reported sample sizes ranging from 9-33 neurons for their EB-1 comets/min analysis.

Unlike EBP-2 track measurements (Fig. 4G and 4I), where multiple tracks can be collected from a single neuron, our EBP-2 comets/min are derived from an average rate over a 1-minute movie capturing the dynamics of EBP-2 in a single neuron. This approach limits us to obtaining just one data point per neuron for each individual animal. Additionally, the necessity to capture specific developmental events within this brief window severely constrains our capacity to gather multiple data points in a single experiment. Thus, to match the sample sizes in Figures 4G and 4I, which number in the hundreds, would necessitate dedicating several months exclusively to these measurements. Given that this data is not central to the main findings of our manuscript, we believe that such extensive analysis is beyond the well-established norms for this type of analysis and is not necessary. Moreover, we are cautious of overly expanding these measurements, as doing so could lead to artificial interpretations, potentially highlighting differences that, while statistically noticeable, might not be biologically relevant.

The authors display the two sets of suppression data paired up differently and with different statistical comparisons, and this means that a critical comparison is missing. In 6B in the control genetic background vehicle and LatA are compared, but the comparable genotypes in 6E would be Control (far left) and CYK-1 alone and these are not compared. However, in the text the authors say "CYK-1 degradation alone did not result in PVD dendrite branching defects," despite this comparison not being shown.

While the LatA suppression is convincing and a nice addition, the CYK-1 suppression is weak and may just represent the additive phenotype of CYK-1 alone (which look like it reduces branch number, but this is the missing comparison) and KNL-1. The suppression by CYK-1 is highlighted as a key point throughout and the data is just not strong enough to do that. Is the difference between the Control and KNL-1 in the CYK-1 background actually any less than in the control background? From the figure it does not look like it.

We appreciate the reviewer's suggestion regarding the necessity for a direct comparison between CYK-1 DEG and Control DEG conditions to robustly support the statement that 'CYK-1 degradation alone did not result in PVD dendrite branching defects.' Accordingly, we have now added the statistical significance between these conditions to the revised manuscript. Additionally, we have increased the sample sizes for the CYK-1 DEG conditions to align with those of the other experimental groups. The statistical analysis, adjusted for these sample sizes, revealed no significant differences in arborization parameters between the control and CYK-1 degrader animals, aligning with results reported earlier.

While it is possible that the observed suppression with CYK-1 could represent an additive phenotype, our data do not show any significant differences in branch number or overlap under CYK-1 depletion compared to the control. However, we acknowledge that we currently lack sufficient data to definitively assert that the suppression is directly due to an interaction between KNL-1 and CYK-1 activity. Instead, the findings suggest that perturbations in an actin regulator like CYK-1 are sufficient to partially suppress the KNL-1 degrader phenotype, reinforcing the hypothesis that KNL-1 interacts with actin regulatory pathways. To prevent overstating the significance of this suppression, we have

carefully revised the relevant sections of the manuscript to moderate the interpretations of these results.

The ability of plasma membrane targeted KNL-1 to generate ectopic actin structures is intriguing and quite striking. However, it is difficult to relate it to the rest of the data that suggests the role of KNL-1 is to inhibit actin assembly. It would be good to at least address this discrepancy more directly in the discussion.

We agree with the reviewer that the ability of KNL-1 to generate ectopic actin structures could seem counterintuitive to the data that implies that KNL-1 is inhibiting actin assembly. We have expanded our discussion section to provide potential mechanisms that could explain these contrasting observations.

Minor points:

Different versions of graphing software were used to make graphs- for example in Figure 6 the right hand graphs have points aligned horizontally while left have graphs have them scattered- this is a little distracting.

We appreciate your observation regarding the graphical presentations in Figure 6. We have verified that the same version of graphing software was used to generate all graphs. The variation in the alignment of data points across the graphs primarily results from the inherent differences in the range of values observed in Fig. 6 for the number of 4 degree dendrites compared to those for overlapping menorahs. A broader range in one dataset allows for a more dispersed point distribution, whereas a narrower range results in a visually more aligned distribution. In the future we will consider alternative ways of displaying similar data to avoid such distractions.

Spelling: "obaserved"

Thank you for catching the typo. We have corrected this in our revised manuscript.

November 7, 2024

RE: JCB Manuscript #202311147RR

Dhanya Cheerambathur
University of Edinburgh

Dear Dr. Cheerambathur,

Thank you for submitting your revised manuscript entitled "The Kinetochore Protein KNL-1 Regulates the Actin Cytoskeleton to Control Dendrite Branching." We would be happy to publish your paper in JCB pending final revisions necessary to meet our formatting guidelines (see details below).

A. MANUSCRIPT ORGANIZATION AND FORMATTING:

1) Text limits: Character count for Articles is < 40,000, not including spaces. Count includes title page, abstract, introduction, results, discussion, and acknowledgments. Count does not include materials and methods, figure legends, references, tables, or supplemental legends.

2) Figure formatting: Articles may have up to 10 main text figures. Scale bars must be present on all microscopy images, including inset magnifications. Please avoid pairing red and green for images and graphs to ensure legibility for color-blind readers. If red and green are paired for images, please ensure that the particular red and green hues used in micrographs are distinctive with any of the colorblind types. If not, please modify colors accordingly or provide separate images of the individual channels.

3) Statistical analysis: Error bars on graphic representations of numerical data must be clearly described in the figure legend. The number of independent data points (n) represented in a graph must be indicated in the legend. Please, indicate whether 'n' refers to technical or biological replicates (i.e. number of analyzed cells, samples or animals, number of independent experiments). If independent experiments with multiple biological replicates have been performed, we recommend using distribution-reproducibility SuperPlots (please see Lord et al., JCB 2020) to better display the distribution of the entire dataset, and report statistics (such as means, error bars, and P values) that address the reproducibility of the findings.

Statistical methods should be explained in full in the materials and methods. For figures presenting pooled data the statistical measure should be defined in the figure legends. Please also be sure to indicate the statistical tests used in each of your experiments (both in the figure legend itself and in a separate methods section) as well as the parameters of the test (for example, if you ran a t-test, please indicate if it was one- or two-sided, etc.). Also, if you used parametric tests, please indicate if the data distribution was tested for normality (and if so, how). If not, you must state something to the effect that "Data distribution was assumed to be normal but this was not formally tested."

4) Materials and methods: Should be comprehensive and not simply reference a previous publication for details on how an experiment was performed. Please provide full descriptions (at least in brief) in the text for readers who may not have access to referenced manuscripts. The text should not refer to methods "...as previously described."

5) For all cell lines, vectors, constructs/cDNAs, etc. - all genetic material: please include database / vendor ID (e.g. Addgene, ATCC, etc.) or if unavailable, please briefly describe their basic genetic features, even if described in other published work or gifted to you by other investigators (and provide references where appropriate). Please be sure to provide the sequences for all of your oligos: primers, si/shRNA, RNAi, gRNAs, etc. in the materials and methods. You must also indicate in the methods the source, species, and catalog numbers/vendor identifiers (where appropriate) for all of your antibodies, including secondary. If antibodies are not commercial, please add a reference citation if possible.

6) Microscope image acquisition: The following information must be provided about the acquisition and processing of images:

- a. Make and model of microscope
- b. Type, magnification, and numerical aperture of the objective lenses
- c. Temperature
- d. Imaging medium
- e. Fluorochromes

f. Camera make and model

g. Acquisition software

h. Any software used for image processing subsequent to data acquisition. Please include details and types of operations involved (e.g., type of deconvolution, 3D reconstitutions, surface or volume rendering, gamma adjustments, etc.).

7) References: There is no limit to the number of references cited in a manuscript. References should be cited parenthetically in the text by author and year of publication. Abbreviate the names of journals according to PubMed.

8) Supplemental materials: Articles generally may have up to 5 supplemental figures and 10 videos. You currently exceed this limit but, in this case, we will be able to give you the extra space. Please also note that tables, like figures, should be provided as individual, editable files. A summary of all supplemental material should appear at the end of the Materials and methods section. Please include one brief sentence per item.

9) Video legends: Should describe what is being shown, the cell type or tissue being viewed (including relevant cell treatments, concentration and duration, or transfection), the imaging method (e.g., time-lapse epifluorescence microscopy), what each color represents, how often frames were collected, the frames/second display rate, and the number of any figure that has related video stills or images.

10) eTOC summary: A ~40-50 word summary that describes the context and significance of the findings for a general readership should be included on the title page. The statement should be written in the present tense and refer to the work in the third person. It should begin with "First author name(s) et al..." to match our preferred style.

11) Conflict of interest statement: JCB requires inclusion of a statement in the acknowledgements regarding competing financial interests. If no competing financial interests exist, please include the following statement: "The authors declare no competing financial interests." If competing interests are declared, please follow your statement of these competing interests with the following statement: "The authors declare no further competing financial interests."

12) A separate author contribution section is required following the Acknowledgments in all research manuscripts. All authors should be mentioned and designated by their first and middle initials and full surnames. We encourage use of the CRediT nomenclature (<https://casrai.org/credit/>).

13) ORCID IDs: ORCID IDs are unique identifiers allowing researchers to create a record of their various scholarly contributions in a single place. Please note that ORCID IDs are required for all authors. At resubmission of your final files, please be sure to provide your ORCID ID and those of all co-authors.

14) Journal of Cell Biology now requires a data availability statement for all research article submissions. These statements will be published in the article directly above the Acknowledgments. The statement should address all data underlying the research presented in the manuscript. Please visit the JCB instructions for authors for guidelines and examples of statements at (<https://rupress.org/jcb/pages/editorial-policies#data-availability-statement>).

B. FINAL FILES:

Additionally, JCB encourages authors to submit a short video summary of their work. These videos are intended to convey the main messages of the study to a non-specialist, scientific audience. Think of them as an extended version of your abstract, or a

short poster presentation. We encourage first authors to present the results to increase their visibility. The videos will be shared on social media to promote your work. For more detailed guidelines and tips on preparing your video, please visit <https://rupress.org/jcb/pages/submission-guidelines#videoSummaries>.

Thank you for your attention to these final processing requirements. Please revise and format the manuscript and upload materials within 7 days. If you need an extension for whatever reason, please let us know and we can work with you to determine a suitable revision period.

Thank you for this interesting contribution, we look forward to publishing your paper in Journal of Cell Biology.

Sincerely,

Louis Reichardt, PhD
Monitoring Editor
Journal of Cell Biology

Dan Simon, PhD
Scientific Editor
Journal of Cell Biology